# Bandit and Delayed Feedback in Online Structured Prediction

**Yuki Shibukawa**
The University of Tokyo
Tokyo, Japan
shibu-yu762@g.ecc.u-tokyo.ac.jp

**Taira Tsuchiya**
The University of Tokyo and RIKEN AIP
Tokyo, Japan
tsuchiya@mist.i.u-tokyo.ac.jp

**Shinsaku Sakaue**[*]
CyberAgent
Tokyo, Japan
shinsaku.sakaue@gmail.com

**Kenji Yamanishi**
The University of Tokyo
Tokyo, Japan
yamanishi@g.ecc.u-tokyo.ac.jp

## Abstract

Online structured prediction is a task of sequentially predicting outputs with complex structures based on inputs and past observations, encompassing online classification. Recent studies showed that in the full-information setting, we can achieve finite bounds on the *surrogate regret*, *i.e.,* the extra target loss relative to the best possible surrogate loss. In practice, however, full-information feedback is often unrealistic as it requires immediate access to the whole structure of complex outputs. Motivated by this, we propose algorithms that work with less demanding feedback, *bandit* and *delayed* feedback. For bandit feedback, by using a standard inverse-weighted gradient estimator, we achieve a surrogate regret bound of $O(\sqrt{KT})$ for the time horizon $T$ and the size of the output set $K$. However, $K$ can be extremely large when outputs are highly complex, resulting in an undesirable bound. To address this issue, we propose another algorithm that achieves a surrogate regret bound of $O(T^{2/3})$, which is independent of $K$. This is achieved with a carefully designed pseudo-inverse matrix estimator. Furthermore, we numerically compare the performance of these algorithms, as well as existing ones. Regarding delayed feedback, we provide algorithms and regret analyses that cover various scenarios, including full-information and bandit feedback, as well as fixed and variable delays.

## 1 Introduction

In many machine learning problems, given an input vector from a set $\mathcal{X}$ of input vectors, we aim to predict a vector in a finite output space $\mathcal{Y}$. Multiclass classification is one of the simplest examples, while in other cases, output spaces may have more complex structures. *Structured prediction* refers to such a class of problems with structured output spaces, including multiclass classification, multilabel classification, ranking, and ordinal regression, and it has applications in various fields, ranging from natural language processing to bioinformatics [2, 43]. In structured prediction, training models that directly predict outputs in complex discrete output spaces is typically challenging. Therefore, we often adopt the *surrogate loss framework* [3]—define an intermediate space of score vectors and train models that estimate score vectors from inputs based on surrogate loss functions. Examples of

---

[*]This work was primarily conducted during the period when SS was affiliated with the University of Tokyo and RIKEN AIP.

Table 1: Upper and lower bounds on the surrogate regret in online multiclass classification and OSP. Here, $T$ is the time horizon, $K = |\mathcal{Y}|$ is the size of the output space, and $D$ is the fixed-delay time. Delayed feedback is considered only when "Delayed" appears in the feedback column. In the target loss column, "SELF*" means SELF that satisfies Assumption 3.5. Note that the $O(T^{2/3})$ bounds for SELF* in lines 6 and 9 do not explicitly depend on $K$ but on $d$; in the case of multiclass classification with the 0-1 loss, the dependence on $K$ appears as $d = K$.

| Setting | Reference | Feedback | Target loss | Surrogate regret bound |
|---|---|---|---|---|
| Binary classification | Van der Hoeven et al. [45, Cor. 1] | Graph bandit | 0-1 loss | $\Omega(\sqrt{T})$ $(d = 2)$ |
| Multiclass classification | Van der Hoeven [44, Thm. 4] | Bandit | 0-1 loss | $O(K\sqrt{T})$ |
|  | Van der Hoeven et al. [45, Thm. 1] | Bandit | 0-1 loss | $O(K\sqrt{T})$ |
| Structured prediction | Sakaue et al. [41, Thms. 7 and 8] | Full-info | SELF | $O(1)$ |
|  | **This work** (Theorems 3.4 and D.2) | Bandit | SELF | $O(\sqrt{KT})$ |
|  | **This work** (Theorem 3.6) | Bandit | SELF* | $O(T^{2/3})$ |
|  | **This work** (Theorems 4.3 and E.3) | Full-info & Delayed | SELF | $O(D^2 + 1)$ |
|  | **This work** (Theorems 4.4 and E.4) | Full-info & Delayed | SELF | $O(D + 1)$ |
|  | **This work** (Theorem 4.5) | Full-info & Delayed | SELF | $\Omega(D + 1)$ |
|  | **This work** (Theorem 5.1) | Bandit & Delayed | SELF | $O(\sqrt{(K + D)T})$ |
|  | **This work** (Theorem 5.2) | Bandit & Delayed | SELF* | $O(D^{1/3}T^{2/3})$ |

surrogate losses include squared, logistic, and hinge losses, and a general framework encompassing them is the *Fenchel–Young loss* [7], which we rely on in this study.

Structured prediction can be naturally extended to the online setting, called *Online Structured Prediction* (OSP) [41]. In OSP, at each round $t = 1, \ldots, T$, an environment selects an input–output pair $(\boldsymbol{x}_t, \boldsymbol{y}_t) \in \mathcal{X} \times \mathcal{Y}$. A learner then predicts $\widehat{\boldsymbol{y}}_t \in \mathcal{Y}$ based on the input $\boldsymbol{x}_t$ and incurs a loss $L(\widehat{\boldsymbol{y}}_t; \boldsymbol{y}_t)$, where $L\colon \mathcal{Y} \times \mathcal{Y} \to \mathbb{R}_{\geq 0}$ is the target loss function. Following prior work [41, 44, 45], we focus on the simple yet fundamental case where the learner's model for estimating score vectors is linear.

The goal of the learner is to minimize the cumulative loss $\sum_{t=1}^{T} L(\widehat{\boldsymbol{y}}_t; \boldsymbol{y}_t)$. On the other hand, the best the learner can do in the surrogate loss framework is to minimize the cumulative surrogate loss, namely $\sum_{t=1}^{T} S(\boldsymbol{U}\boldsymbol{x}_t; \boldsymbol{y}_t)$, where $\boldsymbol{U}\colon \mathcal{X} \to \mathbb{R}^d$ is the best offline linear estimator and $S\colon \mathbb{R}^d \times \mathcal{Y} \to \mathbb{R}_{\geq 0}$ is a surrogate loss, which measures the discrepancy between the score vector $\boldsymbol{U}\boldsymbol{x}_t \in \mathbb{R}^d$ and $\boldsymbol{y}_t \in \mathcal{Y}$. Given this, a natural performance measure of the learner's predictions is the surrogate regret, $\mathcal{R}_T$, defined by $\sum_{t=1}^{T} L(\widehat{\boldsymbol{y}}_t; \boldsymbol{y}_t) = \sum_{t=1}^{T} S(\boldsymbol{U}\boldsymbol{x}_t; \boldsymbol{y}_t) + \mathcal{R}_T$. It has recently attracted increasing attention following the seminal work by Van der Hoeven [44] on online classification. The surrogate regret is an appealing, data-dependent performance measure, as it can provide better bounds on the target loss when the surrogate loss of the best offline estimator, $\sum_{t=1}^{T} S(\boldsymbol{U}\boldsymbol{x}_t; \boldsymbol{y}_t)$, is smaller. Further background and a comparison with the standard regret are provided in Appendix C. Of particular relevance to our work, Sakaue et al. [41] recently obtained a finite surrogate regret bound for online structured prediction (OSP) under full-information feedback, *i.e.,* when the learner observes $\boldsymbol{y}_t$ at the end of each round $t$.

However, the assumption that full-information feedback is available is often demanding, especially when outputs have complex structures. For example, in sequential ad assortment on an advertising platform, we may be able to observe only the click-through rate but not which ads were clicked, which boils down to the *bandit feedback* setting [24, 29]. Also, we may only have access to feedback from a while ago when designing an ad assortment for a new user—namely, *delayed feedback* [32, 47]. Similar situations have led to a plethora of studies in various online learning settings. In combinatorial bandits, algorithms under bandit feedback (referred to as full-bandit feedback in their context), instead of full-information feedback, have been widely studied [9, 14, 18, 39]. Delayed feedback is also explored in various other settings, including full-information and bandit feedback [11, 27]. Due to space limitations, we defer a further discussion of the background to Appendix B.

**Our contributions**  To extend the applicability of OSP, this study develops OSP algorithms that can handle weaker feedback—bandit feedback and delayed feedback—instead of full-information feedback. Following the work of Sakaue et al. [41], we consider the case where target loss functions belong to a class called the Structured Encoding Loss Function (SELF) [6, 12], a general class that includes the 0-1 loss in multiclass classification and the Hamming loss in multilabel classification and ranking (see Section 2.4 for the definition). Table 1 summarizes the surrogate regret bounds provided in this study and comparisons with the existing results.

One of the major challenges in the bandit feedback setting is that the true output $\boldsymbol{y}_t$ is not observed, making it impossible to compute the true gradient of the surrogate loss. To address this, we use an inverse-weighted gradient estimator, a common approach that assigns weights to gradients by the inverse probability of choosing each output, establishing an $O(\sqrt{KT})$ surrogate regret upper bound (Theorems 3.4 and D.2), where $K = |\mathcal{Y}|$ denotes the cardinality of $\mathcal{Y}$. This $O(\sqrt{KT})$ bound has a desirable dependence on $T$, matching an $\Omega(\sqrt{T})$ lower bound known in a problem closely related to online multiclass classification with bandit feedback [45, Corollary 1]. Furthermore, our bound is better than the existing $O(K\sqrt{T})$ bounds [44, 45] by a factor of $\sqrt{K}$, although the bound of Van der Hoeven et al. [45] applies to a broader class of surrogate losses and thus it is not directly comparable to ours (see Appendix C.3 for a more detailed discussion). We also conduct numerical experiments on online multiclass classification and find that our methods, which apply to general OSP, are comparable to the existing ones specialized for multiclass classification (see Appendix H.1).

While the $O(\sqrt{KT})$ bound is satisfactory when $K = |\mathcal{Y}|$ is small, $K$ can be extremely large in some structured prediction problems. In multilabel classification with $m$ correct labels, we have $K = \binom{d}{m}$, and in ranking problems with $m$ items, we have $K = m!$. To address this issue, we consider a special case of SELF (denoted by SELF* in Table 1), which still includes the aforementioned examples: the 0-1 loss in multiclass classification and the Hamming loss in multilabel classification and ranking. A technical challenge to resolve the issue lies in designing an appropriate gradient estimator used in online learning methods. To this end, we draw inspiration from pseudo-inverse estimators used in the adversarial linear/combinatorial bandit literature [1, 9, 16]. Indeed, we cannot naively use the existing estimators, and hence we design a new gradient estimator that applies to various structured prediction problems with target losses belonging to the special SELF class. Armed with this gradient estimator, we achieve a surrogate regret upper bound of $O(T^{2/3})$ (Theorem 3.6). This successfully eliminates the explicit dependence on $K$, although the dependence on $T$ increases compared to the $O(\sqrt{KT})$ bound. We also numerically observe the benefit of this approach when $K$ is large, aligning with the implication of the theoretical bounds (see Appendix H.2).

For the delayed feedback setting, the surrogate regret bounds depend on whether the delay time is fixed or variable. We here describe our results for the known fixed-delay time, denoted by $D$, under the full-information setting. It is relatively straightforward to obtain a surrogate regret bound of $O(\sqrt{(D+1)T})$ with standard Online Convex Optimization (OCO) algorithms for the delayed feedback. We improve this to surrogate regret bounds of $O(\min\{D^2 + 1, (D+1)^{2/3}T^{1/3}\})$ (Theorems 4.3 and E.3) and $O(D+1)$ (Theorems 4.4 and E.4), which are notably independent of $T$. The proofs require carefully bridging different lines of research: OCO algorithms for delayed feedback [20, 27] and surrogate regret analysis in OSP. In addition, we provide the lower bound of $\Omega(D+1)$ (Theorem 4.5), which matches the upper bound.

Given the contributions so far, it is natural to explore OSP in environments where both delay and bandit feedback are present. We develop algorithms for this setting by combining the theoretical developments for bandit feedback without delay and delayed full-information feedback, achieving surrogate regret bounds of $O(\sqrt{(D+K)T})$ (Theorem 5.1) and $O(D^{1/3}T^{2/3})$ (Theorem 5.2). It is worth noting that similar surrogate regret bounds can also be obtained in the variable-delay setting—where the delay may differ for each round of feedback—for both full-information and bandit feedback. Due to space constraints, most of the details are provided in Appendix G.

## 2 Preliminaries

This section describes the detailed setting of OSP and key tools used in this work: the Fenchel–Young loss, SELF, and randomized decoding.

**Notation** For any integer $n > 0$, let $[n] = \{1, 2, \dots, n\}$. Let $\|\cdot\|$ denote a norm with $\kappa\|\boldsymbol{y}\| \geq \|\boldsymbol{y}\|_2$ for some $\kappa > 0$ for any $\boldsymbol{y} \in \mathbb{R}^d$. For a matrix $\boldsymbol{W}$, let $\|\boldsymbol{W}\|_{\mathrm{F}} = \sqrt{\mathrm{tr}(\boldsymbol{W}^\top \boldsymbol{W})}$ be the Frobenius norm. Let $\mathbf{1}$ denote the all-ones vector and $\mathbf{e}_i$ the $i$th standard basis vector. For $\mathcal{K} \subset \mathbb{R}^d$, let $\mathrm{conv}(\mathcal{K})$ be its convex hull and $I_{\mathcal{K}} \colon \mathbb{R}^d \to \{0, +\infty\}$ be its indicator function, which takes zero if the argument is contained in $\mathcal{K}$ and $+\infty$ otherwise. For any function $\Omega \colon \mathbb{R}^d \to \mathbb{R} \cup \{+\infty\}$, let $\mathrm{dom}(\Omega) = \{\boldsymbol{y} \in \mathbb{R}^d : \Omega(\boldsymbol{y}) < +\infty\}$ be its effective domain and $\Omega^*(\boldsymbol{\theta}) = \sup\{\langle \boldsymbol{\theta}, \boldsymbol{y}\rangle - \Omega(\boldsymbol{y}) : \boldsymbol{y} \in \mathbb{R}^d\}$ be its convex conjugate. Table 2 in Appendix A summarizes the notation used in this paper.

## 2.1 Online structured prediction (OSP)

We describe the problem setting of OSP. Let $\mathcal{X}$ be the space of input vectors and $\mathcal{Y}$ be the finite set of outputs. Define $K = |\mathcal{Y}|$. Following the literature [7, 41], we assume that $\mathcal{Y}$ is embedded into $\mathbb{R}^d$ in a standard manner, *e.g.*, $\mathcal{Y} = \{\mathbf{e}_1, \ldots, \mathbf{e}_d\}$ in multiclass classification with $d$ classes.

Let $\mathcal{W}$ be a closed convex domain. A linear estimator $\boldsymbol{W} \in \mathcal{W}$ transforms the input vector $\boldsymbol{x}$ into the score vector $\boldsymbol{W}\boldsymbol{x}$. In OSP, at each round $t = 1, \ldots, T$, the environment selects an input $\boldsymbol{x}_t \in \mathcal{X}$ and the true output $\boldsymbol{y}_t \in \mathcal{Y}$. The learner receives $\boldsymbol{x}_t$ and computes the score vector $\boldsymbol{\theta}_t = \boldsymbol{W}_t \boldsymbol{x}_t$ using the linear estimator $\boldsymbol{W}_t$. Then, the learner selects a prediction $\widehat{\boldsymbol{y}}_t$ based on $\boldsymbol{\theta}_t$, which is called *decoding*, and incurs a loss of $L(\widehat{\boldsymbol{y}}_t; \boldsymbol{y}_t)$. Finally, the learner receives feedback, which depends on the problem setting, and updates $\boldsymbol{W}_t$ to $\boldsymbol{W}_{t+1}$ using some online learning algorithm, denoted by ALG, which is applied to the surrogate loss function $\boldsymbol{W} \mapsto S(\boldsymbol{W}\boldsymbol{x}_t; \boldsymbol{y}_t)$.

The goal of the learner is to minimize the cumulative target loss $\sum_{t=1}^{T} L(\widehat{\boldsymbol{y}}_t; \boldsymbol{y}_t)$, which is equivalent to minimizing the surrogate regret $\mathcal{R}_T = \sum_{t=1}^{T} L(\widehat{\boldsymbol{y}}_t; \boldsymbol{y}_t) - \sum_{t=1}^{T} S(\boldsymbol{U}\boldsymbol{x}_t; \boldsymbol{y}_t)$. We assume that the input and output are generated in an oblivious manner. Note that when $\mathcal{Y} = \{\mathbf{e}_1, \ldots, \mathbf{e}_d\}$ and $L(\widehat{\boldsymbol{y}}_t; \boldsymbol{y}_t) = \mathbb{1}[\widehat{\boldsymbol{y}}_t \neq \boldsymbol{y}_t]$, the above setting reduces to online multiclass classification, which was studied in prior work [44, 45]. Let $B = \mathrm{diam}(\mathcal{W})$ denote the diameter of $\mathcal{W}$ measured by $\|\cdot\|_{\mathrm{F}}$ and $C_x = \max_{\boldsymbol{x} \in \mathcal{X}} \|\boldsymbol{x}\|_2$ the maximum Euclidean norm of input vectors in $\mathcal{X}$.

The feedback observed by the learner depends on the problem setting. The most basic setting is the full-information setting, where the true output $\boldsymbol{y}_t$ is observed as feedback at the end of each round $t$, extensively investigated by Sakaue et al. [41]. By contrast, we study the following weaker feedback: In the *bandit feedback* setting, only the value of the loss function is observed. Specifically, at the end of each round $t$, the learner observes the target loss value $L(\widehat{\boldsymbol{y}}_t; \boldsymbol{y}_t)$ as feedback. In the *delayed feedback* setting, the feedback is observed with a certain delay. In this paper, we consider especially a fixed $D$-round delay setting, *i.e.*, no feedback for round $t \leq D$, and for $t > D$, the learner observes either full-information feedback $\boldsymbol{y}_{t-D}$ or bandit feedback $L(\widehat{\boldsymbol{y}}_{t-D}; \boldsymbol{y}_{t-D})$. We also discuss the variable-delay setting in Appendix G.

In this paper, we make the following assumptions:

**Assumption 2.1.** (1) There exists $\nu > 0$ such that for any distinct $\boldsymbol{y}, \boldsymbol{y}' \in \mathcal{Y}$, it holds that $\|\boldsymbol{y} - \boldsymbol{y}'\| \geq \nu$. (2) For each $\boldsymbol{y} \in \mathcal{Y}$, the target loss function $L(\cdot; \boldsymbol{y})$ is defined on $\mathrm{conv}(\mathcal{Y})$, non-negative, and affine w.r.t. its first argument. (3) There exists $\gamma$ such that for any $\boldsymbol{y}' \in \mathrm{conv}(\mathcal{Y})$ and $\boldsymbol{y} \in \mathcal{Y}$, it holds that $L(\boldsymbol{y}'; \boldsymbol{y}) \leq \gamma \|\boldsymbol{y}' - \boldsymbol{y}\|$ and $L(\boldsymbol{y}'; \boldsymbol{y}) \leq 1$. (4) It holds that $L(\boldsymbol{y}'; \boldsymbol{y}) = 0$ if and only if $\boldsymbol{y}' = \boldsymbol{y}$.

As discussed in [41, Section 2.3], these assumptions are natural and hold for various structured prediction problems and target loss functions, including SELF (see Section 2.4 for the formal definition).

## 2.2 Fenchel–Young loss

We use the Fenchel–Young loss [7] as the surrogate loss, which subsumes many representative surrogate losses, such as the logistic loss, Conditional Random Field (CRF) loss [30], and SparseMAP [35]. See [7, Table 1] for more examples.

**Definition 1** ([7, Fenchel–Young loss]). Let $\Omega \colon \mathbb{R}^d \to \mathbb{R} \cup \{+\infty\}$ be a regularization function with $\mathcal{Y} \subset \mathrm{dom}(\Omega)$. The Fenchel–Young loss generated by $\Omega$, denoted by $S_\Omega \colon \mathrm{dom}(\Omega^*) \times \mathrm{dom}(\Omega) \to \mathbb{R}_{\geq 0}$, is defined as $S_\Omega(\boldsymbol{\theta}; \boldsymbol{y}) = \Omega^*(\boldsymbol{\theta}) + \Omega(\boldsymbol{y}) - \langle \boldsymbol{\theta}, \boldsymbol{y} \rangle$.

The Fenchel–Young loss has the following useful properties:

**Proposition 2.2** ([7, Propositions 2 and 3] and [41, Proposition 3]). *Let* $\Psi \colon \mathbb{R}^d \to \mathbb{R} \cup \{+\infty\}$ *be a differentiable, Legendre-type function[2] that is $\lambda$-strongly convex w.r.t. $\|\cdot\|$, and suppose that* $\mathrm{conv}(\mathcal{Y}) \subset \mathrm{dom}(\Psi)$ *and* $\mathrm{dom}(\Psi^*) = \mathbb{R}^d$. *Define* $\Omega = \Psi + I_{\mathrm{conv}(\mathcal{Y})}$ *and let $S_\Omega$ be the Fenchel–Young loss generated by $\Omega$. For any $\boldsymbol{\theta} \in \mathbb{R}^d$, we define the regularized prediction function as* $\widehat{\boldsymbol{y}}_\Omega(\boldsymbol{\theta}) = \arg\max\{\langle \boldsymbol{\theta}, \boldsymbol{y} \rangle - \Omega(\boldsymbol{y}) \mid \boldsymbol{y} \in \mathbb{R}^d\} = \arg\max\{\langle \boldsymbol{\theta}, \boldsymbol{y} \rangle - \Psi(\boldsymbol{y}) \mid \boldsymbol{y} \in \mathrm{conv}(\mathcal{Y})\}$. *Then, for any $\boldsymbol{y} \in \mathcal{Y}$, $S_\Omega(\boldsymbol{\theta}; \boldsymbol{y})$ is differentiable w.r.t. $\boldsymbol{\theta}$, and it satisfies* $\nabla S_\Omega(\boldsymbol{\theta}; \boldsymbol{y}) = \widehat{\boldsymbol{y}}_\Omega(\boldsymbol{\theta}) - \boldsymbol{y}$. *Furthermore, it holds that* $S_\Omega(\boldsymbol{\theta}; \boldsymbol{y}) \geq \frac{\lambda}{2} \|\boldsymbol{y} - \widehat{\boldsymbol{y}}_\Omega(\boldsymbol{\theta})\|^2$.

---

[2]A function $\Psi$ is called Legendre-type if, for any sequence $x_1, x_2, \ldots$ in $\mathrm{int}(\mathrm{dom}(\Psi))$ that converges to a boundary point of $\mathrm{int}(\mathrm{dom}(\Psi))$, it holds that $\lim_{i \to \infty} \|\nabla \Psi(x_i)\|_2 = +\infty$.

In what follows, let $S_t(\boldsymbol{W}) = S_\Omega(\boldsymbol{W}\boldsymbol{x}_t; \boldsymbol{y}_t)$ for simplicity. Importantly, from the properties of the Fenchel–Young loss, there exists some $b > 0$ such that for any $\boldsymbol{W} \in \mathcal{W}$,

$$\|\nabla S_t(\boldsymbol{W})\|_{\mathrm{F}}^2 \leq b S_t(\boldsymbol{W}). \tag{1}$$

Indeed, from Proposition 2.2, we have $\|\nabla S_t(\boldsymbol{W}_t)\|_{\mathrm{F}}^2 = \|\widehat{\boldsymbol{y}}_\Omega(\boldsymbol{\theta}_t) - \boldsymbol{y}_t\|_2^2 \|\boldsymbol{x}_t\|_2^2 \leq C_x^2\kappa^2 \|\widehat{\boldsymbol{y}}_\Omega(\boldsymbol{\theta}_t) - \boldsymbol{y}_t\|^2 \leq \frac{2C_x^2\kappa^2}{\lambda} S_t(\boldsymbol{W}_t)$, where we used $\nabla S_t(\boldsymbol{W}_t) = (\widehat{\boldsymbol{y}}_\Omega(\boldsymbol{\theta}_t) - \boldsymbol{y}_t)\boldsymbol{x}_t^\top$ and $\|\cdot\|_2 \leq \kappa\|\cdot\|$. Thus, (1) holds with $b = 2C_x^2\kappa^2/\lambda$. Below, let $L_t(\boldsymbol{y}) = L(\boldsymbol{y}; \boldsymbol{y}_t)$ and $\boldsymbol{G}_t = \nabla S_t(\boldsymbol{W}_t) = (\widehat{\boldsymbol{y}}_\Omega(\boldsymbol{\theta}_t) - \boldsymbol{y}_t)\boldsymbol{x}_t^\top$.

## 2.3 Examples of structured prediction

We present several instances of structured prediction along with specific parameter values introduced so far; see [41, Section 2.3] for further details.

**Multiclass classification** Let $\mathcal{Y} = \{\mathbf{e}_1, \ldots, \mathbf{e}_d\}$ and $\|\cdot\| = \|\cdot\|_1$. As the 0-1 loss satisfies $L(\boldsymbol{y}; \mathbf{e}_i) = \mathbb{1}[\boldsymbol{y} \neq \mathbf{e}_i] = \sum_{j\neq i} y_j = \frac{1}{2}(1 - y_i + \sum_{j\neq i} y_j) = \frac{1}{2}\|\mathbf{e}_i - \boldsymbol{y}\|_1$, we have $\gamma = \frac{1}{2}$ in Assumption 2.1. Also, $\nu = 2$ holds since $\|\mathbf{e}_i - \mathbf{e}_j\|_1 = 2$ if $i \neq j$. The logistic surrogate loss is a Fenchel–Young loss $S_\Omega$ generated by the negative Shannon entropy $\Omega = \mathsf{H}^s + I_{\Delta_d}$ (up to a constant factor originating from the base of $\log$), where $\mathsf{H}^s(\boldsymbol{y}) = -\sum_{i=1}^d y_i \log y_i$ and $\Delta_d$ is the $(d-1)$-dimensional probability simplex. Since $\Omega$ is a 1-strongly convex function w.r.t. $\|\cdot\|_1$ on $\Delta_d$, we have $\lambda = 1$.

**Multilabel classification** Let $\mathcal{Y} = \{0,1\}^d$ and $\|\cdot\| = \|\cdot\|_2$. When using the Hamming loss as the target loss function $L(\boldsymbol{y}'; \boldsymbol{y}) = \frac{1}{d}\sum_{i=1}^d \mathbb{1}[y'_i \neq y_i]$, Assumption 2.1 is satisfied with $\nu = 1$ and $\gamma = \frac{1}{d}$. The SparseMAP surrogate loss $S_\Omega(\boldsymbol{\theta}, \boldsymbol{y}) = \frac{1}{2}\|\boldsymbol{y} - \boldsymbol{\theta}\|_2^2 - \frac{1}{2}\|\widehat{\boldsymbol{y}}_\Omega(\boldsymbol{\theta}) - \boldsymbol{\theta}\|_2^2$ is a Fenchel–Young loss generated by $\Omega = \frac{1}{2}\|\cdot\|^2 + I_{\mathrm{conv}(\mathcal{Y})}$. Since $\Omega$ is 1-strongly convex w.r.t. $\|\cdot\|_2$, we have $\lambda = 1$.

**Ranking** We consider predicting the ranking of $m$ items. Let $\|\cdot\| = \|\cdot\|_1$, $d = m^2$, and $\mathcal{Y} \subset \{0,1\}^d$ be the set of all vectors representing $m \times m$ permutation matrices. We use the target loss function that counts mismatches, $L(\boldsymbol{y}'; \boldsymbol{y}) = \frac{1}{m}\sum_{i=1}^m \mathbb{1}[y'_{i,j_i} \neq y_{i,j_i}]$, where $j_i \in [m]$ is a unique index with $y_{ij_i} = 1$ for each $i \in [m]$. In this case, Assumption 2.1 is satisfied with $\nu = 4$ and $\gamma = \frac{1}{2m}$. We use a surrogate loss given by $S_\Omega(\boldsymbol{\theta}; \boldsymbol{y}) = \langle \boldsymbol{\theta}, \widehat{\boldsymbol{y}}_\Omega(\boldsymbol{\theta}) - \boldsymbol{y}\rangle + \frac{1}{\zeta}\mathsf{H}^s(\widehat{\boldsymbol{y}}_\Omega(\boldsymbol{\theta}))$, where $\Omega = -\frac{1}{\zeta}\mathsf{H}^s + I_{\mathrm{conv}(\mathcal{Y})}$ and $\zeta$ controls the regularization strength. The first term in $S_\Omega$ measures the affinity between $\boldsymbol{\theta}$ and $\boldsymbol{y}$, while the second term evaluates the uncertainty of $\widehat{\boldsymbol{y}}_\Omega(\boldsymbol{\theta})$. Since $\Omega$ is $\frac{1}{m\zeta}$-strongly convex, we have $\lambda = \frac{1}{m\zeta}$.

## 2.4 Structured encoding loss function (SELF)

We introduce a general class of target loss functions, called the (generalized) Structured Encoding Loss Function (SELF) [6, 12, 13]. A target loss function is SELF if it can be expressed as

$$L(\boldsymbol{y}_t; \widehat{\boldsymbol{y}}_t) = \langle \widehat{\boldsymbol{y}}_t, \boldsymbol{V}\boldsymbol{y}_t + \boldsymbol{b}\rangle + c(\boldsymbol{y}_t), \tag{2}$$

where $\boldsymbol{b} \in \mathbb{R}^d$ is a constant vector, $\boldsymbol{V} \in \mathbb{R}^{d \times d}$ is a constant matrix, and $c\colon \mathcal{Y} \to \mathbb{R}$ is a function. The following examples of target losses, taken from [6, Appendix A], belong to the SELF class:

- Multiclass classification: the 0-1 loss is a SELF with $\boldsymbol{V} = \mathbf{1}\mathbf{1}^\top - \boldsymbol{I}$, $\boldsymbol{b} = \mathbf{0}$, and $c(\boldsymbol{y}) = 0$.
- Multilabel classification: the Hamming loss, $L(\boldsymbol{y}'; \boldsymbol{y}) = \frac{1}{d}\sum_{i=1}^d \mathbb{1}[y'_{t,i} \neq y_i]$, is a SELF with $\boldsymbol{V} = -\frac{2}{d}\boldsymbol{I}$, $\boldsymbol{b} = \frac{1}{d}\mathbf{1}$, and $c(\boldsymbol{y}) = \frac{1}{d}\langle \boldsymbol{y}, \mathbf{1}\rangle$, where $c(\boldsymbol{y})$ is constant if the number of correct labels is fixed.
- Ranking: the Hamming loss $L(\boldsymbol{y}'; \boldsymbol{y}) = \frac{1}{m}\sum_{i=1}^m \mathbb{1}[y'_{i,j_i} \neq y_{i,j_i}]$, where $j_i \in [m]$ is a unique index with $y_{i,j_i} = 1$ for each $i \in [m]$, is a SELF with $\boldsymbol{V} = -\boldsymbol{I}/m$, $\boldsymbol{b} = \mathbf{0}$, and $c(\boldsymbol{y}) = 1$.

Following the work by Sakaue et al. [41], we assume that the target loss function $L$ is a SELF.

## 2.5 Randomized decoding

We employ the randomized decoding [41], which plays an essential role, particularly in deriving an upper bound independent of the output set size $K = |\mathcal{Y}|$ in Section 3.4. The randomized decoding (Algorithm 1) returns either the closest $\boldsymbol{y}^* \in \mathcal{Y}$ to $\widehat{\boldsymbol{y}}_\Omega(\boldsymbol{\theta}) \in \mathrm{conv}(\mathcal{Y})$ (see Proposition 2.2) or a random $\widetilde{\boldsymbol{y}} \in \mathcal{Y}$ satisfying $\mathbb{E}[\widetilde{\boldsymbol{y}} \mid Z = 1] = \widehat{\boldsymbol{y}}_\Omega(\boldsymbol{\theta})$, where $Z$ follows the Bernoulli distribution with a parameter $p$. Intuitively, the parameter $p$ is chosen so that if $\widehat{\boldsymbol{y}}_\Omega(\boldsymbol{\theta})$ is close to $\boldsymbol{y}^*$, the decoding more likely returns $\boldsymbol{y}^*$; otherwise, it more likely returns $\widetilde{\boldsymbol{y}}$, reflecting uncertainty.

A crucial property of the randomized decoding is the following lemma, which we use in the subsequent analysis:

**Lemma 2.3** ([41, Lemma 4]). *For any $(\boldsymbol{\theta}, \boldsymbol{y}) \in \mathbb{R}^d \times \mathcal{Y}$, the randomized decoding $\phi_\Omega$ satisfies $\mathbb{E}[L(\phi_\Omega(\boldsymbol{\theta}); \boldsymbol{y})] \leq \frac{4\gamma}{\lambda\nu} S_\Omega(\boldsymbol{\theta}; \boldsymbol{y})$, where the expectation is taken w.r.t. the randomness of $\phi_\Omega$.*

*Remark* 1. In randomized decoding, computation of $\widehat{\boldsymbol{y}}_\Omega(\boldsymbol{\theta})$ is the dominant cost, but we can compute it efficiently using a Frank–Wolfe-type algorithm (see *e.g.,* Garber and Wolf [22] and Sakaue et al. [41, Section 3.1] for details).

---

**Algorithm 1** Randomized decoding $\phi_\Omega$

**Input:** $\boldsymbol{\theta} \in \mathbb{R}^d$
1: Compute $\widehat{\boldsymbol{y}}_\Omega(\boldsymbol{\theta})$ defined in Proposition 2.2.
2: $\boldsymbol{y}^* \leftarrow \arg\min\{\|\boldsymbol{y} - \widehat{\boldsymbol{y}}_\Omega(\boldsymbol{\theta})\| : \boldsymbol{y} \in \mathcal{Y}\}$.
3: $\Delta^* \leftarrow \|\boldsymbol{y}^* - \widehat{\boldsymbol{y}}_\Omega(\boldsymbol{\theta})\|$, $p \leftarrow \min\{1, 2\Delta^*/\nu\}$.
4: Sample $Z \sim \mathrm{Ber}(p)$.
5: **if** $Z = 0$ **then** $\widehat{\boldsymbol{y}} \leftarrow \boldsymbol{y}^*$.
6: **if** $Z = 1$ **then** $\widehat{\boldsymbol{y}} \leftarrow \widetilde{\boldsymbol{y}}$ where $\widetilde{\boldsymbol{y}}$ is randomly drawn from $\mathcal{Y}$ so that $\mathbb{E}[\widetilde{\boldsymbol{y}}|Z = 1] = \widehat{\boldsymbol{y}}_\Omega(\boldsymbol{\theta})$.
**Output:** $\phi_\Omega(\boldsymbol{\theta}) = \widehat{\boldsymbol{y}}$

---

## 3 Bandit feedback

In this section, we present two OSP algorithms for the bandit feedback setting and analyze their surrogate regret. Our results can be extended to handle bandit and delayed feedback; see Section 5. Here, we focus on the simpler case without delay to provide a clearer exposition of our core ideas.

### 3.1 Randomized decoding with uniform exploration

We discuss the properties of our decoding function, called *Randomized Decoding with Uniform Exploration (RDUE)*, which will be used in subsequent sections. As discussed in Section 2.5, the randomized decoding (Algorithm 1) was introduced as a decoding function [41] for OSP with full-information feedback. However, naively applying it does not lead to a desired bound under bandit feedback due to the lack of exploration. We extend the randomized-decoding framework to handle bandit feedback effectively.

---

**Algorithm 2** Randomized decoding with uniform exploration (RDUE) $\psi_\Omega$

**Input:** $\boldsymbol{\theta} \in \mathbb{R}^n$, $q \in [0, 1]$
1: Sample $X \sim \mathrm{Ber}(q)$.
2: **if** $X = 0$ **then** $\widehat{\boldsymbol{y}} \leftarrow \phi_\Omega(\boldsymbol{\theta})$.
3: **if** $X = 1$ **then** Sample $\boldsymbol{y}^*$ from $\mathcal{Y}$ uniformly at random and $\widehat{\boldsymbol{y}} \leftarrow \boldsymbol{y}^*$.
**Output:** $\psi_\Omega(\boldsymbol{\theta}) = \widehat{\boldsymbol{y}}$

---

RDUE (Algorithm 2) is a procedure that, with probability $q \in [0, 1]$, selects $\widehat{\boldsymbol{y}}$ uniformly at random from $\mathcal{Y}$, and with probability $1 - q$, selects the output of the randomized decoding. Let $p_t(\boldsymbol{y})$ be the probability that the output of RDUE, $\widehat{\boldsymbol{y}}_t$, coincides with $\boldsymbol{y}$ at round $t$. Note that for any $\boldsymbol{y} \in \mathcal{Y}$, it holds that $p_t(\boldsymbol{y}) \geq \frac{q}{K}$ thanks to the uniform exploration. Furthermore, similar to the property of the randomized decoding in Lemma 2.3, RDUE satisfies the following property:

**Lemma 3.1.** *For any $(\boldsymbol{\theta}, \boldsymbol{y}) \in \mathbb{R}^d \times \mathcal{Y}$, RDUE $\psi_\Omega$ satisfies $\mathbb{E}[L(\psi_\Omega(\boldsymbol{\theta}); \boldsymbol{y})] \leq \frac{4\gamma}{\lambda\nu}(1 - q)S_\Omega(\boldsymbol{\theta}; \boldsymbol{y}) + q\frac{K-1}{K}$, where the expectation is taken w.r.t. the internal randomness of RDUE.*

*Proof.* With probability $1 - q$, the randomized decoding is used; otherwise, a uniformly random output is chosen. Thus, $\mathbb{E}[L(\psi_\Omega(\boldsymbol{\theta}); \boldsymbol{y})] \leq (1 - q)\mathbb{E}[L(\phi_\Omega(\boldsymbol{\theta}); \boldsymbol{y})] + q\frac{K-1}{K}$ holds, where $\phi_\Omega$ is the randomized decoding and we used $L(\cdot; \cdot) \leq 1$. Combining this with Lemma 2.3 completes the proof. $\square$

Based on this lemma, we make the following assumption for convenience:

**Assumption 3.2.** There exists $a \in (0, 1)$ such that $\mathbb{E}_t[L_t(\widehat{\boldsymbol{y}}_t)] \leq (1 - a)S_t(\boldsymbol{W}_t) + q$. Here, $\mathbb{E}_t[\cdot]$ denotes the conditional expectation given the past outputs, $\widehat{\boldsymbol{y}}_1, \ldots, \widehat{\boldsymbol{y}}_{t-1}$.

This assumption can be satisfied by using RDUE for $a \leq 1 - \frac{4\gamma}{\lambda\nu}(1 - q)$ if $\lambda > \frac{4\gamma}{\nu}(1 - q)$, due to Lemma 3.1. In what follows, we set $a = 1 - \frac{4\gamma}{\lambda\nu}$. Note that $\lambda > \frac{4\gamma}{\nu} \geq \frac{4\gamma}{\nu}(1 - q)$ holds in the cases of multiclass classification, multilabel classification, and ranking (see Section 2.3 and [41] for details). The purpose of this assumption is to ensure that a reduction in the surrogate loss leads to a proportional reduction in the target loss.

### 3.2 Online gradient descent

We use the adaptive Online Gradient Descent (OGD) algorithm [42] as ALG, which we apply to surrogate loss $S_t$. OGD updates $\boldsymbol{W}_t$ to $\boldsymbol{W}_{t+1}$ by using the gradient $\boldsymbol{G}_t = \nabla S_t(\boldsymbol{W}_t)$ and learning

rate $\eta_t$ as $\boldsymbol{W}_{t+1} \leftarrow \Pi_{\mathcal{W}}(\boldsymbol{W}_t - \eta_t \boldsymbol{G}_t)$, where $\Pi_{\mathcal{W}}(\boldsymbol{Z}) = \arg\min_{\boldsymbol{X} \in \mathcal{W}} \|\boldsymbol{X} - \boldsymbol{Z}\|_{\mathrm{F}}$. OGD achieves the following bound:

**Lemma 3.3** (*e.g.,* [37, Theorem 4.14]). *Let* $\eta_t = B/\sqrt{2\sum_{i=1}^{t} \|\boldsymbol{G}_i\|_{\mathrm{F}}^2}$. *Then, OGD achieves* $\sum_{t=1}^{T} (S_t(\boldsymbol{W}_t) - S_t(\boldsymbol{U})) \leq \sum_{t=1}^{T} \langle \boldsymbol{G}_t, \boldsymbol{W}_t - \boldsymbol{U} \rangle \leq \sqrt{2}B\sqrt{\sum_{t=1}^{T} \|\boldsymbol{G}_t\|_{\mathrm{F}}^2}$ *for any* $\boldsymbol{U} \in \mathcal{W}$.

### 3.3 Algorithm based on inverse-weighted gradient estimator with $O(\sqrt{KT})$ regret

We present an algorithm that achieves a surrogate regret upper bound of $O(\sqrt{KT})$.

**Algorithm based on inverse-weighted gradient estimator**    In the bandit setting, the true output $\boldsymbol{y}_t$ is not observed, and thus we need to estimate the gradient of $S_t(\boldsymbol{W}_t)$ required for updating $\boldsymbol{W}_t$. To do this, we use the inverse-weighted gradient estimator $\widehat{\boldsymbol{G}}_t = \frac{\mathbb{1}[\widehat{\boldsymbol{y}}_t = \boldsymbol{y}_t]}{p_t(\boldsymbol{y}_t)} \boldsymbol{G}_t$, where $\boldsymbol{G}_t = \nabla S_t(\boldsymbol{W}_t) = (\widehat{\boldsymbol{y}}_{\Omega}(\boldsymbol{\theta}_t) - \boldsymbol{y}_t)\boldsymbol{x}_t^{\top}$. Note that $\widehat{\boldsymbol{G}}_t$ is unbiased, *i.e.,* $\mathbb{E}_t[\widehat{\boldsymbol{G}}_t] = \boldsymbol{G}_t$. We use RDUE with $q = B\sqrt{K/T}$ as the decoding function (assuming $T \geq B^2 K$ for simplicity). For ALG, we employ the adaptive OGD in Section 3.2 with the learning rate of $\eta_t = B/\sqrt{2\sum_{i=1}^{t} \|\widehat{\boldsymbol{G}}_i\|_{\mathrm{F}}^2}$.

*Remark* 2.  This study defines the bandit feedback as the value of the target loss function $L_t(\widehat{\boldsymbol{y}}_t)$. Note, however, that the above algorithm operates using only the weaker feedback of $\mathbb{1}[\widehat{\boldsymbol{y}}_t \neq \boldsymbol{y}_t]$.

**Regret bounds and analysis**    The above algorithm achieves the following surrogate regret bound:

**Theorem 3.4.** *The above algorithm achieves the surrogate regret of* $\mathbb{E}[\mathcal{R}_T] \leq \left(\frac{b}{2a} + 1\right)B\sqrt{KT}$.

This upper bound achieves the rate of $\sqrt{T}$, which matches the existing surrogate regret upper bound for bandit multiclass classification in [45]. Regarding the dependence on $K$, our bound improves the existing $O(K\sqrt{T})$ bound in [44, 45] by a factor of $\sqrt{K}$. Note, however, that the $O(K\sqrt{T})$ bound in [45] applies to a broader class of surrogate loss functions. For example, in $K$-class classification, their bound applies to the base-$k$ logistic loss for $k \leq K$, while ours is restricted to the base-2 logistic loss. A more detailed discussion is given in Appendix C.3. As for tightness, an $\Omega(\sqrt{T})$ lower bound is provided in [45] for the graph feedback setting, a variant of the bandit feedback model. This suggests that the $\sqrt{T}$ dependence would be close to being tight, although this lower bound does not directly apply to the bandit setting. Therefore, whether the rate of $\sqrt{T}$ is improvable or not is left open.

*Proof of Theorem 3.4.* From the convexity of $S_t$ and the unbiasedness of $\widehat{\boldsymbol{G}}_t$, $\mathbb{E}[\sum_{t=1}^{T} (S_t(\boldsymbol{W}_t) - S_t(\boldsymbol{U}))] \leq \mathbb{E}[\sum_{t=1}^{T} \langle \widehat{\boldsymbol{G}}_t, \boldsymbol{W}_t - \boldsymbol{U} \rangle]$. From Lemma 3.3 and Jensen's inequality, this is further upper bounded as $\mathbb{E}[\sum_{t=1}^{T} \langle \widehat{\boldsymbol{G}}_t, \boldsymbol{W}_t - \boldsymbol{U} \rangle] \leq \sqrt{2}B\sqrt{\mathbb{E}[\sum_{t=1}^{T} \|\widehat{\boldsymbol{G}}_t\|_{\mathrm{F}}^2]} \leq B\sqrt{\frac{2bK}{q}\mathbb{E}[\sum_{t=1}^{T} S_t(\boldsymbol{W}_t)]}$, where we used $\mathbb{E}_t[\|\widehat{\boldsymbol{G}}_t\|_{\mathrm{F}}^2] = \frac{\|\boldsymbol{G}_t\|_{\mathrm{F}}^2}{p_t(\boldsymbol{y}_t)} \leq \frac{K}{q}\|\boldsymbol{G}_t\|_{\mathrm{F}}^2 \leq \frac{bK}{q} S_t(\boldsymbol{W}_t)$, which follows from $p_t(\boldsymbol{y}) \geq K/q$ and (1). From Assumption 3.2, $\mathbb{E}[\mathcal{R}_T] \leq \mathbb{E}[\sum_{t=1}^{T} ((1-a)S_t(\boldsymbol{W}_t) - S_t(\boldsymbol{U}))] + qT \leq B\sqrt{\frac{2bK}{q}\mathbb{E}[\sum_{t=1}^{T} S_t(\boldsymbol{W}_t)]} - a\mathbb{E}[\sum_{t=1}^{T} S_t(\boldsymbol{W}_t)] + qT \leq \frac{bB^2K}{2aq} + qT$, where we used $c_1\sqrt{x} - c_2 x \leq c_1^2/(4c_2)$ for $x \geq 0$, $c_1 \geq 0$, and $c_2 > 0$. Plugging $q = B\sqrt{K/T}$ into the last inequality completes the proof. $\square$

We can also prove a surrogate regret bound of $O(\sqrt{KT\log(1/\delta)} + \log(1/\delta))$, which holds with probability $1 - \delta$. The precise statement and proof are provided in Appendix D.2. To prove this high-probability bound, we follow the analysis of Theorem 3.4 and use Bernstein's inequality. To address the challenges posed by the randomness introduced by bandit feedback, we adopt an approach similar to that used in [45], and arguably, we have simplified their analysis.

### 3.4 Algorithm based on pseudo-inverse matrix estimator with $O(T^{2/3})$ regret

We provide an algorithm with a new estimator that achieves a $K$-independent surrogate bound, and we identify the conditions and the class of loss functions under which this new estimator can be used.

While the surrogate regret bound of $O(\sqrt{KT})$ achieves the presumably tight dependence on $T$, the dependence on $K = |\mathcal{Y}|$ is undesirable for general structured prediction. In fact, we have $K = \binom{d}{m}$ in multilabel classification with $m$ correct labels and $K = m!$ in ranking with $m$ items. To address this issue, we present an algorithm that significantly improves the dependence on $K$ when the target loss function belongs to a special class of SELF (2) with the following additional assumptions:

**Assumption 3.5.** (i) The matrix $V$ is known and invertible, and $b$ and $c(\cdot)$ are known and non-negative. (ii) Let $Q = \mathbb{E}_{y \sim \mu}[yy^\top]$, where $\mu$ is the uniform distribution over $\mathcal{Y}$. At least one of the following two conditions holds: (ii-a) $Q$ is invertible, or (ii-b) for any $y \in \mathcal{Y}$, $Vy$ lies in the linear subspace spanned by vectors in $\mathcal{Y}$. (iii) For some $\omega > 0$, it holds that $\mathrm{tr}(V^{-1}Q^+(V^{-1})^\top) \leq \omega$. (iv) For any $\widehat{y}_t \in \mathcal{Y}$ and $y_t \in \mathcal{Y}$, it holds that $|\langle \widehat{y}_t, V y_t \rangle| \leq 1$.

The first and fourth conditions are true in the examples in Section 2.4, assuming that the number of correct labels, $m$, is fixed in multilabel classification. (While the fourth condition does not hold when $m > d/2$ in multilabel classification, we can flip 0 and 1 in the labels to redefine the problem that satisfies the condition.) The second one holds if $\mathcal{Y}$ consists of $d$ linearly independent vectors or $V$ is proportional to the identity matrix; either is true in the examples. Also, deriving reasonable bounds on $\omega$ in those examples is not difficult; see also Appendix D.4 for details.

**Algorithm based on pseudo-inverse matrix estimator**   As with Section 3.3, we consider estimating the gradient. Let $P_t = \mathbb{E}_{y \sim p_t}[yy^\top]$ and define the estimator $\widetilde{y}_t$ of $y_t$ by $\widetilde{y}_t = V^{-1} P_t^+ \widehat{y}_t \langle \widehat{y}_t, V y_t \rangle$, where $P_t^+$ is the Moore–Penrose pseudo-inverse matrix of $P_t$. Note that, given that $b$ and $c(\cdot)$ are known, we can compute $\langle \widehat{y}_t, V y_t \rangle = L_t(\widehat{y}_t) - \langle \widehat{y}_t, b \rangle - c(y_t)$. Importantly, $\widetilde{y}_t$ is unbiased, *i.e.*, $\mathbb{E}_t[\widetilde{y}_t] = y_t$ from the second requirement of Assumption 3.5.

By using this $\widetilde{y}_t$, we define the pseudo-inverse matrix estimator $\widetilde{G}_t$ by $\widetilde{G}_t = (\widehat{y}_\Omega(\theta_t) - \widetilde{y}_t) x_t^\top$, which is also unbiased, *i.e.*, $\mathbb{E}_t[\widetilde{G}_t] = G_t$. Our estimator is inspired by those used in adversarial linear bandits and adversarial combinatorial full-bandits [1, 9, 16].

We use RDUE with $q = (4\omega B^2 C_x^2/T)^{1/3}$ as the decoding function (assuming $T \geq 4\omega B^2 C_x^2$ for simplicity). To update $W_t$, we use OGD in Section 3.2 as ALG with $\eta_t = B/\sqrt{2 \sum_{i=1}^t \|\widetilde{G}_i\|_F^2}$.

**Regret bounds**   This algorithm achieves the following surrogate regret bound independent of $K$:

**Theorem 3.6.** *The above algorithm achieves* $\mathbb{E}[\mathcal{R}_T] = O(\omega^{1/3}T^{2/3})$.

The proof can be found in Appendix D.3. Note that we leverage the structure of OSP when using the pseudo-inverse matrix estimator, which largely differs from the existing approaches to surrogate regret analysis for online classification and OSP [41, 44, 45]. With the pseudo-inverse matrix estimator, we can upper bound the second moment of the gradient estimator $\widetilde{G}_t$ without $K$, which allows for the surrogate regret bound that does not explicitly involves $K$. This is in contrast to the inverse-weighted gradient estimator in Section 3.3. The inverse-weighted gradient estimator involves division by $p_t$, whose lower bound comes from uniform exploration on $\mathcal{Y}$; consequently, its upper bound depends on $K = |\mathcal{Y}|$. In other words, the above pseudo-inverse matrix estimator offers an alternative way to obtain an unbiased gradient estimator while eschewing uniform exploration on $\mathcal{Y}$. However, this comes at the price of a somewhat looser bound on the second moment, which increases the dependence on $T$.

As a corollary of Theorem 3.6, we can derive specific bounds for each problem as follows:

**Corollary 3.7.** *The above algorithm achieves* $\mathbb{E}[\mathcal{R}_T] = O(d^{2/3}T^{2/3})$ *in multiclass classification with the 0-1 loss* $(\omega = d^2)$, $\mathbb{E}[\mathcal{R}_T] = O((d^5/m(d-m))^{1/3}T^{2/3})$ *in multilabel classification with $m$ correct labels and the Hamming loss* $(\omega = d^5/4m(d-m))$, *and* $\mathbb{E}[\mathcal{R}_T] = O(m^{5/3}T^{2/3})$ *in ranking with the number of items $m$ and the Hamming loss* $(\omega = m^5)$.

The proof of Corollary 3.7 is deferred to Appendix D.4. The bound for multilabel classification with $m$ correct labels can be significantly better than the $O(\sqrt{KT})$ bound in Section 3.3 since $K = \sqrt{\binom{d}{m}}$; similarly, the bound for ranking can be much better than the $O(\sqrt{KT})$ bound since $K = \sqrt{m!}$.

**Complexity of computing $P_t^+$**   The matrix $P_t$ equals the sum of $\mathbb{E}_{y \sim p_t}[yy^\top]$ and $\mathbb{E}_{y \sim \mu}[yy^\top]$. The expectation $\mathbb{E}_{y \sim p_t}[yy^\top]$ can be calculated analytically in the multiclass and multilabel clas-

sification and ranking. For $\mathbb{E}_{\boldsymbol{y}\sim\mu}[\boldsymbol{y}\boldsymbol{y}^\top]$, when $\widehat{\boldsymbol{y}}_\Omega(\boldsymbol{\theta})$ is obtained via the Frank–Wolfe algorithm, $p_t$ is obtained from the convex combination coefficients, whose support size is at most $O(d)$ as implied by Carathéodory's theorem (cf. [4]). Therefore, we can compute $\boldsymbol{P}_t$ in $O(d^3)$ time, and the pseudo-inversion takes the same order of complexity.

## 4 Delayed full-information feedback

This section discusses OSP with fixed-delay full-information feedback and presents two algorithms that achieve surrogate regret bounds of $O(\min\{D^2 + 1,\ (D + 1)^{2/3}T^{1/3}\})$ and $O(D + 1)$, which are better than the $O(\sqrt{(D+1)T})$ bound obtained by a standard OCO algorithm under delayed feedback [27]. Although the first upper bound is worse than the second, we include it here as a preliminary step toward the algorithm for the delayed and bandit feedback setting described in Section 5.

Below, we make the following assumption based on the randomized decoding of [41].

**Assumption 4.1.** There exists a constant $a \in (0, 1)$ that satisfies $\mathbb{E}_t[L_t(\widehat{\boldsymbol{y}}_t)] \leq (1 - a)S_t(\boldsymbol{W}_t)$.

From Lemma 2.3, if $\lambda > \frac{4\gamma}{\nu}$, this condition is satisfied with $a = 1 - \frac{4\gamma}{\lambda\nu}$ by using the randomized decoding. We suppose that such a decoding function is used in this section.

### 4.1 Algorithm based on ODAFTRL with $O(\min\{D^2 + 1, (D + 1)^{2/3}T^{1/3}\})$ regret

**Algorithm** We employ the Optimistic Delayed Adaptive FTRL algorithm (ODAFTRL) [20] as ALG, which we detail in Appendix E.1 for completeness. ODAFTRL computes the linear estimator by $\boldsymbol{W}_{t+1} \in \arg\min_{\boldsymbol{W}\in\mathcal{W}}\{\sum_{i=1}^{t-D}\langle\boldsymbol{G}_i, \boldsymbol{W}\rangle + \frac{\lambda_t}{2}\|\boldsymbol{W}\|_F^2\}$, where $\lambda_t \geq 0$ is the regularization parameter. By updating $\lambda_t$ using an AdaHedge-type algorithm (AdaHedgeD), ODAFTRL achieves the following AdaGrad-type regret upper bound:

**Lemma 4.2** (Informal version of [20, Theorem 12]). *Consider the delayed full-information setting. For any $\boldsymbol{U} \in \mathcal{W}$, ODAFTRL with the AdaHedgeD update of $\lambda_t$ achieves a regret bound of* $\sum_{t=1}^T (S_t(\boldsymbol{W}_t) - S_t(\boldsymbol{U})) = O\Big(\sqrt{\sum_{t=1}^T \|\boldsymbol{G}_t\|_F^2} + D\sum_{t=1}^T\sum_{s=t-D}^t\|\boldsymbol{G}_s\|_F^2\Big).$

**Regret bounds and analysis** The above algorithm achieves the following surrogate regret bound:

**Theorem 4.3.** *The above algorithm achieves* $\mathbb{E}[\mathcal{R}_T] = O(\min\{D^2 + 1, (D + 1)^{2/3}T^{1/3}\})$.

Recall that the proof ideas for deriving surrogate regret bounds in the non-delayed setting [41, 45] differ from those in the standard OCO and multi-armed bandits, and thus we cannot naively extend the analyses of the algorithms for delayed feedback in those settings to our case. Below is the proof sketch, and the complete proof is given in Appendix E.2.

*Proof sketch.* From Lemma 4.2 with $\|\boldsymbol{G}_t\|_F^2 \leq bS_t(\boldsymbol{W}_t)$ in (1) and Cauchy–Schwarz, we have $\sum_{t=1}^T (S_t(\boldsymbol{W}_t) - S_t(\boldsymbol{U})) = O(D\sqrt{S_{1:T}})$, where $S_{1:T} = \sum_{t=1}^T S_t(\boldsymbol{W}_t)$. Thus, Assumption 4.1 implies $\mathbb{E}[\mathcal{R}_T] \leq \sum_{t=1}^T (S_t(\boldsymbol{W}_t) - S_t(\boldsymbol{U})) - a\sum_{t=1}^T S_t(\boldsymbol{W}_t) = O(D\sqrt{S_{1:T}}) - aS_{1:T} = O(D^2)$, where we used $c_1\sqrt{x} - c_2x \leq c_1^2/(4c_2)$ for $x \geq 0$, $c_1 \geq 0$, and $c_2 > 0$. $\square$

We can also prove a high-probability surrogate regret bound of $O(\min\{D^2 + 1, (D + 1)^{2/3}T^{1/3}\} + \log(1/\delta))$, which holds with probability at least $1 - \delta$. See Appendix E.3 for the proof.

### 4.2 Algorithm based on BOLD with $O(D + 1)$ regret

**Algorithm** We use the Black-box Online Learning under Delayed feedback (BOLD) [27] as ALG. BOLD constructs $D + 1$ independent instances of any deterministic non-delayed online learning algorithm (called BASE) denoted as $\text{BASE}_0, \text{BASE}_1, \ldots, \text{BASE}_D$. This algorithm selects which instance to use according to the value of remainder $r_t \in \{0, \ldots, D\}$, which satisfies $r_t = t - k(D+1)$ for some $k \in \mathbb{Z}_{\geq 0}$. At each round $t$, BOLD invokes $\text{BASE}_{r_t}$. Here, we adopt OGD as BASE. The pseudocode of BOLD is given in Appendix E.4.

**Regret bound** The above algorithm achieves the following surrogate regret bound:

**Theorem 4.4.** *The above algorithm achieves* $\mathbb{E}[\mathcal{R}_T] = O(D + 1)$.

The proof is given in Appendix E.4. The upper bound in Theorem 4.4 matches the following lower bound, whose proof is provided in Appendix E.5.

**Theorem 4.5.** *Let $d \geq 2$. For $B = \Omega(\log(dT))$, there exists a sequence $\{(\boldsymbol{x}_t, \boldsymbol{y}_t)\}_{t=1}^T$ with $\|\boldsymbol{x}_t\|_2 = 1$ such that the surrogate regret with respect to the logistic surrogate loss of any possibly randomized algorithm is lower bounded by $\mathbb{E}[\mathcal{R}_T] = \Omega\big(B^2(D+1)/(\log d)^2\big)$.*

## 5 Delayed bandit feedback

Given the results so far, it is natural to explore OSP with delayed bandit feedback. We construct algorithms for this setting by combining the theoretical developments from Sections 3 and 4.1.

### 5.1 Algorithm for bandit delayed feedback with $O(\sqrt{(K+D)T})$ regret

We adopt RDUE with $q = B\sqrt{K/T}$ for decoding (assuming $T \geq B^2K$), the inverse-weighted gradient estimator $\widehat{\boldsymbol{G}}_t$, and ODAFTRL with AdaHedgeD as ALG. Then, the following bound holds:

**Theorem 5.1.** *The above algorithm achieves* $\mathbb{E}[\mathcal{R}_T] = O(\sqrt{(K+D)T})$.

The proof can be found in Appendix F.2. This upper bound incurs an additional additive $O(\sqrt{DT})$ factor compared to the bound in the non-delayed case in Theorem 3.4. Whether this surrogate regret upper bound is optimal remains open. While an $\Omega(\sqrt{T})$ surrogate regret lower bound exists in the graph feedback setting [45], no such lower bound is known for the bandit non-delayed setting, and constructing lower bounds under delayed feedback would be more difficult.

### 5.2 Algorithm for bandit delayed feedback with $O(D^{1/3}T^{2/3})$ regret

We provide an algorithm that improves the dependence on $K$ from Section 5.1. We make the same assumptions on the target loss function as Section 3.4. We use RDUE with $q = \big(\omega B^2 C_x^2 D/T\big)^{1/3}$ for decoding (assuming $T \geq \omega B^2 C_x^2 D$), the pseudo-inverse matrix estimator $\widetilde{\boldsymbol{G}}_t$, and ODAFTRL with the AdaHedgeD update as ALG. Then, the following bound holds:

**Theorem 5.2.** *The above algorithm achieves* $\mathbb{E}[\mathcal{R}_T] = O(D^{1/3}T^{2/3})$.[3]

The proof can be found in Appendix F.3. Due to the presence of the delay, the surrogate regret bound worsens by a factor of $D^{1/3}$ compared to the non-delayed bandit setting. Additionally, we present algorithms for the variable-delay setting, which we defer to Appendix G due to space limitations.

The algorithm in this section employs ODAFTRL rather than BOLD for the following reasons. First, ODAFTRL leads to at least as good regret upper bounds as BOLD under bandit delayed feedback. BOLD-based algorithms with the inverse-weighted gradient estimator and the pseudo-inverse matrix estimator attain regret upper bounds of $O(\sqrt{KDT})$ and $O(D^{1/3}T^{2/3})$, respectively, which are not better than the bounds in Theorems 5.1 and 5.2 obtained with ODAFTRL. Second, as noted by Flaspohler et al. [20], approaches such as BOLD, which run multiple parallel instances, cause each instance to operate independently and observe only $T/(D+1)$ losses. This reduction can significantly worsen empirical performance, particularly when $T$ is not very large relative to $D$.

## 6 Conclusion

We have developed several algorithms for online structured prediction under bandit and delayed feedback and analyzed their surrogate regret. Among these contributions, of particular note is the algorithm for bandit feedback whose surrogate regret bound does not explicitly depend on the output set size $K$, achieved by leveraging the pseudo-inverse matrix estimator. An important direction for future work is to investigate the corresponding lower bounds in the bandit feedback setting. The existing lower bound of $\Omega(\sqrt{T})$ in the graph-feedback setting [45] suggests that a similar bound likely holds here as well; however, this has yet to be proven for our settings, and the tightness of our upper bounds remains an open question.

---

[3]Here, unlike in the previous sections, we use $D$ instead of $D + 1$, since this algorithm is not intended to handle the non-delayed case of $D = 0$.

## Acknowledgments and Disclosure of Funding

The authors would like to express their sincere gratitude to the anonymous reviewers for their insightful feedback and constructive suggestions, which have significantly improved the manuscript, particularly the discussion of the upper bound under delayed feedback. TT is supported by JST ACT-X Grant Number JPMJAX210E and JSPS KAKENHI Grant Number JP24K23852, SS was supported by JST ERATO Grant Number JPMJER1903, and KY is supported by JSPS KAKENHI Grant Number JP24H00703.

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

# A  Notation

Table 2 summarizes the symbols used in this paper.

Table 2: Notation

| Symbol | Meaning |
|---|---|
| $T \in \mathbb{N}$ | Time horizon |
| $d \in \mathbb{N}$ | Dimension of output space $\mathcal{Y}$ |
| $B = \mathrm{diam}(\mathcal{W})$ | Diameter of $\mathcal{W}$ |
| $C_x = \max_{\boldsymbol{x} \in \mathcal{X}} \|\boldsymbol{x}\|_2$ | Maximum norm of input vectors in $\mathcal{X}$ |
| $C_y$ | The maximum of the largest Euclidean norm of vectors in $\mathrm{conv}(\mathcal{Y})$ or the diameter of $\mathrm{conv}(\mathcal{Y})$ |
| $K = |\mathcal{Y}|$ | Cardinality of $\mathcal{Y}$ |
| ALG | Algorithm for updating linear estimators |
| $\widehat{\boldsymbol{y}}_t \in \mathcal{Y}$ | Output chosen by the learner at round $t$ |
| $p_t(\boldsymbol{y})$ | Probability that $\boldsymbol{y}$ is chosen as $\widehat{\boldsymbol{y}}_t$ at time $t$ |
| $L \colon \mathcal{Y} \times \mathcal{Y} \to \mathbb{R}_{\geq 0}$ | Target loss function |
| $L_t(\widehat{\boldsymbol{y}}_t) = L(\widehat{\boldsymbol{y}}_t; \boldsymbol{y}_t)$ | Value of target loss $L(\widehat{\boldsymbol{y}}_t; \boldsymbol{y}_t)$ |
| $S_\Omega \colon \mathbb{R}^n \times \mathcal{Y} \to \mathbb{R}_{\geq 0}$ | Fenchel–Young loss generated by $\Omega$ |
| $S_t(\boldsymbol{W}) = S_\Omega(\boldsymbol{W}\boldsymbol{x}_t; \boldsymbol{y}_t)$ | Shorthand of surrogate loss $S_\Omega(\boldsymbol{W}\boldsymbol{x}_t; \boldsymbol{y}_t)$ |
| $\mathcal{R}_T = \sum_{t=1}^{T} \left( L_t(\widehat{\boldsymbol{y}}_t) - S_t(\boldsymbol{U}) \right)$ | Surrogate regret |
| $\boldsymbol{G}_t = \nabla S_t(\boldsymbol{W}_t)$ | Gradient of surrogate loss |
| $\widehat{\boldsymbol{G}}_t$ | Inverse weighted estimator |
| $\widetilde{\boldsymbol{G}}_t$ | Pseudo-inverse matrix estimator |
| $\mu$ | Uniform distribution over $\mathcal{Y}$ |
| $\boldsymbol{P}_t = \mathbb{E}_{\boldsymbol{y} \sim p_t}[\boldsymbol{y}\boldsymbol{y}^\top]$ | Second moment matrix under $p_t$ |
| $\boldsymbol{Q} = \mathbb{E}_{\boldsymbol{y} \sim \mu}[\boldsymbol{y}\boldsymbol{y}^\top]$ | Second moment matrix under $\mu$ |
| $\lambda_{\min}(\boldsymbol{A})$ | Minimum eigenvalue of matrix $\boldsymbol{A}$ |
| $\omega$ | Upper bound of $\mathrm{tr}(\boldsymbol{V}^{-1}\boldsymbol{Q}^+\boldsymbol{V})$ |
| $\mathbb{E}_t[\cdot]$ | Conditional expectation given $\widehat{\boldsymbol{y}}_1, \ldots, \widehat{\boldsymbol{y}}_{t-1}$ |
| $D \in \mathbb{N}$ | Fixed-delay time |
| $\tau_t$ | Variable delay time |
| $\tau_* = \max_t \tau_t$ | Maximum value of delay time |
| $\rho(t)$ | Time step of the $t$th feedback from SOLID to BASE |
| $\widetilde{\tau}_t = t - 1 - \sum_{s=1}^{\rho(t)-1} \mathbb{1}[s + \tau_s < \rho(t)]$ | The number of feedback from SOLID to BASE pending during the $t$th feedback |

# B  Additional related work

We discuss additional related work that could not be included in the main text.

**Structured prediction**  Before the development of the Fenchel–Young loss framework, Niculae et al. [35] proposed SparseMAP, which used the squared $\ell_2$-norm regularization. The Fenchel–Young loss, described in Section 2.2, is built upon the idea of SparseMAP. The Structure Encoding Loss Function (SELF) was introduced by Ciliberto et al. [12, 13] to analyze the relationship between surrogate and target losses, a concept known as Fisher consistency. For a more extensive literature review, we refer the reader to Sakaue et al. [41, Appendix A].

**Online classification with full and bandit feedback** In the full-information setting, the perceptron is one of the most representative algorithms for binary classification [40], and the multiclass setting has also been extensively studied [15, 19]. Online logistic regression is another relevant research stream, with Foster et al. [21] being a particularly representative study. The study of the bandit setting was initiated by Kakade et al. [29], and it has since been extensively explored in subsequent research [5, 21, 25]. However, to the best of our knowledge, no prior work has addressed general structured prediction under bandit feedback. One of the most relevant studies is the work by Gentile and Orabona [24], who investigated online multilabel classification and ranking. However, their setting assumes access to feedback of the form $\{\mathbb{1}[\boldsymbol{y}_{t,i} \neq \widehat{\boldsymbol{y}}_{t,i}]\}_i$, which is more informative than bandit feedback and differs from our setting. Van der Hoeven [44] introduced the surrogate regret in the context of online multiclass classification. This study has been extended to the setting where observations are determined by a directed graph [45] and to structured prediction [41]. For a more extensive overview of the literature on online classification, we refer the reader to Van der Hoeven [44].

**Delayed feedback** The study of delayed feedback was initiated by Weinberger and Ordentlich [47]. Since then, it has been extensively explored in various online learning settings, primarily in the full-information setting of online convex optimization [20, 27, 28, 34]. Algorithms for delayed bandit feedback have been studied mainly in the context of multi-armed bandits and their variants [11, 26, 33, 46, 48]. In online classification, research considering delay is scarce; the only work is that of Manwani and Agarwal [32] to our knowledge. There are several differences between their work and ours. Among them, a key distinction is that their study focuses on multiclass classification, whereas we address the more general OSP.

## C  Discussion on the surrogate regret

Our work employs the surrogate regret as the performance measure, which represents the excess target loss relative to the surrogate loss achieved by the best offline estimator. This differs from the standard regret, which is defined solely in terms of the target loss. Below, we discuss the motivation and background of the surrogate regret, and compare it to the standard regret.

### C.1  Background and motivation

Although the term "surrogate regret" has only recently come into use, its concept dates back to the classic analysis of the perceptron [36, 40]. Specifically, the celebrated convergence of the perceptron under linear separability can be interpreted as a finite upper bound on the surrogate regret, where the hinge loss of the best offline estimator, $\sum_{t=1}^{T} S_t(\boldsymbol{U})$, equals zero; see Orabona [37, Section 8.2]. Since then, similar performance measures have continued to attract considerable attention in the literature [8, 10, 19, 21, 23, 29]. The concept of the surrogate regret was highlighted in the recent work by Van der Hoeven [44] on online classification, and the terminology was explicitly used in the subsequent work by Van der Hoeven et al. [45]. Later, Sakaue et al. [41] extended this concept to online structured prediction.

The surrogate regret is designed to evaluate how small the cumulative target loss can be made, sharing the same spirit as the standard regret in this regard. The main motivation for using the surrogate regret lies in the empirical observation that the cumulative surrogate loss can often be made very small. An extreme case is the linearly separable setting considered in the convergence analysis of the perceptron, where $\sum_{t=1}^{T} S_t(\boldsymbol{U}) = 0$ holds for the hinge loss. Thus, the surrogate regret naturally captures the data-dependent easiness of a problem and yields better upper bounds on the cumulative target loss as the cumulative surrogate loss becomes smaller.

### C.2  Comparison to the standard regret

In online classification, given a hypothesis class $\mathcal{H}$ consisting of mappings from $\mathcal{X}$ to $\Delta_d$, the standard regret is defined as $\sum_{t=1}^{T} \mathbb{1}[\widehat{\boldsymbol{y}}_t \neq \boldsymbol{y}_t] - \inf_{h \in \mathcal{H}} \sum_{t=1}^{T} \mathbb{1}[h(\boldsymbol{x}_t) \neq \boldsymbol{y}_t]$. Unlike the surrogate regret, it is defined solely in terms of the target loss. At a conceptual level, the standard regret focuses more on worst-case analysis under the agnostic setting, whereas the surrogate regret is designed to benefit from data-dependent analysis, as discussed above. For the standard regret, Daniely et al. [17] established a lower bound of $\Omega(\sqrt{\mathrm{Ldim}(\mathcal{H})T})$, where $\mathrm{Ldim}(\mathcal{H})$ denotes the Littlestone dimension

of $\mathcal{H}$. This does not contradict the finite upper bound on the surrogate regret, since the cumulative surrogate loss may grow with $T$.

### C.3 Discussion on the difference in surrogate loss functions

As in Section 1, the surrogate regret, $\mathcal{R}_T$, is defined by $\sum_{t=1}^{T} L(\widehat{\boldsymbol{y}}_t; \boldsymbol{y}_t) = \sum_{t=1}^{T} S(\boldsymbol{U}\boldsymbol{x}_t; \boldsymbol{y}_t) + \mathcal{R}_T$, which means the choice of the surrogate loss function, $S$, affects the bound on the cumulative loss $\sum_{t=1}^{T} L(\widehat{\boldsymbol{y}}_t; \boldsymbol{y}_t)$. Van der Hoeven et al. [45, Theorem 1], which applies to a more general setting than bandit feedback, implies $\mathcal{R}_T = O(K\sqrt{T})$ for the bandit setting with $S$ being a logistic loss defined with the base-$K$ logarithm. On the other hand, our bound of $\mathcal{R}_T = O(\sqrt{KT})$ applies to the logistic loss $S$ defined with the base-2 logarithm. As a result, while our bound on $\mathcal{R}_T$ is better, the $\sum_{t=1}^{T} S(\boldsymbol{U}\boldsymbol{x}_t; \boldsymbol{y}_t)$ term can be worse; this is why we cannot directly compare our $O(\sqrt{KT})$ bound with the $O(K\sqrt{T})$ bound in Van der Hoeven et al. [45, Theorem 1]. We may use the decoding procedure in Van der Hoeven et al. [45], instead of RDUE, to recover their bound that applies to the base-$K$ logistic loss. It should be noted that their method is specific to multiclass classification; naively extending their method to structured prediction formulated as $|\mathcal{Y}|$-class classification results in the undesirable dependence on $K = |\mathcal{Y}|$, as is also discussed in Sakaue et al. [41]. By contrast, our pseudo-inverse estimator, combined with RDUE, can rule out the explicit dependence on $K$, at the cost of the increase from $\sqrt{T}$ to $T^{2/3}$.

## D  Details omitted from Section 3

This section provides the omitted details of Section 3.

### D.1  Concentration inequality

To prove high probability regret bounds, we will use the following concentration inequality.

**Lemma D.1** (Bernstein's inequality, *e.g.,* [8, Lemma A.8])**.** *Let $Z_1, \ldots, Z_T$ be a martingale difference sequence and $\delta \in (0, 1)$. If there exist $a$ and $v$ which satisfy $|Z_t| \le a$ for any $t \in [T]$ and $\sum_{t=1}^{T} \mathbb{E}_t\big[Z_t^2\big] \le v$ , then with probability at least $1 - \delta$, it holds that*

$$\sum_{t=1}^{T} Z_t \le \sqrt{2v \log \frac{1}{\delta}} + \frac{\sqrt{2}}{3} a \log \frac{1}{\delta}.$$

### D.2  Proof of high probability bound

Here, we provide the proof of a high probability bound. Hereafter, we let $S_{\max} = \max_{\boldsymbol{W} \in \mathcal{W}} S_t(\boldsymbol{W})$ and $\widehat{S}_t(\boldsymbol{W}) = v_t S_t(\boldsymbol{W}) = \frac{\mathbb{1}[\boldsymbol{y}_t = \widehat{\boldsymbol{y}}_t]}{p_t(\widehat{\boldsymbol{y}}_t)} S_t(\boldsymbol{W})$. The following theorem is the formal version of the high probability bound under the bandit feedback:

**Theorem D.2.** *Consider the bandit and non-delayed setting. Let*

$$\mathcal{C} = \left( \frac{3}{2(a + \xi - 1)} + 1 \right) K S_{\max} \log(2/\delta) + \frac{B^2 K b}{2(1 - \xi)}.$$

*Then, for any $T \ge \mathcal{C}$ and $\delta \in (0, 1/2)$, with probability at least $1 - \delta$, the algorithm in Section 3.3 with $q = \sqrt{\mathcal{C}/T}$ achieves*

$$\mathcal{R}_T \le 2\sqrt{\mathcal{C}T} + \sqrt{2 \log(2/\delta)} (\mathcal{C}T)^{1/4} + \left( \frac{1 - a}{2(a + \xi - 1)} + 2 \right) \log(2/\delta).$$

Before proving this theorem, we provide the following lemma:

**Lemma D.3.** *It holds that*

$$\sum_{t=1}^{T} \Big( \mathbb{E}_t[L_t(\widehat{\boldsymbol{y}}_t)] - \widehat{S}_t(\boldsymbol{U}) \Big) \le \sum_{t=1}^{T} \Big( (1 - a) S_t(\boldsymbol{W}_t) - \widehat{S}_t(\boldsymbol{W}_t) \Big) + qT + \sqrt{2}B \sqrt{\frac{b}{q} \sum_{t=1}^{T} v_t S_t(\boldsymbol{W}_t)}.$$

*Proof.* We have

$$\sum_{t=1}^{T}\Big(\mathbb{E}_t[L_t(\widehat{\boldsymbol{y}}_t)] - \widehat{S}_t(\boldsymbol{U})\Big) = \sum_{t=1}^{T}\Big(\mathbb{E}_t[L_t(\widehat{\boldsymbol{y}}_t)] - \widehat{S}_t(\boldsymbol{W}_t)\Big) + \sum_{t=1}^{T}\Big(\widehat{S}_t(\boldsymbol{W}_t) - \widehat{S}_t(\boldsymbol{U})\Big).$$

From Assumption 3.2, the first term is bounded as

$$\sum_{t=1}^{T}\Big(\mathbb{E}_t[L_t(\widehat{\boldsymbol{y}}_t)] - \widehat{S}_t(\boldsymbol{W}_t)\Big) \le \sum_{t=1}^{T}\Big((1-a)S_t(\boldsymbol{W}_t) - \widehat{S}_t(\boldsymbol{W}_t)\Big) + qT,$$

and the second term is bounded as

$$\sum_{t=1}^{T}\Big(\widehat{S}_t(\boldsymbol{W}_t) - \widehat{S}_t(\boldsymbol{U})\Big) \le \sqrt{2}B\sqrt{\sum_{t=1}^{T}\|\widehat{\boldsymbol{G}}_t\|_{\mathrm{F}}^2} = \sqrt{2}B\sqrt{\sum_{t=1}^{T}v_t^2\|\boldsymbol{G}_t\|_{\mathrm{F}}^2}$$

$$\le \sqrt{2}B\sqrt{b\sum_{t=1}^{T}v_t^2 S_t(\boldsymbol{W}_t)} \le \sqrt{2}B\sqrt{\frac{bK}{q}\sum_{t=1}^{T}v_t S_t(\boldsymbol{W}_t)},$$

where we used Lemma 3.3 and $v_t \le K/q$. Combining the above three, we obtain

$$\sum_{t=1}^{T}\Big(\mathbb{E}_t[L_t(\widehat{\boldsymbol{y}}_t)] - \widehat{S}_t(\boldsymbol{U})\Big) \le \sum_{t=1}^{T}\Big((1-a)S_t(\boldsymbol{W}_t) - \widehat{S}_t(\boldsymbol{W}_t)\Big) + qT + \sqrt{2}B\sqrt{\frac{bK}{q}\sum_{t=1}^{T}v_t S_t(\boldsymbol{W}_t)},$$

which completes the proof. $\qquad\square$

*Proof of Theorem D.2.* The surrogate regret can be decomposed as

$$\mathcal{R}_T = \sum_{t=1}^{T}(L_t(\widehat{\boldsymbol{y}}_t) - \mathbb{E}_t[L_t(\widehat{\boldsymbol{y}}_t)]) + \sum_{t=1}^{T}(\mathbb{E}_t[L_t(\widehat{\boldsymbol{y}}_t)] - S_t(\boldsymbol{U})). \tag{3}$$

We first upper bound the first term in (3). Let $Z_t = L_t(\widehat{\boldsymbol{y}}_t) - \mathbb{E}_t[L_t(\widehat{\boldsymbol{y}}_t)]$ for simplicity. Then, we have $Z_t \le 1$, $\mathbb{E}_t[Z_t] = 0$, and $\mathbb{E}_t[Z_t^2] \le \mathbb{E}_t[(L_t(\widehat{\boldsymbol{y}}_t))^2] \le (1-a)S_t(\boldsymbol{W}_t) + q$. Hence, from Bernstein's inequality in Lemma D.1, for any $\delta' \in (0,1)$, at least $1 - \delta'$ we have

$$\sum_{t=1}^{T}Z_t \le \sqrt{2\log(1/\delta')\sum_{t=1}^{T}((1-a)S_t(\boldsymbol{W}_t) + q)} + \frac{\sqrt{2}}{3}\log(1/\delta'). \tag{4}$$

We next consider the second term in (3). Define $r_t = S_t(\boldsymbol{U}) - \xi S_t(\boldsymbol{W}_t)$ for some $\xi \in (0,1)$, which will be determined later, and let $v_t = \mathbb{1}[\boldsymbol{y}_t = \widehat{\boldsymbol{y}}_t]/p_t(\widehat{\boldsymbol{y}}_t) \le K/q$ for simplicity. Then, we have $\mathbb{E}_t[v_t r_t - r_t] = 0$, $|v_t r_t - r_t| \le KS_{\max}/q$, and

$$\mathbb{E}_t[(v_t r_t - r_t)^2] \le \mathbb{E}_t[(v_t r_t)^2] \le \frac{KS_{\max}}{q}|r_t| \le \frac{KS_{\max}}{q}(S_t(\boldsymbol{U}) + S_t(\boldsymbol{W}_t)).$$

Hence, from Bernstein's inequality in Lemma D.1, for any $\delta'' \in (0,1)$, with probability at least $1 - \delta''$ we have

$$\sum_{t=1}^{T}(v_t r_t - r_t) \le \sqrt{3\log(1/\delta'')\sum_{t=1}^{T}\frac{KS_{\max}}{q}(S_t(\boldsymbol{U}) + S_t(\boldsymbol{W}_t))} + \frac{\sqrt{2}KS_{\max}}{3q}\log(1/\delta''). \tag{5}$$

Below, we proceed by case analysis.

**When $\sum_{t=1}^{T}S_t(\boldsymbol{U}) \le \sum_{t=1}^{T}S_t(\boldsymbol{W}_t)$.** We first consider the case of $\sum_{t=1}^{T}S_t(\boldsymbol{U}) \le \sum_{t=1}^{T}S_t(\boldsymbol{W}_t)$. From Lemma D.3, we have

$$\sum_{t=1}^{T}\mathbb{E}_t[L_t(\widehat{\boldsymbol{y}}_t)] - qT$$

$$\leq \sum_{t=1}^{T} v_t S_t(\boldsymbol{U}) + \sum_{t=1}^{T}((1-a)S_t(\boldsymbol{W}_t) - v_t S_t(\boldsymbol{W}_t)) + \sqrt{2}B\sqrt{\frac{bK}{q}\sum_{t=1}^{T} v_t S_t(\boldsymbol{W}_t)}$$

$$= \sum_{t=1}^{T} v_t \underbrace{(S_t(\boldsymbol{U}) - \xi S_t(\boldsymbol{W}_t))}_{=r_t} - (1-\xi)\sum_{t=1}^{T} v_t S_t(\boldsymbol{W}_t)$$

$$+ (1-a)\sum_{t=1}^{T} S_t(\boldsymbol{W}_t) + \sqrt{2}B\sqrt{\frac{bK}{q}\sum_{t=1}^{T} v_t S_t(\boldsymbol{W}_t)}$$

$$\leq \sum_{t=1}^{T} v_t r_t + (1-a)\sum_{t=1}^{T} S_t(\boldsymbol{W}_t) + \frac{B^2 K b}{2q(1-\xi)},$$

where the last inequality follows from $c_1\sqrt{x} - c_2 x \leq c_1^2/(4c_2)$ for $x \geq 0$, $c_1 \geq 0$, and $c_2 > 0$. From the concentration result provided in (5), this is further bounded as

$$\sum_{t=1}^{T} \mathbb{E}_t[L_t(\widehat{\boldsymbol{y}}_t)] - qT \leq \sum_{t=1}^{T}(S_t(\boldsymbol{U}) - \xi S_t(\boldsymbol{W}_t)) + \sqrt{3\log(1/\delta'')\sum_{t=1}^{T}\frac{KS_{\max}}{q}(S_t(\boldsymbol{U}) + S_t(\boldsymbol{W}_t))}$$

$$+ \frac{\sqrt{2}KS_{\max}}{3q}\log(1/\delta'') + (1-a)\sum_{t=1}^{T} S_t(\boldsymbol{W}_t) + \frac{B^2 K b}{2q(1-\xi)},$$

where we recall that $r_t = S_t(\boldsymbol{U}) - \xi S_t(\boldsymbol{W}_t)$. Rearranging the last inequality and using the inequality that $\sum_{t=1}^{T} S_t(\boldsymbol{U}) \leq \sum_{t=1}^{T} S_t(\boldsymbol{W}_t)$, we obtain

$$\sum_{t=1}^{T}(\mathbb{E}_t[L_t(\widehat{\boldsymbol{y}}_t)] - S_t(\boldsymbol{U})) \leq qT + \sqrt{6\log(1/\delta'')\sum_{t=1}^{T}\frac{KS_{\max}}{q}S_t(\boldsymbol{W}_t)} + \frac{\sqrt{2}KS_{\max}}{3q}\log(1/\delta'')$$

$$+ (1-a-\xi)\sum_{t=1}^{T} S_t(\boldsymbol{W}_t) + \frac{B^2 K b}{2q(1-\xi)}.$$

In what follows, we let $\delta' = \delta'' = \delta/2$ and $\xi = \frac{(4+c)\gamma}{\lambda\nu}$ for a sufficiently small constant $c > 0$, which satisfies $a + \xi > 1$. Then, plugging (4) and the last inequality in (3), with probability at least $1 - \delta$, we obtain

$$\mathcal{R}_T \leq \sqrt{2\log(2/\delta)\sum_{t=1}^{T}((1-a)S_t(\boldsymbol{W}_t) + q)} + \frac{\sqrt{2}}{3}\log(2/\delta) + qT$$

$$+ \sqrt{6\log(2/\delta)\sum_{t=1}^{T}\frac{KS_{\max}}{q}S_t(\boldsymbol{W}_t)} + \frac{\sqrt{2}KS_{\max}}{3q}\log(2/\delta)$$

$$+ (1-a-\xi)\sum_{t=1}^{T} S_t(\boldsymbol{W}_t) + \frac{B^2 K b}{2q(1-\xi)}$$

$$\leq \frac{1}{2(a+\xi-1)}\left((1-a) + \frac{3KS_{\max}}{q}\right)\log(2/\delta) + \sqrt{2qT\log(2/\delta)} + \frac{\sqrt{2}}{3}\log(2/\delta) + qT$$

$$+ \frac{\sqrt{2}KS_{\max}}{3q}\log(2/\delta) + \frac{B^2 b}{2q(1-\xi)}$$

$$\leq \frac{1}{q}\left(\frac{3KS_{\max}\log(2/\delta)}{2(a+\xi-1)} + KS_{\max}\log(2/\delta) + \frac{B^2 K b}{2(1-\xi)}\right) + qT + \sqrt{2qT\log(2/\delta)}$$

$$+ \frac{1}{2(a+\xi-1)}(1-a)\log(2/\delta) + \frac{\sqrt{2}}{3}\log(2/\delta)$$

$$= \frac{\mathcal{C}}{q} + qT + \sqrt{2qT\log(2/\delta)} + \frac{1}{2(a+\xi-1)}(1-a)\log(2/\delta) + \frac{\sqrt{2}}{3}\log(2/\delta).$$

Using the definition of $q = \sqrt{\mathcal{C}/T}$ with the last inequality, we obtain

$$\mathcal{R}_T \leq 2\sqrt{\mathcal{C}T} + (\mathcal{C}T)^{1/4}\sqrt{\log(2/\delta)} + \left(\frac{1-a}{2(a+\xi-1)} + \frac{\sqrt{2}}{3}\right)\log(2/\delta).$$

**When $\sum_{t=1}^{T} S_t(\boldsymbol{U}) > \sum_{t=1}^{T} S_t(\boldsymbol{W}_t)$.** We next consider the case of $\sum_{t=1}^{T} S_t(\boldsymbol{U}) > \sum_{t=1}^{T} S_t(\boldsymbol{W}_t)$.
We have

$$\mathcal{R}_T = \sum_{t=1}^{T}(L_t(\widehat{\boldsymbol{y}}_t) - \mathbb{E}_t[L_t(\widehat{\boldsymbol{y}}_t)]) + \sum_{t=1}^{T}(\mathbb{E}_t[L_t(\widehat{\boldsymbol{y}}_t)] - S_t(\boldsymbol{U}))$$

$$\leq \sqrt{2\log(1/\delta')\sum_{t=1}^{T}((1-a)S_t(\boldsymbol{W}_t)+q)} + \frac{\sqrt{2}}{3}\log(1/\delta') + \sum_{t=1}^{T}(\mathbb{E}_t[L_t(\widehat{\boldsymbol{y}}_t)] - S_t(\boldsymbol{W}_t))$$

$$\leq \sqrt{2\log(1/\delta')\sum_{t=1}^{T}((1-a)S_t(\boldsymbol{W}_t)+q)} + \frac{\sqrt{2}}{3}\log(1/\delta') + \sum_{t=1}^{T}(-aS_t(\boldsymbol{W}_t)+q)$$

$$\leq \frac{(1-a)\log(1/\delta')}{2a} + \sqrt{2qT\log(1/\delta')} + \frac{\sqrt{2}}{3}\log(1/\delta') + qT,$$

where the first inequality follows from (4) and $\sum_{t=1}^{T} S_t(\boldsymbol{U}) > \sum_{t=1}^{T} S_t(\boldsymbol{W}_t)$, and the second inequality follows from Assumption 3.2, the last inequality follows from $c_1\sqrt{x} - c_2 x \leq c_1^2/(4c_2)$ for $x \geq 0$, $c_1 \geq 0$, and $c_2 > 0$. Substituting $q = \sqrt{\mathcal{C}/T}$ and $\delta' = \delta/2$ in the last inequality, we obtain

$$\mathcal{R}_T \leq \frac{(1-a)\log(2/\delta)}{2a} + \sqrt{2\log(2/\delta)}(\mathcal{C}T)^{1/4} + \frac{\sqrt{2}}{3}\log(2/\delta) + \sqrt{\mathcal{C}T}.$$

This completes the proof. $\qquad\square$

### D.3  Proof of Theorem 3.6

Here, we provide the formal version and the proof of Theorem 3.6.

**Theorem D.4** (Formal version of Theorem 3.6). *The algorithm in Section 3.4 achieves*

$$\mathbb{E}[\mathcal{R}_T] \leq \frac{bB^2}{a} + 2^{5/3}\omega^{1/3}(BC_xT)^{2/3}.$$

We recall that $\boldsymbol{P}_t = \mathbb{E}_t[\widehat{\boldsymbol{y}}_t\widehat{\boldsymbol{y}}_t^\top]$. We then estimate $\boldsymbol{y}_t$ by $\widetilde{\boldsymbol{y}}_t = \boldsymbol{V}^{-1}\boldsymbol{P}_t^+\widehat{\boldsymbol{y}}_t\langle\widehat{\boldsymbol{y}}_t, \boldsymbol{V}\boldsymbol{y}_t\rangle$ and $\boldsymbol{G}_t$ by $\widetilde{\boldsymbol{G}}_t = (\widehat{\boldsymbol{y}}_\Omega(\boldsymbol{\theta}_t) - \widetilde{\boldsymbol{y}}_t)\boldsymbol{x}_t^\top$ under Assumption 3.5. This $\widetilde{\boldsymbol{G}}_t$ satisfies $\mathbb{E}_t[\widetilde{\boldsymbol{G}}_t] = \boldsymbol{G}_t$. To prove Theorem 3.6, we will upper bound $\mathbb{E}_t\left[\|\widetilde{\boldsymbol{G}}_t\|_\mathrm{F}^2\right]$. To do so, we begin by proving the following lemma:

**Lemma D.5.** *Let $\boldsymbol{A}$ and $\boldsymbol{B}$ positive semi-definite matrices with $\mathrm{Im}(\boldsymbol{A}) = \mathrm{Im}(\boldsymbol{B})$ and $\boldsymbol{A} \succeq \boldsymbol{B}$. Then, it holds that $\boldsymbol{A}^+ \preceq \boldsymbol{B}^+$.*

*Proof.* Since $\mathrm{Im}(\boldsymbol{A}) = \mathrm{Im}(\boldsymbol{B})$, there exists an orthogonal matrix $\boldsymbol{R}$, a diagonal matrix $\boldsymbol{\Lambda}$, and an invertible matrix $\boldsymbol{B}'$ that has same dimensions as $\boldsymbol{\Lambda}$ such that

$$\boldsymbol{A} = \boldsymbol{R}\begin{pmatrix} O & O \\ O & \boldsymbol{\Lambda} \end{pmatrix}\boldsymbol{R}^\top \quad \text{and} \quad \boldsymbol{B} = \boldsymbol{R}\begin{pmatrix} O & O \\ O & \boldsymbol{B}' \end{pmatrix}\boldsymbol{R}^\top.$$

Then, we have

$$\boldsymbol{A}^+ = \boldsymbol{R}\begin{pmatrix} O & O \\ O & \boldsymbol{\Lambda}^{-1} \end{pmatrix}\boldsymbol{R}^\top \quad \text{and} \quad \boldsymbol{B}^+ = \boldsymbol{R}\begin{pmatrix} O & O \\ O & \boldsymbol{B}'^{-1} \end{pmatrix}\boldsymbol{R}^\top. \tag{6}$$

From $\boldsymbol{A} \succeq \boldsymbol{B}$, we have $\boldsymbol{\Lambda} \succeq \boldsymbol{B}'$, which implies $\boldsymbol{\Lambda}^{-1} \preceq \boldsymbol{B}'^{-1}$. From this and (6), we have $\boldsymbol{A}^+ \preceq \boldsymbol{B}^+$, as desired. $\qquad\square$

Using this lemma, we prove a property of $\boldsymbol{P}_t$ and an upper bound of $\mathbb{E}_t\big[\mathrm{tr}(\widehat{\boldsymbol{y}}_t\widehat{\boldsymbol{y}}_t^\top)\big]$. In what follows, we use $\lambda_{\min}(\boldsymbol{A})$ to denote the minimum eigenvalue of a matrix $\boldsymbol{A}$.

**Lemma D.6.** *Suppose that* $\mathrm{tr}\big(\boldsymbol{V}^{-1}\boldsymbol{Q}(\boldsymbol{V}^{-1})^\top\big) \leq \omega$ *for* $\boldsymbol{Q} = \mathbb{E}_{\boldsymbol{y}\sim\mu}[\boldsymbol{y}\boldsymbol{y}^\top]$, *where we recall that $\mu$ is the uniform distribution over* $\mathcal{Y}$. *Then, we have*

$$\mathbb{E}_t\big[\mathrm{tr}(\widetilde{\boldsymbol{y}}_t\widetilde{\boldsymbol{y}}_t^\top)\big] \leq \frac{\omega}{q}.$$

*Proof.* By the linearity of expectation and the trace property, we have

$$\mathbb{E}_t\big[\mathrm{tr}(\widetilde{\boldsymbol{y}}_t\widetilde{\boldsymbol{y}}_t^\top)\big] \leq \mathrm{tr}\Big(\boldsymbol{V}^{-1}\boldsymbol{P}_t^+\mathbb{E}_t\big[\widehat{\boldsymbol{y}}_t\widehat{\boldsymbol{y}}_t^\top\big]\boldsymbol{P}_t^+\big(\boldsymbol{V}^{-1}\big)^\top\Big) = \mathrm{tr}\Big(\boldsymbol{V}^{-1}\boldsymbol{P}_t^+\boldsymbol{P}_t\boldsymbol{P}_t^+\big(\boldsymbol{V}^{-1}\big)^\top\Big)$$
$$= \mathrm{tr}\Big(\boldsymbol{V}^{-1}\boldsymbol{P}_t^+\big(\boldsymbol{V}^{-1}\big)^\top\Big),$$

where the first inequality follows from $-1 \leq \langle\widehat{\boldsymbol{y}}_t, \boldsymbol{V}\boldsymbol{y}_t\rangle \leq 1$ and the last equality follows from $\boldsymbol{P}_t^+\boldsymbol{P}_t\boldsymbol{P}_t^+ = \boldsymbol{P}_t^+$. The right-hand side is bounded as follows:

$$\mathrm{tr}\Big(\boldsymbol{V}^{-1}\boldsymbol{P}_t^+\big(\boldsymbol{V}^{-1}\big)^\top\Big) = \sum_{i=1}^d \mathbf{e}_i^\top\boldsymbol{V}^{-1}\boldsymbol{P}_t^+\big(\boldsymbol{V}^{-1}\big)^\top\mathbf{e}_i \leq \sum_{i=1}^d \mathbf{e}_i^\top\boldsymbol{V}^{-1}(q\boldsymbol{Q})^+\big(\boldsymbol{V}^{-1}\big)^\top\mathbf{e}_i$$
$$\leq \mathrm{tr}\Big(\big(\boldsymbol{V}^{-1}\big)^\top\boldsymbol{V}^{-1}(q\boldsymbol{Q})^+\Big) = \frac{1}{q}\mathrm{tr}\big(\boldsymbol{V}^{-1}\boldsymbol{Q}^+(\boldsymbol{V}^{-1})^\top\big) \leq \frac{\omega}{q},$$

where in the first inequality we used Lemma D.5 and in the last inequality we used the assumption that $\mathrm{tr}\big(\boldsymbol{V}^{-1}\boldsymbol{Q}^+(\boldsymbol{V}^{-1})^\top\big) \leq \omega$. This completes the proof. □

Now, we are ready to upper bound $\mathbb{E}_t\Big[\|\widetilde{\boldsymbol{G}}_t\|_{\mathrm{F}}^2\Big]$.

**Lemma D.7.** *Under the same assumption as Lemma D.6, it holds that*

$$\mathbb{E}_t\Big[\|\widetilde{\boldsymbol{G}}_t\|_{\mathrm{F}}^2\Big] \leq 2bS_t(\boldsymbol{W}_t) + \frac{2C_x^2\omega}{q}.$$

*Proof.* We have

$$\|\widetilde{\boldsymbol{G}}_t\|_{\mathrm{F}}^2 = \|(\widehat{\boldsymbol{y}}_\Omega(\boldsymbol{\theta}_t) - \widetilde{\boldsymbol{y}}_t)\boldsymbol{x}_t^\top\|_{\mathrm{F}}^2 \leq 2\|(\widehat{\boldsymbol{y}}_\Omega(\boldsymbol{\theta}_t) - \boldsymbol{y}_t)\boldsymbol{x}_t^\top\|_{\mathrm{F}}^2 + 2\|(\boldsymbol{y}_t - \widetilde{\boldsymbol{y}}_t)\boldsymbol{x}_t^\top\|_{\mathrm{F}}^2$$
$$\leq 2\|\boldsymbol{G}_t\|_{\mathrm{F}}^2 + 2C_x^2\|\boldsymbol{y}_t - \widetilde{\boldsymbol{y}}_t\|_2^2,$$

where we recall $C_x = \mathrm{diam}(\mathcal{X})$. From this inequality, we obtain

$$\mathbb{E}_t\Big[\|\widetilde{\boldsymbol{G}}_t\|_{\mathrm{F}}^2\Big] \leq 2\|\boldsymbol{G}_t\|_{\mathrm{F}}^2 + 2C_x^2\mathbb{E}_t\big[\|\boldsymbol{y}_t - \widetilde{\boldsymbol{y}}_t\|_2^2\big]$$
$$\leq 2bS_t(\boldsymbol{W}_t) + 2C_x^2\big(\|\boldsymbol{y}_t\|_2^2 - 2\boldsymbol{y}_t^\top\mathbb{E}_t[\widetilde{\boldsymbol{y}}_t] + \mathbb{E}_t\big[\|\widetilde{\boldsymbol{y}}_t\|_2^2\big]\big)$$
$$= 2bS_t(\boldsymbol{W}_t) + 2C_x^2\big(\|\boldsymbol{y}_t\|_2^2 - 2\|\boldsymbol{y}_t\|_2^2 + \mathbb{E}_t\big[\|\widetilde{\boldsymbol{y}}_t\|_2^2\big]\big)$$
$$\leq 2bS_t(\boldsymbol{W}_t) + 2C_x^2\mathbb{E}_t\big[\mathrm{tr}(\widetilde{\boldsymbol{y}}_t\widetilde{\boldsymbol{y}}_t^\top)\big] \leq 2bS_t(\boldsymbol{W}_t) + \frac{2C_x^2\omega}{q},$$

where in the second inequality we used $\|\boldsymbol{G}_t\|_{\mathrm{F}}^2 \leq bS_t(\boldsymbol{W}_t)$, in the equality we used $\mathbb{E}_t[\widetilde{\boldsymbol{y}}_t] = \boldsymbol{y}_t$, and in the last inequality we used Lemma D.6. □

Finally, we are ready to prove Theorem D.4.

*Proof of Theorem D.4.* From Assumption 3.2, we have

$$\mathbb{E}[\mathcal{R}_T] \leq \mathbb{E}\left[\sum_{t=1}^T (S_t(\boldsymbol{W}_t) - S_t(\boldsymbol{U}))\right] - a\mathbb{E}\left[\sum_{t=1}^T S_t(\boldsymbol{W}_t)\right] + qT$$
$$\leq \mathbb{E}\left[\sum_{t=1}^T \langle\boldsymbol{G}_t, \boldsymbol{W}_t - \boldsymbol{U}\rangle\right] - a\mathbb{E}\left[\sum_{t=1}^T S_t(\boldsymbol{W}_t)\right] + qT.$$

From Lemma D.7 and the unbiasedness of $\widetilde{\boldsymbol{G}}_t$, the first term in the last inequality is further bounded as

$$\mathbb{E}\left[\sum_{t=1}^{T}\langle\boldsymbol{G}_t, \boldsymbol{W}_t - \boldsymbol{U}\rangle\right] = \mathbb{E}\left[\sum_{t=1}^{T}\left\langle\widetilde{\boldsymbol{G}}_t, \boldsymbol{W}_t - \boldsymbol{U}\right\rangle\right] \le \sqrt{2}B\sqrt{\mathbb{E}\left[\sum_{t=1}^{T}\|\widetilde{\boldsymbol{G}}_t\|_{\mathrm{F}}^2\right]}$$

$$\le 2B\sqrt{b\mathbb{E}\left[\sum_{t=1}^{T}S_t(\boldsymbol{W}_t)\right] + 2BC_x\sqrt{\omega/q}},$$

where the first inequality follows from Lemma 3.3 and the last inequality follows from Lemma D.7 and the subadditivity of $x \mapsto \sqrt{x}$ for $x \ge 0$. Therefore, we obtain

$$\mathbb{E}[\mathcal{R}_T] \le 2B\sqrt{b\mathbb{E}\left[\sum_{t=1}^{T}S_t(\boldsymbol{W}_t)\right]} + 2BC_x\sqrt{\omega/q} - a\mathbb{E}\left[\sum_{t=1}^{T}S_t(\boldsymbol{W}_t)\right] + qT \qquad (7)$$

$$\le \frac{bB^2}{a} + 2BC_x\sqrt{\omega/q} + qT,$$

where we used $c_1\sqrt{x} - c_2 x \le c_1^2/(4c_2)$ for $x \ge 0$, $c_1 \ge 0$, and $c_2 > 0$. Finally, substituting $q = \left(\frac{4\omega B^2 C_x^2}{T}\right)^{1/3}$ in the last inequality gives

$$\mathbb{E}[\mathcal{R}_T] \le \frac{bB^2}{a} + 2^{5/3}\omega^{1/3}(BC_xT)^{2/3},$$

which is the desired bound. □

## D.4 Proof of Corollary 3.7

We derive the surrogate regret upper bounds provided by the algorithm established in Theorem D.4 for online multiclass classification, online multilabel classification, and ranking. Recall that we can achieve

$$\mathbb{E}[\mathcal{R}_T] \le \frac{bB^2}{a} + 2^{5/3}\omega^{1/3}(BC_xT)^{2/3}, \qquad (8)$$

where we recall that $\omega$ is defined as $\mathrm{tr}\left(\boldsymbol{V}^{-1}\boldsymbol{Q}^+(\boldsymbol{V}^{-1})^\top\right) \le \omega$ for $\boldsymbol{Q} = \mathbb{E}_{\boldsymbol{y}\sim\mu}[\boldsymbol{y}\boldsymbol{y}^\top]$. Note that when $\mathrm{span}(\mathcal{Y}) = \mathbb{R}^d$, then the matrix $\boldsymbol{Q}$ is invertible and $\lambda_{\min}(\boldsymbol{Q}) > 0$, and hence

$$\mathrm{tr}\left(\boldsymbol{V}^{-1}\boldsymbol{Q}^+(\boldsymbol{V}^{-1})^\top\right) = \sum_{i=1}^{d}\mathbf{e}_i^\top\boldsymbol{V}^{-1}\boldsymbol{Q}^+(\boldsymbol{V}^{-1})^\top\mathbf{e}_i \le \frac{1}{\lambda_{\min}(\boldsymbol{Q})}\sum_{i=1}^{d}\|(\boldsymbol{V}^{-1})^\top\mathbf{e}_i\|_2^2$$

$$\le \frac{1}{\lambda_{\min}(\boldsymbol{Q})}\|\boldsymbol{V}^{-1}\|_{\mathrm{F}}^2. \qquad (9)$$

Consequently, surrogate regret bounds for specific problems are obtained as follows:

**Multiclass classification with 0-1 loss** We first consider multiclass classification with the 0-1 loss. From $\boldsymbol{V} = \mathbf{1}\mathbf{1}^\top - \boldsymbol{I}$, we have $\|\boldsymbol{V}^{-1}\|_{\mathrm{F}}^2 \le d$ for $d \ge 2$. Recalling that $\mu$ is the uniform distribution over $\mathcal{Y} = \{\mathbf{e}_1, \ldots, \mathbf{e}_d\}$, we have $\mathbb{E}_{\boldsymbol{y}\sim\mu}[(\boldsymbol{y}^\top\boldsymbol{x})^2] = \frac{1}{d}\sum_{i=1}^{d}x_i^2$ for any $\boldsymbol{x} \in \mathbb{R}^d$. Hence, $\lambda_{\min}(\boldsymbol{Q}) = \min_{\|\boldsymbol{x}\|_2=1}\mathbb{E}_{\boldsymbol{y}\sim\mu}[(\boldsymbol{y}^\top\boldsymbol{x})^2] = \frac{1}{d}$, where the first equality follows from [9, Lemma 2]. Since $\mathrm{span}(\mathcal{Y}) = \mathbb{R}^d$ holds in this case, from (9), we can set $\omega = d/\lambda_{\min}(\boldsymbol{Q}) = d^2$. Substituting these into our surrogate regret upper bound in (8), we obtain

$$\mathbb{E}[\mathcal{R}_T] \le \frac{bB^2}{a} + 2^{5/3}(dBC_xT)^{2/3}.$$

**Online multilabel classification with $m$ correct labels and the Hamming loss** We next consider online multilabel classification with the number of correct labels $m$ and the Hamming loss. Since $\boldsymbol{V} = -\frac{2}{d}\boldsymbol{I}$, we have $\|\boldsymbol{V}^{-1}\|_{\mathrm{F}}^2 = \frac{d^3}{4}$. Let $\mathcal{Y} \subset \{0,1\}^d$ be the set of all vectors where exactly $m$

components are 1, and the remaining components are all 0. By drawing $\boldsymbol{y} \in \mathcal{Y}$ according to the uniform distribution over $\mathcal{Y}$, the probability that a given component of $\boldsymbol{y}$ is 1 is $\binom{m-1}{d-1}/\binom{m}{d} = \frac{m}{d}$. Hence, for any $\boldsymbol{x} \in \mathbb{R}^d$ with $\|\boldsymbol{x}\|_2 = 1$, we have

$$\mathbb{E}_{\boldsymbol{y} \sim \mu}\big[(\boldsymbol{y}^\top \boldsymbol{x})^2\big] = \frac{m}{d}\sum_{i=1}^d x_i^2 + \frac{m^2}{d^2}\sum_{i \neq j} x_i x_j = \left(\frac{m}{d}\sum_{i=1}^d x_i\right)^2 + \frac{m(d-m)}{d^2}\|\boldsymbol{x}\|_2^2 \geq \frac{m(d-m)}{d^2}.$$

Thus, $\lambda_{\min}(\boldsymbol{Q}) = \min_{\|\boldsymbol{x}\|_2=1} \mathbb{E}_{\boldsymbol{y} \sim \mu}\big[(\boldsymbol{y}^\top \boldsymbol{x})^2\big] \geq \frac{m(d-m)}{d^2}$ holds, where the equality is from Cesa-Bianchi and Lugosi [9, Lemma 2]. Since we have $\mathrm{span}(\mathcal{Y}) = \mathbb{R}^d$, from (9), we can set $\omega = \frac{d^5}{4m(d-m)} \geq \|\boldsymbol{V}^{-1}\|_\mathrm{F}^2/\lambda_{\min}(\boldsymbol{Q})$. Therefore, our surrogate regret upper bound in (8) is reduced to

$$\mathbb{E}[\mathcal{R}_T] \leq \frac{bB^2}{a} + 2\left(\frac{d^5}{m(d-m)}\right)^{1/3}(BC_xT)^{2/3}.$$

**Ranking with the Hamming loss and the number of items $m$**    We finally consider online ranking with the Hamming loss and the number of items $m$. From Cesa-Bianchi and Lugosi [9, Proposition 4], the smallest positive eigenvalue is at least $1/m$. Hence, since $\boldsymbol{V} = -\frac{1}{m}\boldsymbol{I}$, we have

$$\mathrm{tr}(\boldsymbol{V}^{-1}\boldsymbol{Q}^+(\boldsymbol{V}^{-1})^\top) = m^2\,\mathrm{tr}(\boldsymbol{Q}^+) \leq m^2 \sum_{i=1}^{\mathrm{rank}(\boldsymbol{Q}^+)} m \leq m^5,$$

where we used $\mathrm{rank}(\boldsymbol{Q}^+) \leq d = m^2$, and this allows us to choose $\omega = m^5$. Substituting these values into our surrogate regret upper bound in (8), we obtain

$$\mathbb{E}[\mathcal{R}_T] \leq \frac{bB^2}{a} + (2m)^{5/3}(BC_xT)^{2/3}.$$

# E    Details omitted from Section 4

This section provides the proofs of the theorems in Section 4.

## E.1    Details of Optimistic Delayed Adaptive FTRL (ODAFTRL)

We provide a more detailed explanation of the Optimistic Delayed Adaptive FTRL (ODAFTRL) algorithm used for updating $\boldsymbol{W}_t$ in Section 4.1. Recall that ODAFTRL computes $\boldsymbol{W}_t$ by the following update rule:

$$\boldsymbol{W}_{t+1} = \underset{\boldsymbol{W} \in \mathcal{W}}{\arg\min}\left\{\sum_{i=1}^{t-D}\langle \boldsymbol{G}_i, \boldsymbol{W}\rangle + \frac{\lambda_t\|\boldsymbol{W}\|_\mathrm{F}^2}{2}\right\}, \tag{10}$$

where $\lambda_t \geq 0$ is the regularization parameter. Note that we use the notation $a_{1:t} = \sum_{i=1}^t a_i$ for simplicity in the following. The ODAFTRL algorithm, when using the parameter update called AdaHedgeD, satisfies the following lemma:

**Lemma E.1** ([20, Theorem 12]). *Fix $\alpha > 0$. Let $f_t \colon \mathcal{W} \to \mathbb{R}$ be a convex function for each $t = 1, \ldots, T$. Suppose that we update $\lambda_t$ in (10) by the following AdaHedgeD update:*

$$\lambda_{t+1} = \frac{1}{\alpha}\sum_{s=1}^{t-D}\delta_s,$$

$$\delta_t = \min\{F_{t+1}(\boldsymbol{W}_t) - F_{t+1}(\bar{\boldsymbol{W}}_t), \langle \boldsymbol{G}_t, \boldsymbol{W}_t - \bar{\boldsymbol{W}}_t\rangle, F_{t+1}(\widehat{\boldsymbol{W}}_t) - F_{t+1}(\bar{\boldsymbol{W}}_t) + \langle \boldsymbol{G}_t, \boldsymbol{W}_t - \widehat{\boldsymbol{W}}_t\rangle\}_+,$$

$$\bar{\boldsymbol{W}}_t = \underset{\boldsymbol{W} \in \mathcal{W}}{\arg\min} F_{t+1}(\boldsymbol{W}),$$

$$\widehat{\boldsymbol{W}}_t = \underset{\boldsymbol{W} \in \mathcal{W}}{\arg\min}\left\{F_{t+1}(\boldsymbol{W}) - \min\left\{\frac{\|\boldsymbol{G}_t\|_\mathrm{F}}{\|\boldsymbol{G}_{t-D:t}\|_\mathrm{F}}, 1\right\}\langle \boldsymbol{G}_{t-D:t}, \boldsymbol{W}\rangle\right\}, \text{ and}$$

$$F_{t+1}(\boldsymbol{W}) = \frac{\lambda_t\|\boldsymbol{W}\|_\mathrm{F}^2}{2} + \langle \boldsymbol{G}_{1:t}, \boldsymbol{W}\rangle.$$

*Then, for any $\boldsymbol{U} \in \mathcal{W}$, ODAFTRL achieves*

$$\sum_{t=1}^{T} f_t(\boldsymbol{W}_t) - \sum_{t=1}^{T} f_t(\boldsymbol{U}) \leq \sum_{t=1}^{T} \langle \boldsymbol{G}_t, \boldsymbol{W}_t - \boldsymbol{U} \rangle \leq \left( \frac{B^2}{2\alpha} + 1 \right) \left( 2 \max_{s \in [T]} a_{s-D:s-1} + \sqrt{\sum_{t=1}^{T} a_t^2 + 2\alpha b_t} \right),$$

*where*

$$a_t = B \min\{\|\boldsymbol{G}_{t-D:t}\|_{\mathrm{F}}, \|\boldsymbol{G}_t\|_{\mathrm{F}}\},$$

$$b_t = \mathrm{huber}(\|\boldsymbol{G}_{t-D:t}\|_{\mathrm{F}}, \|\boldsymbol{G}_t\|_{\mathrm{F}}), \textit{ and } \mathrm{huber}(x, y) = \frac{1}{2}x^2 - \frac{1}{2}(|x| - |y|)_+^2 \leq \min\left\{ \frac{1}{2}x^2, |x||y| \right\}.$$

In the following, we let $\alpha = \frac{B^2}{2}$ for simplicity. Then, since $S_t$ is the convex function ,we have

$$\sum_{t=1}^{T} S_t(\boldsymbol{W}_t) - \sum_{t=1}^{T} S_t(\boldsymbol{U}) \leq \sum_{t=1}^{T} \langle \boldsymbol{G}_t, \boldsymbol{W}_t - \boldsymbol{U} \rangle \leq 2 \left( 2 \max_{s \in [T]} a_{s-D:s-1} + \sqrt{\sum_{t=1}^{T} a_t^2 + B^2 b_t} \right).$$

## E.2   Proof of Theorem 4.3

We present Theorem 4.3 in a more detailed form and provide its proof. In what follows, let

$$C_y = \max\left\{ \max_{\boldsymbol{y} \in \mathrm{conv}(\mathcal{Y})} \|\boldsymbol{y}\|_2, \max_{\boldsymbol{y}, \boldsymbol{y}' \in \mathrm{conv}(\mathcal{Y})} \|\boldsymbol{y} - \boldsymbol{y}'\|_2 \right\}$$

denote the maximum of the largest Euclidean norm of vectors in $\mathrm{conv}(\mathcal{Y})$ or the diameter of $\mathrm{conv}(\mathcal{Y})$.

**Theorem E.2** (Formal version of Theorem 4.3). *Let $\alpha = \frac{B^2}{2}$. Then, ODAFTRL with the AdaHedgeD update in online structured prediction with a fixed delay of $D$ achieves*

$$\mathbb{E}[\mathcal{R}_T] \leq 4BC_xC_yD + \frac{2bB^2}{a} + \min\left\{ \frac{b(D+1)^2}{2}, \frac{3}{2} \left( a^{-1} b B^4 C_x^2 C_y^2 (D+1)^2 T \right)^{1/3} \right\}.$$

*Proof.* From the definition of $b_t$, it holds that

$$\sum_{t=1}^{T} b_t \leq \sum_{t=1}^{T} \min\left\{ \frac{1}{2}\|\boldsymbol{G}_{t-D:t}\|_{\mathrm{F}}^2, \|\boldsymbol{G}_{t-D:t}\|_{\mathrm{F}}\|\boldsymbol{G}_t\|_{\mathrm{F}} \right\}$$

$$\leq \min\left\{ \frac{b(D+1)}{2} \sum_{t=1}^{T} \sum_{s=t-D}^{t} S_t(\boldsymbol{W}_t), C_xC_y(D+1) \sqrt{bT \sum_{t=1}^{T} S_t(\boldsymbol{W}_t)} \right\},$$

where we used $\|\sum_{i=1}^{n} \boldsymbol{A}_i\|_{\mathrm{F}}^2 \leq n \sum_{i=1}^{n} \|\boldsymbol{A}_i\|_{\mathrm{F}}^2$ for any matrix $\boldsymbol{A}_i$, the Cauchy–Schwarz inequality, $\|\boldsymbol{G}_t\|_{\mathrm{F}} \leq \|\widehat{\boldsymbol{y}}_{\Omega}(\boldsymbol{\theta}_t) - \boldsymbol{y}_t\|_2 \|\boldsymbol{x}_t\|_2 \leq C_xC_y$, and $\|\boldsymbol{G}_t\|_{\mathrm{F}}^2 \leq bS_t(\boldsymbol{W}_t)$. Combining this inequality with Lemma E.1 and the definition of $a_t$, we have

$$\sum_{t=1}^{T} (S_t(\boldsymbol{W}_t) - S_t(\boldsymbol{U}))$$

$$\leq 4B \max_{s \in [T]} \sum_{i=s-D}^{s-1} \|\boldsymbol{G}_i\|_{\mathrm{F}}$$

$$+ 2B \sqrt{ \sum_{t=1}^{T} \|\boldsymbol{G}_t\|_{\mathrm{F}}^2 + \min\left\{ \frac{b(D+1)}{2} \sum_{t=1}^{T} \sum_{s=t-D}^{t} S_t(\boldsymbol{W}_t), C_xC_y(D+1) \sqrt{bT \sum_{t=1}^{T} S_t(\boldsymbol{W}_t)} \right\} }$$

$$\leq 4BC_xC_yD + \sqrt{ bB^2 \sum_{t=1}^{T} S_t(\boldsymbol{W}_t) }$$

$$+ 2 \min\left\{ \sqrt{ \frac{b(D+1)^2}{2} \sum_{t=1}^{T} S_t(\boldsymbol{W}_t) }, \left( bB^4 C_x^2 C_y^2 (D+1)^2 T \sum_{t=1}^{T} S_t(\boldsymbol{W}_t) \right)^{1/4} \right\},$$

where we used $\|\boldsymbol{G}_t\|_{\mathrm{F}}^2 \leq bS_t(\boldsymbol{W}_t)$ and the subadditivity of $x \mapsto \sqrt{x}$ for $x \geq 0$ in the last inequality. From this inequality and Assumption 4.1, we can the evaluate surrogate regret as

$$
\mathbb{E}[\mathcal{R}_T] \leq \sum_{t=1}^{T} \left( (1-a)S_t(\boldsymbol{W}_t) - S_t(\boldsymbol{U}) \right)
$$

$$
\leq 4BC_xC_yD + \sqrt{bB^2 \sum_{t=1}^{T} S_t(\boldsymbol{W}_t)}
$$

$$
+ 2\min\left\{ \sqrt{\frac{b(D+1)^2}{2} \sum_{t=1}^{T} S_t(\boldsymbol{W}_t)}, \left( bB^4C_x^2C_y^2(D+1)^2 T \sum_{t=1}^{T} S_t(\boldsymbol{W}_t) \right)^{1/4} \right\} - a\sum_{t=1}^{T} S_t(\boldsymbol{W}_t)
$$

$$
\leq 4BC_xC_yD + \frac{2bB^2}{a} + \min\left\{ \frac{b(D+1)^2}{a}, \frac{3}{2}\left(a^{-1}bB^4C_x^2C_y^2(D+1)^2 T\right)^{1/3} \right\},
$$

where in the last inequality we used $c_1\sqrt{x} - c_2 x \leq c_1^2/(4c_2)$ and $c_1 x - c_2 x^4 \leq (3/4)\left(c_1^4/(4c_2)\right)^{1/3}$, which hold for any $x \geq 0$, $c_1 \geq 0$, and $c_2 > 0$. $\qquad\square$

### E.3 High-probability regret bound

We present the result of the high probability bound in a more detailed form and provide its proof.

**Theorem E.3.** *Let $\alpha = \frac{B^2}{2}$ and $\delta \in (0,1)$. Then, ODAFTRL with the AdaHedgeD update in online structured prediction with a fixed delay of D achieves*

$$
\mathcal{R}_T \leq 4BC_xC_yD + \frac{\sqrt{2}}{3} \log \frac{1}{\delta}
$$

$$
+ \frac{\left( \sqrt{(1-a)\log\frac{1}{\delta}} + \sqrt{2bB^2} \right)^2}{a} + \min\left\{ \frac{b(D+1)^2}{a}, \frac{3}{2}\left(a^{-1}bB^4C_x^2C_y^2(D+1)^2 T\right)^{1/3} \right\},
$$

*with probability at least $1 - \delta$.*

*Proof.* We decompose $\mathcal{R}_T$ into

$$
\mathcal{R}_T = \sum_{t=1}^{T} \left( L_t(\widehat{\boldsymbol{y}}_t) - \mathbb{E}_t[L_t(\widehat{\boldsymbol{y}}_t)] \right) + \sum_{t=1}^{T} \left( \mathbb{E}_t[L_t(\widehat{\boldsymbol{y}}_t)] - S_t(\boldsymbol{U}) \right). \tag{11}
$$

Let $Z_t = L_t(\widehat{\boldsymbol{y}}_t) - \mathbb{E}_t[L_t(\widehat{\boldsymbol{y}}_t)]$. Then, we have $|Z_t| \leq 1$ and $\mathbb{E}_t[Z_t^2] \leq \mathbb{E}_t[L_t(\widehat{\boldsymbol{y}}_t)] \leq (1-a)S_t(\boldsymbol{W}_t)$ from Assumption 4.1. Hence, from Lemma D.1, with probability at least $1 - \delta$, the first term in (11) is upper bounded as

$$
\sum_{t=1}^{T} Z_t \leq \sqrt{2(1-a)\sum_{t=1}^{T} S_t(\boldsymbol{W}_t) \log\frac{1}{\delta}} + \frac{\sqrt{2}}{3} \log\frac{1}{\delta}. \tag{12}
$$

From Assumption 4.1 and Lemma E.1, the second term in (11) is also upper bounded as

$$
\sum_{t=1}^{T} \left( \mathbb{E}_t[L_t(\widehat{\boldsymbol{y}}_t)] - S_t(\boldsymbol{U}) \right)
$$

$$
\leq 4BC_xC_yD + \sqrt{bB^2 \sum_{t=1}^{T} S_t(\boldsymbol{W}_t)}
$$

$$
+ 2\min\left\{ \sqrt{\frac{b(D+1)^2}{2} \sum_{t=1}^{T} S_t(\boldsymbol{W}_t)}, \left( bB^4C_x^2C_y^2(D+1)^2 T \sum_{t=1}^{T} S_t(\boldsymbol{W}_t) \right)^{1/4} \right\} - a\sum_{t=1}^{T} S_t(\boldsymbol{W}_t), \tag{13}
$$

---

**Algorithm 3** Black-box Online Learning under Delayed feedback (BOLD)

---

**Input:** BASE instances $\text{BASE}_0, \text{BASE}_1, \ldots, \text{BASE}_D$
1: **for** time step $t = 1, 2, \ldots, T$ **do**
2:     Set $r \leftarrow r_t = t - k(D+1)$, where $r_t \in \{0, \ldots, D\}$ and $k \in \mathbb{Z}_{\geq 0}$.
3:     Set $\boldsymbol{W}_t \leftarrow \boldsymbol{W}_r$ as the prediction for the current time step.
4:     Receive the delayed feedback.
5:     Update $\text{BASE}_r$ with the feedback.
6:     $\boldsymbol{W}_r \leftarrow$ the next prediction of $\text{BASE}_r$.

---

where we used the subadditivity of $x \mapsto \sqrt{x}$ for $x \geq 0$. Therefore, substituting (12) and (13) into (11) gives

$$
\mathcal{R}_T \leq 4BC_xC_yD + \frac{\sqrt{2}}{3}\log\frac{1}{\delta} + \left(\sqrt{2(1-a)\log\frac{1}{\delta}} + 2\sqrt{bB^2}\right)\sqrt{\sum_{t=1}^{T}S_t(\boldsymbol{W}_t)}
$$

$$
+ \min\left\{\sqrt{2b(D+1)^2\sum_{t=1}^{T}S_t(\boldsymbol{W}_t)}, 2\left(bB^4C_x^2C_y^2(D+1)^2T\sum_{t=1}^{T}S_t(\boldsymbol{W}_t)\right)^{1/4}\right\} - a\sum_{t=1}^{T}S_t(\boldsymbol{W}_t)
$$

$$
\leq 4BC_xC_yD + \frac{\sqrt{2}}{3}\log\frac{1}{\delta} + \frac{\left(\sqrt{(1-a)\log\frac{1}{\delta}} + \sqrt{2bB^2}\right)^2}{a}
$$

$$
+ \min\left\{\frac{b(D+1)^2}{a}, \frac{3}{2}\left(a^{-1}bB^4C_x^2C_y^2(D+1)^2T\right)^{1/3}\right\},
$$

where we used $c_1\sqrt{x} - c_2 x \leq c_1^2/(4c_2)$ and $c_1 x - c_2 x^4 \leq (3/4)\left(c_1^4/(4c_2)\right)^{1/3}$ for $x \geq 0$, $c_1 \geq 0$, and $c_2 > 0$ in the last inequality. This is the desired bound. $\qquad\square$

### E.4   Algorithm based on $(D+1)$-copies of online algorithms

Here, we provide the detail of Section 4.2. The pseudocode of BOLD for fixed delay $D$ is Algorithm 3. By using Algorithm 3, we can achive the following bound:

**Theorem E.4** (Formal version of Theorem 4.4)**.** *BOLD with adaptive OGD achieves the surrogate regret of*

$$
\mathbb{E}[\mathcal{R}_T] \leq \frac{bB^2(D+1)}{2a}.
$$

*Proof.* Let $T_j$ be the set of rounds $t$ for which the remainder when dividing $t$ by $D+1$ is equal to $j-1$, *i.e.*,$T_j = \{t \mid r_t = j-1\}$. By partitioning $T$ into these disjoint sets, we have

$$
\sum_{t=1}^{T}(S_t(\boldsymbol{W}_t) - S_t(\boldsymbol{U})) \leq \sum_{j=1}^{D+1}\left(\sum_{\tau \in T_j}(S_\tau(\boldsymbol{W}_\tau) - S_\tau(\boldsymbol{U}))\right).
$$

Applying OGD in Section 3.2 with the learning rate of $\eta_t = B/\sqrt{2\sum_{i=1}^{t}\|\boldsymbol{G}_t\|_{\mathrm{F}}^2}$ to each independent block, we obtain

$$
\sum_{\tau \in T_j}(S_\tau(\boldsymbol{W}_\tau) - S_\tau(\boldsymbol{U})) \leq \sqrt{2}B\sqrt{\sum_{\tau \in T_j}\|\boldsymbol{G}_\tau\|_{\mathrm{F}}^2}.
$$

Thus, it holds that

$$
\sum_{t=1}^{T}(S_t(\boldsymbol{W}_t) - S_t(\boldsymbol{U})) \leq \sqrt{2}B\sum_{j=1}^{D+1}\sqrt{\sum_{\tau \in T_j}\|\boldsymbol{G}_\tau\|_{\mathrm{F}}^2}
$$

$$
\leq \sqrt{2B^2(D+1)\sum_{t=1}^{T}\|\boldsymbol{G}_t\|_{\mathrm{F}}^2} \leq \sqrt{2bB^2(D+1)\sum_{t=1}^{T}S_t(\boldsymbol{W}_t)},
$$

where the second inequality follows from the Cauchy–Schwarz inequality and the last inequality follows from (1). Therefore, combining this and Assumption 4.1, we have

$$\mathbb{E}[\mathcal{R}_T] \leq \sum_{t=1}^{T}(S_t(\boldsymbol{W}_t) - S_t(\boldsymbol{U})) - a\sum_{t=1}^{T} S_t(\boldsymbol{W}_t)$$

$$\leq \sqrt{2bB^2(D+1)\sum_{t=1}^{T} S_t(\boldsymbol{W}_t)} - a\sum_{t=1}^{T} S_t(\boldsymbol{W}_t) \leq \frac{bB^2(D+1)}{2a},$$

where we used $c_1\sqrt{x} - c_2 x \leq c_1^2/(4c_2)$ for $x \geq 0$, $c_1 \geq 0$, and $c_2 > 0$ in the last inequality. $\qquad\square$

### E.5 Proof of Theorem 4.5

Here we provide the proof of Theorem 4.5.

*Proof.* Assume for simplicity that $M = (B^2 - \log^2(dT))/(\log(2d))^2$ is a positive integer. We partition the time indices $t = 1, \ldots, (D+1)M$ into $M$ blocks by grouping every consecutive $D+1$ rounds. In each block, we set an identical input vector and a true class. Specifically, we define the input vectors and the true classes as follows.

For each $s = 1, \ldots, M+1$, sample a true class $i_s \in [d]$ uniformly at random. Set $\boldsymbol{x}_t = \mathbf{e}_s$ for $t = (D+1)(s-1)+1, \ldots, (D+1)s$ for each $s = 1, \ldots, M$, and $\boldsymbol{x}_t = \mathbf{e}_{M+1}$ for $t > (D+1)M$. Define the offline estimator $\boldsymbol{U}' \in \mathbb{R}^{d \times (M+1)}$ such that its $s$-th column $(s = 1, \ldots, M)$ is $\log(2d)\mathbf{e}_{i_s}$, and its $(M+1)$-th column is $\log(dT)\mathbf{e}_{i_{M+1}}$. Note that $\|\boldsymbol{U}'\|_{\mathrm{F}}^2 = M(\log 2d)^2 + (\log(dT))^2 = B^2$ holds.

We denote $\widehat{\boldsymbol{y}}_s^i = \widehat{\boldsymbol{y}}_{(D+1)(s-1)+i}$, $\boldsymbol{y}_s^i = \boldsymbol{y}_{(D+1)(s-1)+i}$, and $S_s^i = S_{(D+1)(s-1)+i}$. Note that, within each block, the corresponding true class is not observed at the beginning of the $D+1$ rounds. Thus, for each $s = 1, \ldots, M+1$ and $i = 1, \ldots, D+1$, we have $\mathbb{E}[\mathbb{1}[\widehat{\boldsymbol{y}}_s^i \neq \boldsymbol{y}_s^i]] \geq 1 - \frac{1}{d}$. By the same calculation as that of Sakaue et al. [41, Theorem 13], we also have $S_s^i(\boldsymbol{U}') \leq \frac{1}{2}\left(1 - \frac{1}{d}\right)$. Thus, for each $i = 1, \ldots, D+1$, we have

$$\sum_{s=1}^{M} \mathbb{E}[\mathbb{1}[\widehat{\boldsymbol{y}}_s^i \neq \boldsymbol{y}_s^i]] - \sum_{s=1}^{M} S_s^i(\boldsymbol{U}') \geq \frac{M}{2}\left(1 - \frac{1}{d}\right) \geq \frac{M}{4} = \Omega\left(\frac{B^2}{(\log d)^2}\right).$$

Summing over $i = 1, \ldots, D+1$ yields

$$\sum_{t=1}^{(D+1)M} \mathbb{E}[\mathbb{1}[\widehat{\boldsymbol{y}}_t \neq \boldsymbol{y}_t]] - \sum_{t=1}^{(D+1)M} S_t(\boldsymbol{U}') = \Omega\left(\frac{B^2(D+1)}{(\log d)^2}\right).$$

The contribution of rounds $t > (D+1)M$ to the surrogate regret is non-negative. In fact, by the definition of $\boldsymbol{U}'$, we have $\sum_{t>(D+1)M} S_t(\boldsymbol{U}') \leq \frac{T-(D+1)M}{T}\left(1 - \frac{1}{d}\right) \leq 1 - \frac{1}{d}$, and we also have $\sum_{t>(D+1)M} \mathbb{E}[\mathbb{1}[\widehat{\boldsymbol{y}}_t \neq \boldsymbol{y}_t]] \geq 1 - \frac{1}{d}$ since $i_{M+1}$ is selected uniformly at random.

Therefore, it holds that

$$\mathbb{E}[\mathcal{R}_T] \geq \sum_{t=1}^{T} \mathbb{E}[\mathbb{1}[\widehat{\boldsymbol{y}}_t \neq \boldsymbol{y}_t]] - \sum_{t=1}^{T} S_t(\boldsymbol{U}') = \Omega\left(\frac{B^2(D+1)}{(\log d)^2}\right),$$

which completes the proof. $\qquad\square$

## F  Details omitted from Section 5

This section provides the omitted proofs of the theorems in Section 5.

### F.1 Common analysis

We provide the analysis that is commonly used in the proofs of Theorems 5.1 and 5.2. Although we use $\widehat{\boldsymbol{G}}_t$ as a notation for the estimator for convenience in this subsection, the same argument applies equally to $\widetilde{\boldsymbol{G}}_t$. We use ODAFTRL with the AdaHedgeD update in Appendix E.1 as ALG. Here, we recall that $\mathbb{E}[\sum_{t=1}^T (S_t(\boldsymbol{W}_t) - S_t(\boldsymbol{U}))] \leq \mathbb{E}[\sum_{t=1}^T \langle \widehat{\boldsymbol{G}}_t, \boldsymbol{W}_t - \boldsymbol{U} \rangle]$ from the convexity of $S_t$ and the unbiasness of $\widehat{\boldsymbol{G}}_t$. From Lemma E.1, it holds that

$$\sum_{t=1}^T \langle \widehat{\boldsymbol{G}}_t, \boldsymbol{W}_t - \boldsymbol{U} \rangle \leq 2 \left( 2 \max_{t \in [T]} a_{t-D:t-1} + \sqrt{\sum_{t=1}^T a_t^2 + B^2 b_t} \right), \tag{14}$$

where

$$a_t = B \min\{\|\widehat{\boldsymbol{G}}_{t-D:t}\|_{\mathrm{F}}, \|\widehat{\boldsymbol{G}}_t\|_{\mathrm{F}}\} \quad \text{and} \quad b_t \leq \min\left\{ \frac{1}{2}\|\widehat{\boldsymbol{G}}_{t-D:t}\|_{\mathrm{F}}^2, \|\widehat{\boldsymbol{G}}_{t-D:t}\|_{\mathrm{F}}\|\widehat{\boldsymbol{G}}_t\|_{\mathrm{F}} \right\}.$$

By the definition of $a_t$, we have

$$\mathbb{E}\left[ \max_{t \in [T]} a_{t-D:t-1} \right] \leq B\mathbb{E}\left[ \max_{t \in [T]} \sum_{s=t-D}^{t-1} \|\widehat{\boldsymbol{G}}_s\|_{\mathrm{F}} \right], \tag{15}$$

and thus

$$\mathbb{E}\left[ \sum_{t=1}^T \langle \widehat{\boldsymbol{G}}_t, \boldsymbol{W}_t - \boldsymbol{U} \rangle \right] \leq 2 \left( 2\mathbb{E}\left[ \max_{t \in [T]} a_{t-D:t-1} \right] + \sqrt{\mathbb{E}\left[ \sum_{t=1}^T a_t^2 \right]} + B\sqrt{\mathbb{E}\left[ \sum_{t=1}^T b_t \right]} \right)$$

$$\leq 2B \left( 2\mathbb{E}\left[ \max_{t \in [T]} \sum_{s=t-D}^{t-1} \|\widehat{\boldsymbol{G}}_s\|_{\mathrm{F}} \right] + \sqrt{\mathbb{E}\left[ \sum_{t=1}^T \|\widehat{\boldsymbol{G}}_t\|_{\mathrm{F}}^2 \right]} + \sqrt{\mathbb{E}\left[ \sum_{t=1}^T b_t \right]} \right), \tag{16}$$

where we used the subadditivity of $x \mapsto \sqrt{x}$ for $x \geq 0$. The last term in the last inequality is further bounded as

$$\mathbb{E}\left[ \sum_{t=1}^T b_t \right] \leq \mathbb{E}\left[ \sum_{t=1}^T \|\widehat{\boldsymbol{G}}_{t-D:t}\|_{\mathrm{F}}\|\widehat{\boldsymbol{G}}_t\|_{\mathrm{F}} \right] \leq \mathbb{E}\left[ \sum_{t=1}^T \|\widehat{\boldsymbol{G}}_t\|_{\mathrm{F}}^2 \right] + \mathbb{E}\left[ \sum_{t=1}^T \|\widehat{\boldsymbol{G}}_t\|_{\mathrm{F}} \sum_{s=t-D}^{t-1} \|\widehat{\boldsymbol{G}}_s\|_{\mathrm{F}} \right]$$

$$= \mathbb{E}\left[ \sum_{t=1}^T \|\widehat{\boldsymbol{G}}_t\|_{\mathrm{F}}^2 \right] + \mathbb{E}\left[ \sum_{t=1}^T \mathbb{E}_t\left[ \|\widehat{\boldsymbol{G}}_t\|_{\mathrm{F}} \right] \sum_{s=t-D}^{t-1} \|\widehat{\boldsymbol{G}}_s\|_{\mathrm{F}} \right], \tag{17}$$

where the second inequality follows from the triangle inequality and the equality follows from the law of total expectation.

### F.2 Proof of Theorem 5.1

We provide the complete version of Theorem 5.1:

**Theorem F.1** (Formal version of Theorem 5.1). *The algorithm in Section 5.1 achieves*

$$\mathbb{E}[\mathcal{R}_T] \leq 4BC_xC_yD + \left( \frac{4bB}{a} + 1 \right)\sqrt{KT} + 2BC_xC_y\sqrt{DT} = O(\sqrt{(K+D)T}).$$

*Proof.* First, we will upper bound $\mathbb{E}\left[ \sum_{t=1}^T b_t \right]$. From (17), we have

$$\mathbb{E}\left[ \sum_{t=1}^T b_t \right] \leq \mathbb{E}\left[ \sum_{t=1}^T \|\widehat{\boldsymbol{G}}_t\|_{\mathrm{F}}^2 \right] + \mathbb{E}\left[ \sum_{t=1}^T \mathbb{E}_t\left[ \|\widehat{\boldsymbol{G}}_t\|_{\mathrm{F}} \right] \sum_{s=t-D}^{t-1} \|\widehat{\boldsymbol{G}}_s\|_{\mathrm{F}} \right]$$

$$\leq \mathbb{E}\left[ \sum_{t=1}^T \|\widehat{\boldsymbol{G}}_t\|_{\mathrm{F}}^2 \right] + C_xC_y\mathbb{E}\left[ \sum_{t=1}^T \sum_{s=t-D}^{t-1} \mathbb{E}_s\left[ \|\widehat{\boldsymbol{G}}_s\|_{\mathrm{F}} \right] \right]$$

$$\leq \mathbb{E}\left[ \sum_{t=1}^T \|\widehat{\boldsymbol{G}}_t\|_{\mathrm{F}}^2 \right] + C_x^2C_y^2DT, \tag{18}$$

where the second and third inequality follow from the inequality $\mathbb{E}_t\left[\|\widehat{\boldsymbol{G}}_t\|_{\mathrm{F}}\right] = \|\boldsymbol{G}_t\|_{\mathrm{F}} \le C_x C_y$. Hence, from $\mathbb{E}_t\left[\|\widehat{\boldsymbol{G}}_t\|_{\mathrm{F}}\right] \le C_x C_y$, (16) and (18), it holds that

$$\mathbb{E}\left[\sum_{t=1}^{T}\langle\widehat{\boldsymbol{G}}_t, \boldsymbol{W}_t - \boldsymbol{U}\rangle\right] \le 2B\left(2C_x C_y D + \sqrt{\mathbb{E}\left[\sum_{t=1}^{T}\|\widehat{\boldsymbol{G}}_t\|_{\mathrm{F}}^2\right]} + \sqrt{\mathbb{E}\left[\sum_{t=1}^{T}\|\widehat{\boldsymbol{G}}_t\|_{\mathrm{F}}^2\right] + C_x^2 C_y^2 DT}\right)$$

$$\le 4BC_x C_y D + 4B\sqrt{\mathbb{E}\left[\sum_{t=1}^{T}\|\widehat{\boldsymbol{G}}_t\|_{\mathrm{F}}^2\right]} + 2BC_x C_y\sqrt{DT}$$

$$\le 4BC_x C_y D + 4B\sqrt{\frac{bK}{q}\mathbb{E}\left[\sum_{t=1}^{T}S_t(\boldsymbol{W}_t)\right]} + 2BC_x C_y\sqrt{DT}, \qquad (19)$$

where in the second inequality we used the subadditivity of $x \mapsto \sqrt{x}$ for $x \ge 0$ and in the last inequality we used

$$\mathbb{E}_t\left[\|\widehat{\boldsymbol{G}}_t\|_{\mathrm{F}}^2\right] = \frac{\|\boldsymbol{G}_t\|_{\mathrm{F}}^2}{p_t(\boldsymbol{y}_t)} \le \frac{K}{q}\|\boldsymbol{G}_t\|_{\mathrm{F}}^2 \le \frac{bK}{q}S_t(\boldsymbol{W}_t).$$

Therefore, combining all the above arguments yields

$$\mathbb{E}[\mathcal{R}_T] \le \mathbb{E}\left[\sum_{t=1}^{T}(S_t(\boldsymbol{W}_t) - S_t(\boldsymbol{U}))\right] - a\mathbb{E}\left[\sum_{t=1}^{T}S_t(\boldsymbol{W}_t)\right] + qT$$

$$\le \mathbb{E}\left[\sum_{t=1}^{T}\langle\widehat{\boldsymbol{G}}_t, \boldsymbol{W}_t - \boldsymbol{U}\rangle\right] - a\mathbb{E}\left[\sum_{t=1}^{T}S_t(\boldsymbol{W}_t)\right] + qT$$

$$\le 4BC_x C_y D + 4B\sqrt{\frac{bK}{q}\mathbb{E}\left[\sum_{t=1}^{T}S_t(\boldsymbol{W}_t)\right]} + 2BC_x C_y\sqrt{DT} - a\mathbb{E}\left[\sum_{t=1}^{T}S_t(\boldsymbol{W}_t)\right] + qT$$

$$\le 4BC_x C_y D + \frac{4bB^2}{a}\frac{bK}{q} + 2BC_x C_y\sqrt{DT} + qT,$$

where the first inequality follows from Assumption 3.2, the second inequality follows from the convexity of $S_t$ and the unbiasedness of $\widehat{\boldsymbol{G}}_t$, and the last inequality follows from $c_1\sqrt{x} - c_2 x \le c_1^2/(4c_2)$ for $x \ge 0$, $c_1 \ge 0$, and $c_2 > 0$. Finally, from $q = B\sqrt{K/T}$, we obtain

$$\mathbb{E}[\mathcal{R}_T] \le 4BC_x C_y D + \left(\frac{4bB}{a} + 1\right)\sqrt{KT} + 2BC_x C_y\sqrt{DT},$$

which is the desired bound. $\qquad\square$

### F.3 Proof of Theorem 5.2

We provide the complete version of Theorem 5.2:

**Theorem F.2** (Formal version of Theorem 5.2)**.** *The algorithm in Section 5.2 achieves*

$$\mathbb{E}[\mathcal{R}_T] = 4BC_x C_y(2D + \sqrt{DT}) + \frac{8bB^2}{a} + O\left(\omega^{1/3}D^{1/3}T^{2/3}\right).$$

*Proof.* First, we will derive an upper bound of $\mathbb{E}\left[\sum_{t=1}^{T}b_t\right]$. We first observe that

$$\mathbb{E}_t\left[\|\widetilde{\boldsymbol{G}}_t\|_{\mathrm{F}}\right] = \mathbb{E}_t\left[\|(\widehat{\boldsymbol{y}}_{\Omega}(\boldsymbol{\theta}_t) - \widetilde{\boldsymbol{y}}_t)\boldsymbol{x}_t^{\top}\|_{\mathrm{F}}\right]$$

$$\le \mathbb{E}_t[\|\boldsymbol{G}_t\|_{\mathrm{F}} + C_x\|\boldsymbol{y}_t - \widetilde{\boldsymbol{y}}_t\|_2] \le \|\boldsymbol{G}_t\|_{\mathrm{F}} + C_x\mathbb{E}_t[\|\boldsymbol{y}_t\|_2 + \|\widetilde{\boldsymbol{y}}_t\|_2]$$

$$\le \|\boldsymbol{G}_t\|_{\mathrm{F}} + C_x C_y + C_x\mathbb{E}_t\left[\sqrt{\mathrm{tr}(\widetilde{\boldsymbol{y}}_t\widetilde{\boldsymbol{y}}_t^{\top})}\right] \le \|\boldsymbol{G}_t\|_{\mathrm{F}} + C_x C_y + C_x\sqrt{\mathbb{E}_t\left[\mathrm{tr}(\widetilde{\boldsymbol{y}}_t\widetilde{\boldsymbol{y}}_t^{\top})\right]}$$

$$\le \|\boldsymbol{G}_t\|_{\mathrm{F}} + C_x C_y + \sqrt{C_x^2\omega/q} \le 2C_x C_y + \sqrt{C_x^2\omega/q}, \qquad (20)$$

where the first inequality follows from $C_x \geq \|\boldsymbol{x}_t\|_2$, the third inequality follows from $C_y \geq \|\boldsymbol{y}_t\|_2$, the fourth inequality follows from Jensen's inequality, and the fifth inequality follows from Lemma D.6. Thus, combining (17) with the last inequality, we have

$$
\begin{aligned}
\mathbb{E}\left[\sum_{t=1}^{T} b_t\right] &\leq \mathbb{E}\left[\sum_{t=1}^{T} \|\widetilde{\boldsymbol{G}}_t\|_{\mathrm{F}}^2\right] + \mathbb{E}\left[\sum_{t=1}^{T} \mathbb{E}_t\left[\|\widetilde{\boldsymbol{G}}_t\|_{\mathrm{F}}\right] \sum_{s=t-D}^{t-1} \|\widetilde{\boldsymbol{G}}_s\|_{\mathrm{F}}\right] \\
&\leq \mathbb{E}\left[\sum_{t=1}^{T} \|\widetilde{\boldsymbol{G}}_t\|_{\mathrm{F}}^2\right] + \left(2C_x C_y + \sqrt{\frac{C_x^2 \omega}{q}}\right) \mathbb{E}\left[\sum_{t=1}^{T} \sum_{s=t-D}^{t-1} \mathbb{E}_s\left[\|\widetilde{\boldsymbol{G}}_s\|_{\mathrm{F}}\right]\right] \\
&\leq \mathbb{E}\left[\sum_{t=1}^{T} \|\widetilde{\boldsymbol{G}}_t\|_{\mathrm{F}}^2\right] + DT\left(2C_x C_y + \sqrt{\frac{C_x^2 \omega}{q}}\right)^2.
\end{aligned}
\tag{21}
$$

Hence, from (16), (20), and (21), we have

$$
\begin{aligned}
&\mathbb{E}\left[\sum_{t=1}^{T} \langle \widetilde{\boldsymbol{G}}_t, \boldsymbol{W}_t - \boldsymbol{U} \rangle\right] \\
&\leq 2B\left(2\mathbb{E}\left[\max_{t \in [T]} \sum_{s=t-D}^{t-1} \|\widehat{\boldsymbol{G}}_s\|_{\mathrm{F}}\right] + \sqrt{\mathbb{E}\left[\sum_{t=1}^{T} \|\widehat{\boldsymbol{G}}_t\|_{\mathrm{F}}^2\right]} + \sqrt{\mathbb{E}\left[\sum_{t=1}^{T} b_t\right]}\right) \\
&\leq 2B\left(4C_x C_y D + 2C_x D\sqrt{\omega/q} + \sqrt{\mathbb{E}\left[\sum_{t=1}^{T} \|\widetilde{\boldsymbol{G}}_t\|_{\mathrm{F}}^2\right]} + \sqrt{\mathbb{E}\left[\sum_{t=1}^{T} b_t\right]}\right) \\
&\leq 8BC_x C_y D + 4BC_x D\sqrt{\omega/q} + 4B\sqrt{\mathbb{E}\left[\sum_{t=1}^{T} \|\widetilde{\boldsymbol{G}}_t\|_{\mathrm{F}}^2\right]} + 2B\left(2C_x C_y + C_x\sqrt{\omega/q}\right)\sqrt{DT} \\
&\leq 8BC_x C_y D + 4BC_x D\sqrt{\omega/q} \\
&\quad + 4B\sqrt{2\sum_{t=1}^{T}\left(bS_t(\boldsymbol{W}_t) + \frac{C_x^2 \omega}{q}\right)} + 2B\left(2C_x C_y + C_x\sqrt{\omega/q}\right)\sqrt{DT},
\end{aligned}
\tag{22}
$$

where the first inequality follows from (16), the second inequality follows from (20), the third inequality follows from (21) and the subadditivity of $x \mapsto \sqrt{x}$ for $x \geq 0$, and the last inequality follows from Lemma D.7. Therefore, combining all the above arguments yields

$$
\begin{aligned}
\mathbb{E}[\mathcal{R}_T] &\leq \mathbb{E}\left[\sum_{t=1}^{T}(S_t(\boldsymbol{W}_t) - S_t(\boldsymbol{U}))\right] - a\mathbb{E}\left[\sum_{t=1}^{T} S_t(\boldsymbol{W}_t)\right] + qT \\
&\leq \mathbb{E}\left[\sum_{t=1}^{T}\langle \widetilde{\boldsymbol{G}}_t, \boldsymbol{W}_t - \boldsymbol{U}\rangle\right] - a\mathbb{E}\left[\sum_{t=1}^{T} S_t(\boldsymbol{W}_t)\right] + qT \\
&\leq 8BC_x C_y D + 4B\left(\sqrt{2b\mathbb{E}\left[\sum_{t=1}^{T} S_t(\boldsymbol{W}_t)\right]} + C_x\sqrt{2\omega T/q}\right) + 4BC_x D\sqrt{\omega/q} \\
&\quad + 2B\left(2C_x C_y + C_x\sqrt{\omega/q}\right)\sqrt{DT} - a\mathbb{E}\left[\sum_{t=1}^{T} S_t(\boldsymbol{W}_t)\right] + qT \\
&\leq 4BC_x C_y(2D + \sqrt{DT}) + \frac{8bB^2}{a} + 4BC_x D\sqrt{\omega/q} + 2BC_x(\sqrt{D} + 2\sqrt{2})\sqrt{\omega T/q} + qT,
\end{aligned}
$$

where the first inequality follows from Assumption 3.2, the third inequality follows from (22) and the subadditivity of $x \mapsto \sqrt{x}$ for $x \geq 0$, and the last inequality follows from the definition of $\varepsilon$ and $c_1\sqrt{x} - c_2 x \leq c_1^2/(4c_2)$ for $x \geq 0$, $c_1 \geq 0$, and $c_2 > 0$. Finally, substituting $q = \left(\frac{\omega B^2 C_x^2 D}{T}\right)^{1/3}$ gives the desired bound. $\qquad\square$

---
**Algorithm 4** Single-instance Online Learning In Delayed environments (SOLID)
---
**Input:** BASE, the first prediction $\boldsymbol{W}$ of BASE
 1: **for** time step $t = 1, 2, \ldots, T$ **do**
 2:     Set $\boldsymbol{W}_t \leftarrow \boldsymbol{W}$ as the prediction for the current time step.
 3:     Receive the feedbacks $H_t$ that arrive at the end of time step $t$.
 4:     **for all** feedback in $H_t$ **do**
 5:         Update BASE with feedback.
 6:         $\boldsymbol{W} \leftarrow$ the next prediction of BASE.
---

## G   Variable delay

This section provides the algorithms and analyses under the variable-delay setting, which is a natural extension of the fixed-delay setting. As the notation for the variable-delay setting, let $\tau_t$ denote the delay time of the feedback received at time $t$, and define $\tau_* = \max_t \tau_t$. Under this setting, by leveraging the Single-instance Online Learning In Delayed environments (SOLID) [28], we achieve a surrogate regret bound of $O(\sqrt{\tau_{1:T}} + \tau_*)$ in the full-information setting (Theorem G.2), and bounds of $O(\sqrt{KT} + \sqrt{\tau_{1:T}} + T^{2/3} + \tau_*)$ (Theorem G.3) and $O(T^{1/6}\sqrt{\tau_{1:T}} + \tau_*)$ (Theorem G.4) in the bandit setting.

### G.1   Single-instance Online Learning In Delayed environments (SOLID)

We provide a detail of SOLID algorithm used for updating $\boldsymbol{W}_t$ under the variable-delay setting. Consider any deterministic non-delayed online learning algorithm (call it BASE). SOLID is an algorithm that, regardless of the original arrival time of the feedback, provides the feedback to BASE in the order in which it is observed, and makes predictions based on the outputs of BASE (Algorithm 4). Below, let $\rho(t)$ denote the time step of the $t$th feedback from SOLID to BASE for any $t \in [T]$, as in [28]. When we use OGD as BASE, SOLID achieves the following bound:

**Lemma G.1** ([28, Theorem 5]). *Let BASE OGD with learning rate*

$$\widetilde{\eta}_t = \sqrt{2}R\left(\sqrt{\sum_{s=1}^{t}(\|\boldsymbol{G}_{\rho(s)}\|_{\mathrm{F}}^2 + 2\|\boldsymbol{G}_{\rho(s)}\|_{\mathrm{F}} \sum_{i=t-\widetilde{\tau}_t}^{t-1}\|\boldsymbol{G}_{\rho(i)}\|_{\mathrm{F}}) + C_x^2 C_y^2(\tau_*^2 + \tau_*)}\right)^{-1},$$

*where $R > 0$ satisfies $\widetilde{\eta}_T \sum_{t=1}^{T}\|\boldsymbol{U} - \boldsymbol{W}_t\|_{\mathrm{F}}^2 \leq 4R^2$. Then, SOLID achieves*

$$\sum_{t=1}^{T}(S_t(\boldsymbol{W}_t) - S_t(\boldsymbol{U}))$$

$$\leq 2\sqrt{2}R\sqrt{\sum_{t=1}^{T}\|\boldsymbol{G}_t\|_{\mathrm{F}}^2 + 2\sum_{t=1}^{T}\|\boldsymbol{G}_{\rho(t)}\|_{\mathrm{F}} \sum_{s=t+1}^{T}\|\boldsymbol{G}_{\rho(s)}\|_{\mathrm{F}}\mathbb{1}\{s - \widetilde{\tau}_s \leq t\}} + C_x C_y R\sqrt{2(\tau_*^2 + \tau_*)}.$$

### G.2   Full-information

Here, we provide the algorithm for the variable-delay full-information setting. This subsection assumes Assumption 4.1, as in Section 4.

**Algorithm**   We use SOLID with OGD as ALG for updating $\boldsymbol{W}_t$.

**Regret bound and analysis**   The above algorithm achieves the following bound:

**Theorem G.2.** *SOLID with OGD update in online structured prediction with a delay of $\tau_t$ achieves*

$$\mathbb{E}[\mathcal{R}_T] \leq \frac{2bR^2}{a} + 4C_x C_y R\sqrt{\tau_{1:T}} + C_x C_y R\sqrt{2(\tau_*^2 + \tau_*)} = O(\sqrt{\tau_{1:T}} + \tau_*).$$

*Proof.* Using SOLID with OGD (Lemma G.1), we have

$$\sum_{t=1}^{T} (S_t(\boldsymbol{W}_t) - S_t(\boldsymbol{U}))$$

$$\leq 2\sqrt{2}R\sqrt{\sum_{t=1}^{T}\|\boldsymbol{G}_t\|_{\mathrm{F}}^2 + 2\sum_{t=1}^{T}\|\boldsymbol{G}_{\rho(t)}\|_{\mathrm{F}}\sum_{s=t+1}^{T}\|\boldsymbol{G}_{\rho(s)}\|_{\mathrm{F}}\mathbb{1}\{s - \widetilde{\tau}_s \leq t\}} + C_x C_y R\sqrt{2(\tau_*^2 + \tau_*)}$$

$$\leq 2\sqrt{2}R\sqrt{b\sum_{t=1}^{T}S_t(\boldsymbol{W}_t) + 4C_x C_y R\sqrt{\tau_{1:T}}} + C_x C_y R\sqrt{2(\tau_*^2 + \tau_*)}, \tag{23}$$

where we used $\|\boldsymbol{G}_t\|_{\mathrm{F}}^2 \leq bS_t(\boldsymbol{W}_t)$, $\|\boldsymbol{G}_t\|_{\mathrm{F}} \leq C_x C_y$, $\sum_{s=t+1}^{T}\mathbb{1}\{s - \widetilde{\tau}_s \leq t\} = \widetilde{\tau}_s$, $\sum_{t=1}^{T}\widetilde{\tau}_t = \sum_{t=1}^{T}\tau_t$, and the subadditivity of $x \mapsto \sqrt{x}$ for $x \geq 0$ in the last inequality. Therefore, from this inequality, it holds that

$$\mathbb{E}[\mathcal{R}_T] \leq \sum_{t=1}^{T}(S_t(\boldsymbol{W}_t) - S_t(\boldsymbol{U})) - a\sum_{t=1}^{T}S_t(\boldsymbol{W}_t)$$

$$\leq 2\sqrt{2}R\sqrt{b\sum_{t=1}^{T}S_t(\boldsymbol{W}_t) + 4C_x C_y R\sqrt{\tau_{1:T}} + C_x C_y R\sqrt{2(\tau_*^2 + \tau_*)}} - a\sum_{t=1}^{T}S_t(\boldsymbol{W}_t)$$

$$\leq \frac{2bR^2}{a} + 4C_x C_y R\sqrt{\tau_{1:T}} + C_x C_y R\sqrt{2(\tau_*^2 + \tau_*)},$$

where the first inequality follows from Assumption 4.1, the second inequality follows from (23), and the last inequality follows from $c_1\sqrt{x} - c_2 x \leq c_1^2/(4c_2)$ for $x \geq 0$, $c_1 \geq 0$, and $c_2 > 0$. This is the desired bound. $\qquad\square$

This result is superior to the algorithm designed for the fixed-delay feedback in that it can handle variable-delay feedback. When $D \geq \tau_*$ is known and small, we may also use the algorithm developed for fixed-delay feedback to achieve the $O(D^2)$ bound.

## G.3   Bandit feedback

We provide algorithms for the variable-delay bandit setting. This subsection assumes Assumption 3.2, as in Section 3. Then, by using the inverse-weighted gradient estimator and the pseudo-inverse matrix estimator as gradient estimators, we can achieve surrogate regret upper bounds of $O(\sqrt{KT} + \sqrt{\tau_{1:T}} + \tau_*)$ and $O(T^{1/6}\sqrt{\tau_{1:T}} + T^{2/3} + \tau_*)$, respectively. Below, we provide details of these results.

### G.3.1   Algorithm based on inverse-weighted gradient estimator with $O(\sqrt{KT} + \sqrt{\tau_{1:T}} + \tau_*)$ regret

Here, we introduce an algorithm with a surrogate regret upper bound of $O(\sqrt{KT} + \sqrt{\tau_{1:T}} + \tau_*)$.

**Algorithm**   We use RDUE with $q = R\sqrt{K/T}$ for decoding (assuming $T \geq R^2 K$), the gradient estimator $\widehat{\boldsymbol{G}}_t$ as in Section 3.3, and SOLID with OGD as ALG.

**Regret bound and analysis**   The above algorithm achieves the following surrogate regret bound:

**Theorem G.3.** *The above algorithm achieves*

$$\mathbb{E}[\mathcal{R}_T] \leq \left(\frac{2b}{a} + 1\right)R\sqrt{KT} + 4C_x C_y R\sqrt{\tau_{1:T}} + C_x C_y R\sqrt{2(\tau_*^2 + \tau_*)} = O(\sqrt{KT} + \sqrt{\tau_{1:T}} + \tau_*).$$

This result is an extension of the bound under the fixed-delay setting. In particular, if $\tau_t = D$ for any $t$, we obtain $\mathbb{E}[\mathcal{R}_T] = O(\sqrt{(K + D)T})$.

*Proof.* First, we recall that $\mathbb{E}_t[\|\boldsymbol{G}_t\|_{\mathrm{F}}] \leq C_x C_y$ and $\mathbb{E}_t[\|\boldsymbol{G}_t\|_{\mathrm{F}}^2] \leq bKS_t(\boldsymbol{W}_t)/q$ from the proof of Theorem 5.1. Using SOLID with OGD (Lemma G.1) and the subadditivity of $x \mapsto \sqrt{x}$ for $x \geq 0$, we have

$$\mathbb{E}\left[\sum_{t=1}^{T}(S_t(\boldsymbol{W}_t) - S_t(\boldsymbol{U}))\right]$$

$$\leq 2\sqrt{2}R\mathbb{E}\left[\sqrt{\sum_{t=1}^{T}\|\widehat{\boldsymbol{G}}_t\|_{\mathrm{F}}^2 + 2\sum_{t=1}^{T}\sum_{s=t+1}^{T}\|\widehat{\boldsymbol{G}}_{\rho(t)}\|_{\mathrm{F}}\|\widehat{\boldsymbol{G}}_{\rho(s)}\|_{\mathrm{F}}\mathbb{1}\{s - \widehat{\tau}_s \leq t\}}\right]$$

$$+ C_x C_y R\sqrt{2(\tau_*^2 + \tau_*)}$$

$$\leq 2\sqrt{2}R\mathbb{E}\left[\sqrt{\sum_{t=1}^{T}\|\widehat{\boldsymbol{G}}_t\|_{\mathrm{F}}^2}\right] + 4R\mathbb{E}\left[\sqrt{\sum_{t=1}^{T}\sum_{s=t+1}^{T}\|\widehat{\boldsymbol{G}}_{\rho(t)}\|_{\mathrm{F}}\|\widehat{\boldsymbol{G}}_{\rho(s)}\|_{\mathrm{F}}\mathbb{1}\{s - \widehat{\tau}_s \leq t\}}\right]$$

$$+ C_x C_y R\sqrt{2(\tau_*^2 + \tau_*)}. \tag{24}$$

The second term is bounded as

$$\mathbb{E}\left[\sum_{t=1}^{T}\sum_{s=t+1}^{T}\|\widehat{\boldsymbol{G}}_{\rho(t)}\|_{\mathrm{F}}\|\widehat{\boldsymbol{G}}_{\rho(s)}\|_{\mathrm{F}}\mathbb{1}\{s - \widehat{\tau}_s \leq t\}\right]$$

$$\leq \mathbb{E}\left[\sum_{t=1}^{T}\sum_{s=t+1}^{T}\mathbb{E}_{\rho_{\max}}\left[\|\widehat{\boldsymbol{G}}_{\rho_{\max}}\|_{\mathrm{F}}\right]\|\widehat{\boldsymbol{G}}_{\rho_{\min}}\|_{\mathrm{F}}\mathbb{1}\{s - \widehat{\tau}_s \leq t\}\right]$$

$$\leq C_x C_y \mathbb{E}\left[\sum_{t=1}^{T}\sum_{s=t+1}^{T}\mathbb{E}_{\rho_{\min}}\left[\|\widehat{\boldsymbol{G}}_{\rho_{\min}}\|\right]_{\mathrm{F}}\mathbb{1}\{s - \widetilde{\tau}_s \leq t\}\right] \leq C_x^2 C_y^2 \sum_{t=1}^{T}\tau_t, \tag{25}$$

where we assumed $\rho_{\max} = \max\{\rho(t), \rho(s)\}$ and $\rho_{\min} = \min\{\rho(t), \rho(s)\}$, used the tower property in the first and second inequalities, and used $\mathbb{E}_t[\|\boldsymbol{G}_t\|_{\mathrm{F}}] \leq C_x C_y$ in the second and last inequalities. Hence, it holds that

$$\mathbb{E}\left[\sum_{t=1}^{T}(S_t(\boldsymbol{W}_t) - S_t(\boldsymbol{U}))\right] \leq 2\sqrt{2}R\sqrt{\frac{bK}{q}\mathbb{E}\left[\sum_{t=1}^{T}S_t(\boldsymbol{W}_t)\right]} + 4C_x C_y R\sqrt{\tau_{1:T}} + C_x C_y R\sqrt{2(\tau_*^2 + \tau_*)}, \tag{26}$$

where the inequality follows from $\mathbb{E}_t[\|\boldsymbol{G}_t\|_{\mathrm{F}}^2] \leq bKS_t(\boldsymbol{W}_t)/q$ and (25). Therefore, combining all the above arguments yields

$$\mathbb{E}[\mathcal{R}_T] \leq \mathbb{E}\left[\sum_{t=1}^{T}(S_t(\boldsymbol{W}_t) - S_t(\boldsymbol{U}))\right] - a\mathbb{E}\left[\sum_{t=1}^{T}S_t(\boldsymbol{W}_t)\right] + qT$$

$$\leq 2\sqrt{2}R\sqrt{\frac{bK}{q}\mathbb{E}\left[\sum_{t=1}^{T}S_t(\boldsymbol{W}_t)\right]} + 4C_x C_y R\sqrt{\tau_{1:T}}$$

$$+ C_x C_y R\sqrt{2(\tau_*^2 + \tau_*)} - a\mathbb{E}\left[\sum_{t=1}^{T}S_t(\boldsymbol{W}_t)\right] + qT$$

$$\leq \frac{2bR^2K}{aq} + 4C_x C_y R\sqrt{\tau_{1:T}} + C_x C_y R\sqrt{2(\tau_*^2 + \tau_*)} + qT,$$

where the first inequality follows from Assumption 3.2, the second inequality follows from (26), and the last inequality follows from $c_1\sqrt{x} - c_2 x \leq c_1^2/(4c_2)$ for $x \geq 0$, $c_1 \geq 0$, and $c_2 > 0$. Finally, choosing $q = R\sqrt{K/T}$ gives the desired bound. $\qquad\square$

### G.3.2 Algorithm based on pseudo-inverse matrix estimator with $O(T^{1/6}\sqrt{\tau_{1:T}} + T^{2/3} + \tau_*)$ regret

Here, we provide an algorithm that achieves a surrogate regret upper bound of $O(T^{1/6}\sqrt{\tau_{1:T}} + T^{2/3} + \tau_*)$. This subsection assumes Assumption 3.5 in addition to Assumption 3.2.

**Algorithm** We use RDUE with $q = \left(\frac{\omega R^2 C_x^2}{T}\right)^{1/3}$ for decoding (assuming $T \geq \omega R^2 C_x^2$), the gradient estimator $\widetilde{\boldsymbol{G}}_t$ as in Section 3.4, and SOLID with OGD as ALG.

**Regret bound and analysis** The algorithm described above achieves the following surrogate regret bound:

**Theorem G.4.** *The above algorithm achieves*

$$
\mathbb{E}[\mathcal{R}_T] \leq \frac{4bR^2K}{a} + 8C_xC_yR\sqrt{\tau_{1:T}} + C_xC_yR\sqrt{2(\tau_*^2 + \tau_*)} + O\left(\omega^{1/3}R^{2/3}T^{1/6}\sqrt{\tau_{1:T}} + T^{2/3}\right)
$$
$$
= O(T^{1/6}\sqrt{\tau_{1:T}} + T^{2/3} + \tau_*).
$$

*Proof.* First, we recall that $\mathbb{E}_t[\|\widetilde{\boldsymbol{G}}_t\|_{\mathrm{F}}] \leq 2C_xC_y + \sqrt{C_x^2\omega/q}$ and $\mathbb{E}_t[\|\widetilde{\boldsymbol{G}}_t\|_{\mathrm{F}}^2] \leq 2bKS_t(\boldsymbol{W}_t) + \frac{2C_x^2\omega}{q}$ hold from (20) and Lemma D.7, respectively. Following the same steps as in the proof of Theorem G.3, we obtain

$$
\mathbb{E}\left[\sum_{t=1}^T \sum_{s=t+1}^T \|\widetilde{\boldsymbol{G}}_{\rho(t)}\|_{\mathrm{F}}\|\widetilde{\boldsymbol{G}}_{\rho(s)}\|_{\mathrm{F}}\mathbb{1}\{s - \widetilde{\tau}_s \leq t\}\right] \leq \left(2C_xC_y + \sqrt{C_x^2\omega/q}\right)^2 \sum_{t=1}^T \tau_t. \quad (27)
$$

Therefore, by using these inequalities and (24), we get

$$
\mathbb{E}[\mathcal{R}_T] \leq \mathbb{E}\left[\sum_{t=1}^T (S_t(\boldsymbol{W}_t) - S_t(\boldsymbol{U}))\right] - a\mathbb{E}\left[\sum_{t=1}^T S_t(\boldsymbol{W}_t)\right] + qT
$$
$$
\leq 4R\mathbb{E}\left[\sqrt{bK\mathbb{E}\left[\sum_{t=1}^T S_t(\boldsymbol{W}_t)\right] + \frac{C_x^2\omega}{q}}\right] + 4R\left(2C_xC_y + \sqrt{\frac{C_x^2\omega}{q}}\right)\sqrt{\tau_{1:T}}
$$
$$
+ C_xC_yR\sqrt{2(\tau_*^2 + \tau_*)} - a\mathbb{E}\left[\sum_{t=1}^T S_t(\boldsymbol{W}_t)\right] + qT
$$
$$
\leq \frac{4bR^2K}{a} + 4R\sqrt{\frac{C_x^2\omega}{q}} + 4R\left(2C_xC_y + \sqrt{\frac{C_x^2\omega}{q}}\right)\sqrt{\tau_{1:T}} + C_xC_yR\sqrt{2(\tau_*^2 + \tau_*)} + qT,
$$

where the first inequality follows from Assumption 3.2, the second inequality follows from Lemma D.7, (24), and (27), the last inequality follows from $c_1\sqrt{x} - c_2 x \leq c_1^2/(4c_2)$ for $x \geq 0$, $c_1 \geq 0$, and $c_2 > 0$. Finally, by substituting $q = \left(\frac{\omega R^2 C_x^2}{T}\right)^{1/3}$, we can obtain the desired bound. $\quad\square$

# H   Numerical experiments

This section presents the results of numerical experiments for online multiclass classification and multilabel classification under bandit feedback on MNIST and synthetic data. All experiments were run on a system with 16GB of RAM, Apple M3 CPU, and in Python 3.11.7 on a macOS Sonoma 14.6.1. The code is provided in the supplementary material.

## H.1   Multiclass classification

**Setup** We describe the experimental setup. We compare four algorithms: Gaptron [44] with logistic loss, Gappletron [45] with logistic loss and hinge loss, and our algorithm in Section 5.1. Theoretically, these methods have their own advantages: ours enjoys a surrogate regret bound of $O(\sqrt{KT})$, which is better than the $O(K\sqrt{T})$ bounds of the others; however, Gaptron/Gappletron can work with a broader class of surrogate losses. This section aims to compare those methods from the empirical perspective.

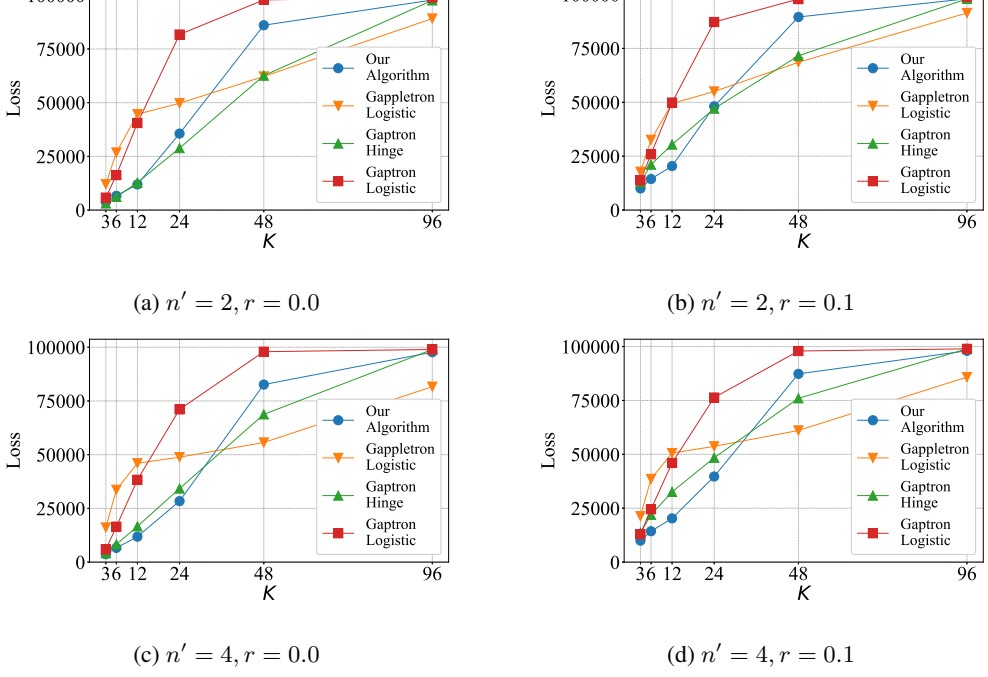

(a) $n' = 2, r = 0.0$

(b) $n' = 2, r = 0.1$

(c) $n' = 4, r = 0.0$

(d) $n' = 4, r = 0.1$

Figure 1: Results of the synthetic experiments in multiclass classification with bandit feedback. In all figures, the horizontal axis represents the number of classes $K$, and the vertical axis represents the cumulative target loss.

**Details of algorithms**  As the algorithm ALG for updating the linear estimator, we employ the OGD in Section 3.2. Following [45], we use the learning rate of $\eta_t = B/\sqrt{2(10^{-8} + \sum_{i=1}^{t}\|\widetilde{\boldsymbol{G}}_t\|_F^2)}$ and no projection is performed in OGD. Here, the addition of $10^{-8}$ to the denominator is to prevent division by zero. Although $B = \mathrm{diam}(\mathcal{W})$ is unknown, we fixed $B = 10$ in all experiments, regardless of whether this value represents the actual diameter. All other parameters are set according to theoretical values. Under these parameter settings, we repeat experiments 20 times.

### H.1.1  Synthetic data

We also run experiments on synthetic data to facilitate comparisons across different values of $K$.

**Data generation**  We describe the procedure for generating synthetic data. The synthetic data were generated by using the same procedure as Van der Hoeven et al. [45]. The input vector consists of a binary vector with entries of 0 and 1, and is composed of two parts. The first part corresponds to a unique feature vector associated with the label, and the second part is randomly selected and unrelated to the label. Specifically, the data is generated as follows. We generate $K \in \mathbb{N}$ unique feature vectors of length $10n'$ as follows. First, we randomly select an integer $s$ uniformly from the range $[n', 5n']$, then randomly choose $s$ elements from a zero vector of length $10n'$ and set them to 1. The input vector is obtained by concatenating the feature vector of a randomly chosen class with a vector of length $30n'$, in which exactly $5n'$ elements are randomly set to 1. Additionally, with probability $r$, the corresponding class label is replaced with a randomly chosen label to introduce noise. The resulting input vector thus has length $n = 40n'$. These input vectors are generated for $T$ rounds. Based on this procedure, we create datasets for $n' \in \{2, 4\}$ and $r \in \{0.0, 0.1\}$.

**Results**  The results on the synthetic data are shown in Figure 1. Our algorithm achieves comparable or better performance than the existing algorithms for datasets with $K \leq 24$. In contrast, when $K = 48$ or 96, the cumulative losses of our algorithm are larger than those of Gaptron with the hinge loss and Gappletron with the logistic loss. Note that these observations do not contradict the theoretical results: for large $K$, the upper bound on the cumulative 01 loss of Gappletron can

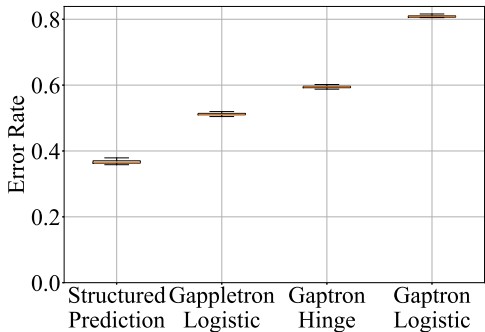

Figure 2: A box plot of error rates of the MNIST experiment for multiclass classification with bandit feedback.

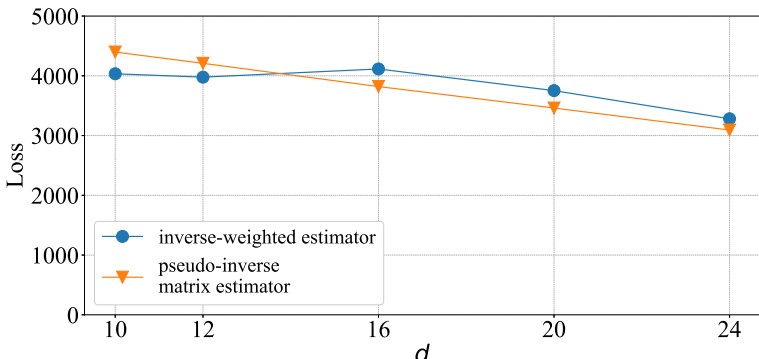

Figure 3: Results of the synsetic experiments in multilabel classification with bandit feedback. The horizontal axis shows the number of labels, and the vertical axis indicates the cumulative target loss.

be tighter than ours because of differences in the surrogate loss functions (see Appendix C.3 for details). Nevertheless, our structured prediction method does not fully demonstrate its potential in this setting, as the setup favors algorithms specialized for multiclass classification. It is also worth noting that by using the same decoding function as theirs, our approach can achieve the same order of the cumulative losses in online multiclass classification with bandit feedback.

### H.1.2 Real-world data

We also evaluate the algorithms on the MNIST dataset [31], a widely used benchmark of handwritten digit images.

**Result** The box plot in Figure 2 summarizes the misclassification rates. It shows that our method achieves the lowest misclassification rate, even though it is not specifically designed for multiclass classification, outperforming the existing algorithms on this real dataset with $K = 10$.

### H.2 Multilabel classification

In the results presented in Section 3, the algorithm based on the pseudo-inverse matrix estimator achieves a tighter upper bound in its dependence on $K$ compared to the one based on the inverse-weighted gradient estimator. To examine whether this theoretical result can also be observed empirically, we conduct experiments on multilabel classification with a fixed number of correct labels.

**Setup**   We compare two algorithms: the algorithm based on the inverse-weighted gradient estimator in Section 3.3 and the one based on the pseudo-inverse matrix estimator in Section 3.4.

**Data generation**   We generate synthetic data using the multilabel classification data-generation function in *scikit-learn* [38]. Specifically, we employ the `make_multilabel_classification` method in scikit-learn to generate $T$ multilabel samples with feature dimension $n$, label dimension $d$, and an average of $m$ correct labels per sample. We then extract only those samples that have exactly $m$ correct labels and repeat this process until we obtain $T = 10^4$ such samples. Based on this procedure, we create datasets with $n = 50$, $d \in \{10, 12, 16, 20, 24\}$, $m = 5$, and $T = 10^4$.

**Details of algorithms**   As the algorithm ALG for updating the linear estimator, we employ OGD as described in Section 3.2 with learning rate $\eta_t = B/\sqrt{2(10^{-8} + \sum_{i=1}^{t} \|\widetilde{\boldsymbol{G}}_t\|_{\mathrm{F}}^2)}$ and orthogonal projection. The small constant $10^{-8}$ in the denominator prevents division by zero. We fixed $B = 50$ for all experiments, and the other parameters were set according to their theoretical values. Under these settings, each experiment was repeated 10 times.

**Results**   The results are shown in Figure 3. When $d$ is small, the algorithm based on the inverse-weighted gradient estimator incurs a smaller loss, whereas when $d$ is large, the algorithm based on the pseudo-inverse matrix estimator performs better. The superiority of the inverse-weighted gradient estimator for small $d$ aligns with the theoretical result that it has a more favorable dependence on $T$. Similarly, the better performance of the pseudo-inverse matrix estimator for large $d$ agrees with the theoretical result that it does not depend explicitly on $K$. These experimental results thus provide empirical support for our theoretical findings.

