# OpenReview forum: "Bandit and Delayed Feedback in Online Structured Prediction"
_NeurIPS.cc/2025/Conference — NeurIPS 2025 poster_

### Official Review · Reviewer_4VTE · 2025-06-30

**Clarity:** 3
**Significance:** 2
**Originality:** 3
**Rating:** 4
**Confidence:** 3

**Summary:**

This paper studies the problem of online structured prediction with bandit and delayed feedback. Unlike the standard way of measuring the performance of online algorithms, the authors consider an alternative notion called surrogate regret, based on some surrogate loss (Fenchel-Young loss) introduced in prior work. While a standard bandit algorithm achieves a surrogate regret of O((KT)^{1/2}), where T is the time horizon and K is the size of the output set, the authors propose a new bandit algorithm achieving a regret of O(T^{2/3}) when the loss has some special form. Analogous results are also shown for the delayed feedback setting.

**Questions:**

See the weakness part above.

**Ethical Concerns:**

["NO or VERY MINOR ethics concerns only"]

**Final Justification:**

I am raising my score as the authors have addressed some of my initial concerns. Nonetheless, I hope they will clarify the notion of surrogate regret when revising the paper, as readers unfamiliar with it may have similar concerns to those I initially had.

**Limitations:**

Yes.

**Quality:**

2

**Strengths And Weaknesses:**

Strength:
The use of Fenchel-Young loss for structured prediction tasks seems interesting. The results are new. The regret bound of the proposed algorithm is independent of K, which is appealing when the size of the output set is large.

Weakness:
While using a carefully chosen surrogate loss to facilitate algorithm design is a good idea, I believe that the performance of an algorithm should still be evaluated with respect to the original target loss, or regret defined in terms of the target loss, which we ultimately care about. The surrogate regret proposed in this paper compares the target loss of the online algorithm to the surrogate loss of a fixed offline comparator, which raises concerns about interpretability and fairness. This mixed comparison (target vs. surrogate loss) may lead to misleading conclusions. A small surrogate regret does not necessarily imply strong performance on the original task, as the regret is not measured in a consistent way.
For instance, in the full-information setting with delay D=1, one can actually achieve constant surrogate regret, whereas it is well known that under the standard regret definition (based on the target loss), even the no-delay setting admits a lower bound of about T^{1/2}. It seems that the surrogate regret upper bounds achieved by this paper are based on the assumption that the target loss is less than the surrogate loss by about a factor of (1-a), and the surrogate regret bounds contain a factor of 1/a. This means that if one uses a surrogate loss close to the targe loss, with the parameter a approaching zero, than the surrogate regret could become arbitrarily large.

---

> ### Author Rebuttal · Authors · 2025-07-31
>
> Thank you very much for your thoughtful and constructive review. Below are our responses to your comments.
>
> ## **On the surrogate regret**
>
> We thank the reviewer for raising this important point. Firstly, we would like to clarify a possible misunderstanding in the following statement by the reviewer:
> > The surrogate regret proposed in this paper compares the target loss of the online algorithm to the surrogate loss of a fixed offline comparator, which raises concerns about interpretability and fairness. This mixed comparison (target vs. surrogate loss) may lead to misleading conclusions. A small surrogate regret does not necessarily imply strong performance on the original task, as the regret is not measured in a consistent way. ...
>
> The surrogate regret is not a new concept proposed in this paper. It was explicitly introduced by Van der Hoeven (2020, [38]) and has been a central concept in the subsequent work (Van der Hoeven et al., 2021, [39]; Sakaue et al., 2024, [35]). Below, we elaborate on why this measure is appropriate and justified in our setting.
>
> As you rightly noted, the relationship between the target loss and the surrogate loss deserves careful attention. This issue has been thoroughly discussed in prior work, particularly in Sakaue et al. (2024, [35]), and our choice of target and surrogate losses in this paper follows the standard framework established in the literature. Accordingly, the surrogate loss used in our setting does not give the algorithm any unfair advantage, nor does it result in pathological behavior where the constant $a$ becomes arbitrarily small and causes the regret bound to deteriorate excessively. For example, in the case of classification with the logistic loss, which is a standard surrogate loss, the constant is $a = 1 - \ln 2 \approx 0.3$, which is not excessively small and ensures that the resulting bound, scaling as $1/a$, remains meaningful.
>
> More broadly, if one were to analyze the regret solely with respect to the target loss, one would need to consider hypothesis classes consisting of models that predict discrete labels $\boldsymbol{y}$ from input features $\boldsymbol{x}$. However, such models are often computationally intractable to train. In practice, it is standard to adopt the surrogate loss framework, as advocated in the seminal work by Bartlett et al. (2006, [3]), as discussed in Section 1. In this approach, we can efficiently learn estimators that map a feature $\boldsymbol{x}$ to a score vector $\boldsymbol{\theta}$ in the continuous Euclidean space $\mathbb{R}^d$, by applying gradient-based methods to a surrogate loss.
>
> From this perspective, it is both natural and important to study how well we can bound the target loss in comparison to the best possible performance achievable under the surrogate loss framework. This leads to the study of the surrogate regret, which indeed generalizes the finite mistake bound of the classical Perceptron on linearly separable data (Rosenblatt 1958, [34]; cf. Van der Hoeven, 2020, [38], Section 1) and has been the focus of a growing body of work—Van der Hoeven (2020, [38]), Van der Hoeven et al. (2021, [39]), and Sakaue et al. (2024, [35]). This conceptual foundation is central to our work.
>
> In summary, we understand the reasoning behind the reviewer’s concern, but we emphasize that the surrogate regret framework is theoretically sound and widely accepted and that the choice of target and surrogate losses in our paper is fully aligned with the established literature. Therefore, adopting the surrogate regret framework does not weaken the significance of our contributions. We will clarify this point in the revised version of the paper with further discussion.

---

> > ### Comment · Reviewer_4VTE · 2025-08-04
> >
> > While I agree that using a surrogate loss is a good idea, I remain concerned about the definition of surrogate regret, as it allows for constant surrogate regret even in settings where standard regret measures have known lower bounds of $\Omega(\sqrt{T})$. This makes it unclear how to interpret results that report small surrogate regret and raises questions about the informativeness of this measure. Although a small number of prior works have used such a definition, I am not sure if it is well-accepted by the online learning community. I continue to believe that a more reasonable definition would compare online and offline algorithms using the same loss—either the target loss or the surrogate loss—for both.

---

> > > ### Comment · Reviewer_jZKi · 2025-08-04
> > >
> > > I would like to correct something. Surrogate regret has been studied long before [38]. It obtained a large amount of attention after the analysis of the perceptron algorithm in the non-separable case (Gentile and Littlestone, 1999), the introduction of the second order perceptron by Cesa-Bianchi et al. (2005) and the Banditron algorithm by Kakade et al. (2008), but even before that surrogate regret was studied. See chapter 8.3 of Orabona  (2019) for a brief historic summary.
> > >
> > > To address one of the concerns of reviewer 4VTE: surrogate regret is well accepted by the online learning community. It appears in several textbooks (for example Cesa-Bianchi and Lugosi, 2006, chapter 12) and many papers, including the COLT 2018 best paper (Foster et al, 2018, section 3).
> > >
> > >
> > >
> > > References
> > >
> > >
> > > Cesa-Bianchi, N., Conconi, A., & Gentile, C. (2005). A second-order perceptron algorithm. SIAM Journal on Computing, 34(3), 640-668.
> > >
> > > Cesa-Bianchi, N., & Lugosi, G. (2006). Prediction, learning, and games. Cambridge university press.
> > >
> > > Foster, D. J., Kale, S., Luo, H., Mohri, M., & Sridharan, K. (2018, July). Logistic regression: The importance of being improper. In Conference on learning theory (pp. 167-208). PMLR.
> > >
> > > Gentile, C., & Littlestone, N. (1999, July). The robustness of the p-norm algorithms. In Proceedings of the twelfth annual conference on Computational learning theory (pp. 1-11).
> > >
> > > Kakade, S. M., Shalev-Shwartz, S., & Tewari, A. (2008, July). Efficient bandit algorithms for online multiclass prediction. In Proceedings of the 25th international conference on Machine learning (pp. 440-447).
> > >
> > > Orabona, F. (2019). A modern introduction to online learning. arXiv preprint arXiv:1912.13213.

---

> > > > ### Author Response · Authors · 2025-08-04
> > > > **Reply to Reviewer 4VTE and Reviewer jZKi**
> > > >
> > > > We would like to express our deep appreciation for the dedication of both Reviewer jZKi and Reviewer 4VTE to reviewing our paper.
> > > >
> > > > ### **To Reviewer 4VTE**
> > > >
> > > > We sincerely thank the reviewer for taking the time to follow up. We greatly appreciate this opportunity to deepen the discussion on the use of the surrogate regret.
> > > >
> > > > First, we would like to apologize for any unnecessary confusion. Our rebuttal focused on recent relevant works following Van der Hoeven (2020), where the surrogate regret is particularly explicit. However, as described by Reviewer jZKi, the surrogate regret indeed has a long and well-established history in the online learning community. Its importance continues to grow, especially since the surrogate loss framework has become the default choice in practice due to its computational efficiency.
> > > >
> > > > To clarify, we fully agree that your view on the standard regret, along with related results such as the $\Omega(\sqrt{T})$ lower bound, is also important. We thank the reviewer for drawing our attention to this point. In the revised version, we will include an extensive discussion on the surrogate regret and its relationship to the standard regret, along with references provided by Reviewer jZKi.
> > > >
> > > > We hope the discussion thus far helps resolve your concern regarding the use of the surrogate regret. Please feel free to reach out if any questions remain or if further clarification would be helpful.
> > > >
> > > > ### **To Reviewer jZKi**
> > > >
> > > > We sincerely thank the reviewer for their kind engagement. We feel truly fortunate to have support from such an active and knowledgeable reviewer. We plan to include an extensive discussion on the surrogate regret, along with the historical context you kindly provided.

---

> > > > > ### Comment · Reviewer_4VTE · 2025-08-06
> > > > >
> > > > > From a quick reading of the earlier works and textbooks mentioned by the reviewer jZKi, my impression is that the notion of surrogate regret is used only implicitly, mainly as a tool to bound the target loss (e.g., 0-1 loss). It seems that they all then try to argue that the surrogate loss is small, in order to conclude that the target loss is small. This looks very reasonable to me.
> > > > >
> > > > > On the other hand, the idea of making surrogate regret explicit and aiming to minimize it for general classes of target and surrogate loss functions appears to be a more recent development. What concerns me is the lack of appropriate constraints in this broader setting. In particular, a small surrogate regret does not convey much information unless we also know that the surrogate loss of the offline algorithm is small. Otherwise, we will have strange results such as having constant surrogate regret but large standard regret.
> > > > >
> > > > > Therefore, I believe we should be cautious when studying surrogate regret in general settings without carefully constraining the loss functions involved. In extreme cases, I suspect that even negative surrogate regret might be possible. This also raises questions about the tightness of the upper bounds presented in Section 3, and I am now quite curious about the corresponding lower bounds.
> > > > >
> > > > > This is just my own opinion, based on my very limited knowledge of this line of work, and I may have misunderstood some of those prior work.

---

> > > > > > ### Author Response · Authors · 2025-08-06
> > > > > >
> > > > > > We deeply thank the reviewer again for their continued dedication to this discussion; not many reviewers commit to this level of thoughtful back-and-forth, including independently revisiting earlier studies. Below, we would like to address concerns raised in your reply.
> > > > > > >*"From a quick reading of the earlier works and textbooks mentioned by the reviewer jZKi, my impression is that the notion of surrogate regret is used only implicitly, mainly as a tool to bound the target loss (e.g., 0-1 loss). It seems that they all then try to argue that the surrogate loss is small, in order to conclude that the target loss is small. This looks very reasonable to me."*
> > > > > > >
> > > > > > >*"On the other hand, the idea of making surrogate regret explicit and aiming to minimize it for general classes of target and surrogate loss functions appears to be a more recent development. "*
> > > > > >
> > > > > > Our analysis of the surrogate regret (and those following Van der Hoeven, 2020), as well as the classical analyses in the references provided by Reviewer jZKi, adheres to the same reasonable approach. The difference lies solely in terminology, namely, whether or not the term “surrogate regret” is explicitly used. Specifically, both lines of literature are aligned in spirit with the analysis outlined below:
> > > > > >
> > > > > > 1. **Making the surrogate loss small.** As shown in Sections 3.2 and 3.3, we apply online gradient descent to the surrogate loss using the gradient estimators $\hat{\boldsymbol{G}}_t$, obtaining a bound on the regret solely in terms of the surrogate loss: $$\mathbb{E}\left[ \sum\_{t=1}^TS\_t(\boldsymbol{W}\_t)-\sum\_{t=1}^TS\_t(\boldsymbol{U}) \right]\le\sqrt{2}B\sqrt{\mathbb{E}\left[\sum\_{t=1}^T\\|\hat{\boldsymbol{G}}\_t\\|^2\_{\mathrm{F}}\right]}.$$
> > > > > > 2. **Concluding that the target loss is small.** To translate the above bound into a bound on the target loss, we must specify how the label $\hat{\boldsymbol{y}}_t$ is predicted from the score $\boldsymbol{W}_t\boldsymbol{x}_t$. In our work, RDUE (Algorithm 2) plays this role and provides the bound in Theorem 3.5: $$\mathbb{E}\left[\sum\_{t=1}^TL_t(\hat{\boldsymbol{y}}\_t)\right]\le\sum\_{t=1}^TS\_t(\boldsymbol{U})+\left(\frac{b}{2a}+1 \right)B\sqrt{KT}.$$
> > > > > >
> > > > > > >*"What concerns me is the lack of appropriate constraints in this broader setting. In particular, a small surrogate regret does not convey much information unless we also know that the surrogate loss of the offline algorithm is small. Otherwise, we will have strange results such as having constant surrogate regret but large standard regret."*
> > > > > > >
> > > > > > >*"Therefore, I believe we should be cautious when studying surrogate regret in general settings without carefully constraining the loss functions involved."*
> > > > > >
> > > > > > We believe this part may involve a misunderstanding. While our framework is described using general target losses $L_t$ and surrogate losses $S_t$, it still encompasses the standard classification settings.
> > > > > >
> > > > > > For example, as in lines 168–173, we can apply our method to the 0–1 target loss and the logistic surrogate loss (as with Theorem 6 in the COLT 2018 best paper by Foster et al.). Then, the constant in Assumption 3.2 is $a=1-\ln 2\approx0.3$, and hence the bound in Theorem 3.5 yields a meaningful guarantee, as discussed in the rebuttal.
> > > > > >
> > > > > > That is, **the general framework we adopt, originally proposed by Sakaue et al. (COLT 2024), is indeed broad, but importantly, designed so as to satisfy such "constraints" traditionally considered in standard settings.** We are also confident that our work has been cautious in selecting surrogate losses: lines 263–266 and Appendix C contain a detailed discussion of the subtleties involved in surrogate loss selection, which was noted and appreciated by Reviewer 4Bpe.
> > > > > >
> > > > > > >*"In extreme cases, I suspect that even negative surrogate regret might be possible. This also raises questions about the tightness of the upper bounds presented in Section 3, and I am now quite curious about the corresponding lower bounds."*
> > > > > >
> > > > > > While it would be possible to obtain negative surrogate regret if one uses an inappropriately large surrogate loss, such choices are not intended. Obtaining tight lower bounds is a challenging task, as noted in lines 266–269. Still, the literature does provide lower bounds that demonstrate the surrogate regret cannot be negative in reasonable multiclass classification settings: for instance, Van der Hoeven et al. (2021, Corollary 1) give a lower bound of $\Omega(KB^2+\sqrt{T})$ for the smooth hinge surrogate under graph feedback, and Sakaue et al. (2024, Theorem 13) provide a lower bound of $\Omega(B^2/\ln^2d)$ for the logistic surrogate under full-information feedback. The latter is the full-information version of our example above, and the lower bound directly implies that the surrogate regret cannot be negative under the same setting with weaker bandit feedback.
> > > > > >
> > > > > > Thank you again for your thoughtful engagement throughout. We hope the above addresses your concerns, and we would be happy to clarify any remaining points.

---

> > > > > > > ### Comment · Reviewer_4VTE · 2025-08-07
> > > > > > >
> > > > > > > I am not sure whether you can guarantee that the surrogate loss of the offline algorithm ($\sum_t S_t(U)$) is small. A standard regret bound—of the form $\sum_t S_t(W_t) - \sum_t S_t(U)$—does provide an upper bound on the surrogate loss of the online algorithm ($\sum_t S_t(W_t)$), but it serves as a lower bound on the surrogate loss of the offline algorithm ($\sum_t S_t(U)$). So we still cannot conclude that the target loss of the online algorithm is small. I am really confused.
> > > > > > >
> > > > > > > For specific losses where we know that $\sum_t S_t(U)$ is small, the results on surrogate regret make sense to me—including yours when specialized to these cases. For general losses where we do not know an upper bound on $\sum_t S_t(U)$, I have difficulty interpreting such surrogate regret results. One example is the result in Section 4 in the delayed full-information setting. There you can have constant surrogate regret when the delay is constant, but there is a large $\Omega(\sqrt{T})$ lower bound for the standard regret. It seems to me that you can have such a strange result because the surrogate loss is significantly larger than the target loss, according to Assumption 4.1. Do I misunderstand something?

---

> > > > > > > > ### Author Response · Authors · 2025-08-07
> > > > > > > >
> > > > > > > > We sincerely thank the reviewer for their continued engagement. We believe the confusion arises from what the surrogate regret bound does and does not guarantee. Below, we respond to each point.
> > > > > > > >
> > > > > > > > > *"I am not sure whether you can guarantee that the surrogate loss of the offline algorithm ($\sum_t S_t(U)$) is small. A standard regret bound—of the form $\sum_t S_t(W_t) - \sum_t S_t(U)$—does provide an upper bound on the surrogate loss of the online algorithm ($\sum_t S_t(W_t)$), but it serves as a lower bound on the surrogate loss of the offline algorithm ($\sum_t S_t(U)$). So we still cannot conclude that the target loss of the online algorithm is small. I am really confused."*
> > > > > > > >
> > > > > > > > We would like to clarify that the analysis based on the surrogate regret is intended to say:
> > > > > > > >
> > > > > > > > **"If** $\sum_{t=1}^T S\_t(\boldsymbol{U})$ is small (i.e., the best offline comparator effectively minimizes the surrogate loss), **then** the target loss $\sum_{t=1}^T \mathbb{1}[\hat{\boldsymbol{y}}\_t \neq \boldsymbol{y}\_t]$ is also guaranteed to be small."
> > > > > > > >
> > > > > > > > This argument does **not** claim that $\sum_{t=1}^T S\_t(\boldsymbol{U})$ is small; indeed, such a claim is impossible in general. If no comparator can achieve a small surrogate loss in hindsight, then it is fundamentally impossible to expect small target loss under the surrogate loss framework. This is not unique to our work, but is shared by the long line of studies cited by Reviewer jZKi, originating from the Perceptron. That said, in many practical situations, the surrogate loss **can** be made small efficiently and effectively. The surrogate regret framework aims to take advantage of this favorable scenario to provide strong bounds on the target loss.
> > > > > > > >
> > > > > > > > > *"For specific losses where we know that $\sum_t S_t(U)$ is small, the results on surrogate regret make sense to me—including yours when specialized to these cases. For general losses where we do not know an upper bound on $\sum_t S_t(U)$, I have difficulty interpreting such surrogate regret results."*
> > > > > > > >
> > > > > > > > We believe there may be a misinterpretation of what is meant by “general losses.” While we write the surrogate loss as a general symbol $S\_t$ for notational convenience and conceptual generality (following Sakaue et al., 2024), in practice, $S\_t$ is always instantiated as a **specific loss**, such as the logistic loss. Once the surrogate loss is specified, the constants $a$ and $b$ appearing in the surrogate regret bounds (e.g., Theorem 3.5) are determined accordingly, yielding a concrete and interpretable bound on the target loss.
> > > > > > > >
> > > > > > > > > *"One example is the result in Section 4 in the delayed full-information setting. There you can have constant surrogate regret when the delay is constant, but there is a large $\Omega(\sqrt{T})$ lower bound for the standard regret. It seems to me that you can have such a strange result because the surrogate loss is significantly larger than the target loss, according to Assumption 4.1. Do I misunderstand something?"*
> > > > > > > >
> > > > > > > > Let us clarify using typical examples.
> > > > > > > >
> > > > > > > > The $\Omega(\sqrt{T})$ lower bound likely refers to results in the agnostic setting, such as Daniely et al. (2011, "Multiclass Learnability and the ERM principle"), specifically:
> > > > > > > > $$
> > > > > > > > \sum_{t=1}^T\mathbb{1}[\hat{\boldsymbol{y}}\_t\neq\boldsymbol{y}\_t]-\inf_{h\in\mathcal{H}}\sum_{t=1}^T\mathbb{1}[h(\boldsymbol{x}\_t)\neq\boldsymbol{y}\_t]=\Omega\left(\sqrt{\mathrm{Ldim}(\mathcal{H})T}\right),
> > > > > > > > $$
> > > > > > > > where $\mathcal{H}$ is a hypothesis class and $\mathrm{Ldim}$ is the Littlestone dimension.
> > > > > > > >
> > > > > > > > By contrast, typical constant surrogate regret bounds, such as Van der Hoeven (2020, Theorem 1), are of the form:
> > > > > > > > $$
> > > > > > > > \sum_{t=1}^T \mathbb{1}[\hat{\boldsymbol{y}}\_t \neq \boldsymbol{y}\_t] - \sum_{t=1}^T S\_t(\boldsymbol{U}) = O(\\| \boldsymbol{U} \\|^2),
> > > > > > > > $$
> > > > > > > > where $S\_t$ is, e.g., the logistic loss. Our Theorem 4.3 in Section 4 is, roughly speaking, an extension of this to the delayed setting.
> > > > > > > >
> > > > > > > > Both approaches are valid and serve different purposes; one never makes the other strange. The $\Omega(\sqrt{T})$ lower bound on the standard regret highlights the theoretical worst-case limitation in the agnostic setting, whereas the surrogate regret bound can offer stronger bounds on the target loss **if** $\sum_{t=1}^T S\_t(\boldsymbol{U})$ is small (e.g., when the data is nearly linearly separable). This conditionality is not a flaw, but a reasonable approach to obtaining strong guarantees in practice.
> > > > > > > >
> > > > > > > > To address the comment on Assumption 4.1, surrogate losses are commonly designed to upper bound the target loss (cf. Figure 1 in Masnadi-Shirazi and Vasconcelos, 2008, "On the Design of Loss Functions for Classification"), but they are **not** intended to be inappropriately large, as discussed in our last reply.
> > > > > > > >
> > > > > > > > We thank you once again for your deep engagement and thoughtful comments. We will make sure to elaborate more on the interpretation of the surrogate regret in the revised version. Please don’t hesitate to let us know if further clarification would be helpful.

---

> > > > > > > > > ### Comment · Reviewer_4VTE · 2025-08-07
> > > > > > > > >
> > > > > > > > > Thank you for your clarification. It seems that interpreting results on surrogate regret and standard regret requires different mindsets. I have learned a lot from our discussions.

---

### Official Review · Reviewer_4Bpe · 2025-07-01

**Clarity:** 3
**Significance:** 4
**Originality:** 3
**Rating:** 5
**Confidence:** 2

**Summary:**

The paper studies online structured prediction under bandit and delayed feedback. In the bandit setup it proposes an algorithm guaranteeing a surrogate regret of $O(\sqrt{KT})$, where $K$ is the size of the output set, and an algorithm with surrogate regret of $O(T^{2/3})$ independent of $K$. In the case of delayed feedback with delay $D$, the paper proposes an algorithm guaranteeing regret $O(\min(D^2,D^{2/3}T^{1/3}))$ in the full information setting, and also provides algorithms with surrogate regret guarantees for the case in which both delay and bandit feedback are present.

**Questions:**

None

**Ethical Concerns:**

["NO or VERY MINOR ethics concerns only"]

**Final Justification:**

After reading the responses I remain quite positive about this work and will maintain my score.

**Limitations:**

yes

**Quality:**

4

**Strengths And Weaknesses:**

The paper addresses a well-motivated problem. The result presented in Section 3.4 is particularly noteworthy and I find it rather surprising.

Overall, the theoretical results appear to be sound, and the manuscript is generally written with care and clarity. I particularly appreciated the authors for their clear (and honest) comparison with related work (e.g., discussion on pages 261–269).

However, in some sections, the manuscript becomes quite dense with technical details, which may hinder its understanding especially for readers not deeply familiar with the specific techniques employed. In some parts of the paper, a more intuitive or high-level exposition accompanying the formal developments would help in understanding the main conceptual contributions. One example is Section 3.4, where the pseudo-inverse matrix estimator is introduced.

The results up to Section 3.3 rely on a combination of standard tools for handling bandit feedback and prior  techniques (e.g., from \[35]). While the derivations are technically non-trivial, this part of the paper seems to lack substantial conceptual contributions/new ideas. A similar observation may apply to the delayed feedback setting: the presentation does not clearly articulate what new ideas or techniques are necessary to obtain the results, making it challenging to assess the novelty without delving deeply into the proofs.

Finally, Assumption 3.6 (iii) is not explicitly discussed, despite the apparent importance of the parameter $\omega$ in the subsequent guarantees.

---

> ### Author Rebuttal · Authors · 2025-07-31
>
> Thank you for taking the time to provide this valuable review. Below are our responses to the comments.
>
> ## **Intuitive explanations**
>
> > However, in some Sections, the manuscript becomes quite dense with technical details, which may hinder its understanding especially for readers not deeply familiar with the specific techniques employed. In some parts of the paper, a more intuitive or high-level exposition accompanying the formal developments would help in understanding the main conceptual contributions. One example is Section 3.4, where the pseudo-inverse matrix estimator is introduced.The results up to Section 3.3 rely on a combination of standard tools for handling bandit feedback and prior techniques (e.g., from [35]). While the derivations are technically non-trivial, this part of the paper seems to lack substantial conceptual contributions/new ideas. A similar observation may apply to the delayed feedback setting: the presentation does not clearly articulate what new ideas or techniques are necessary to obtain the results, making it challenging to assess the novelty without delving deeply into the proofs.
> We apologize for the lack of clarity in the original explanation. Below we provide further clarification on the examples cited in Section 3.4 and the details related to delayed feedback in Sections 4 and 5.
>
> **Regarding Section 3.4**
>
> - We propose a pseudo-inverse matrix estimator inspired by studies on adversarial linear bandits and adversarial combinatorial full-bandits. By utilizing the loss values, the estimator achieves a $K$-independent bound that is particularly effective for complex structured prediction problems, such as multilabel classification.
> - Technically, unlike inverse-weighted gradient estimators, our approach constructs an unbiased estimator without relying on the inverse weighting by probability $p\_t$. This is crucial for removing the dependence on $K$, which arises from the lower bound on $p\_t$. The upper bound for this approach is proven using properties of positive semi-definite matrices and pseudo-inverse matrices (see Lemmas D.5–D.7 for details).
> - Additionally, we identify the conditions under which this estimator can be used and characterize the class of loss functions that satisfy these conditions. This is also technically novel.
>
> **Regarding Sections 4 and 5**
> - We discovered that existing algorithms for online convex optimization with delay, which achieve AdaGrad-type regret bounds, can be effectively utilized.
> - In Section 4, the core idea is already presented in the proof sketch of Theorem 4.3. The key technical point is leveraging the negative term via the AdaGrad-type regret bound and the inequality $\\|\boldsymbol{G}\_t\\|\_\mathrm{F}^2 \le bS\_t(\boldsymbol{W}\_t)$. (These techniques remain important even when applying the $D$-copy approach mentioned in the rebuttal to Reviewer jZKi.)
> - The algorithm and proof ideas in Section 5 are essentially a combination of the discussions already presented in Sections 3 and 4.
>
> We will emphasize these points more clearly in the revision and do our best to provide more intuitive explanations.
>
> ## **Parameter $\omega$ of Assumption 3.6 (iii)**
>
> > Finally, Assumption 3.6 (iii) is not explicitly discussed, despite the apparent importance of the parameter $\omega$ in the subsequent guarantees.
>
> Thank you for the comment. As pointed out, the value of $\omega$ is indeed important. In the revised version, we will update Corollary 3.8 to include not only the final upper bound but also the explicit values of $\omega$ for each setting—namely, multiclass classification, multilabel classification, and ranking. Specifically, $\omega$ is $d^2$ for multiclass classification, $d^5 / 4m(d - m)$ for multilabel classification with $m$ correct labels, and $m^5$ for the ranking (see Appendix D.4 for details).

---

### Official Review · Reviewer_jZKi · 2025-07-02

**Clarity:** 3
**Significance:** 3
**Originality:** 3
**Rating:** 5
**Confidence:** 4

**Summary:**

This paper considers online structured prediction with surrogate losses. This is a sequential setting where in each round t = 1,…, T the learner observes a context, issues a prediction from a discrete set and subsequently suffers a non-convex loss. The paper mostly considers the bandit setting, where the learner only sees the loss for the chosen prediction. In the delayed version of this problem the learner only observes the loss after some rounds. The goal is to control the surrogate regret, which is the difference between cumulative the losses suffered by the learner, and the minimized cumulative surrogate loss.

The authors exploit a recently developed technique that sharpens the relation between the discrete loss of the learner and the surrogate loss of the learner if the surrogate loss is a fenchel young loss. In the bandit setting this leads to a surrogate regret bound of order $\sqrt{KT}$, where K is the cardinality of the prediction set. In the delayed feedback setting with constant delay D with full information feedback, the authors obtain a $D^2$ surrogate regret bound. In the delayed feedback setting with constant delay D with bandit feedback, the authors obtain a $\sqrt{T(K + D)}$ surrogate regret bound.

**Questions:**

I am wondering of my construction for the constant delay actually works. If so, would this be optimal?

I am wondering if the bounds for the delayed bandit setting are tight

**Ethical Concerns:**

["NO or VERY MINOR ethics concerns only"]

**Final Justification:**

The authors addressed my concern about simple improvements being possible in the delayed bandit feedback setting. I maintain that the results in the delayed full information setting could be improved, but I also believe that the contributions in this paper are interesting for the general community.

**Limitations:**

yes

**Paper Formatting Concerns:**

.

**Quality:**

3

**Strengths And Weaknesses:**

For the remainder of this review I will focus on the multiclass classication setting with the logistic loss, a special case of the setting considered here. For the non-delayed bandit feedback setting, the results presented in this paper present a stark improvement over prior work. In prior work, the best known bound is Ksqrt{T}, which is a factor sqrt{K} worse.

The results for the delayed full information are a bit weak I suspect. If we have constant delay, we could just run D copies of the non-delayed algorithm, where each copy makes a prediction every D time steps. Since the non-delayed algorithm has surrogate regret O(1), this would lead to a surrogate regret bound of order D. I think that a similar surrogate regret bound should be obtainable for delay that is not constant.

It is not clear if the results for the delayed bandit setting are tight. Nevertheless, both results are new and show that the randomized decoding technique from earlier work can be extended to obtain interesting results beyond the setting the technique was originally developed for.

---

> ### Author Rebuttal · Authors · 2025-07-31
>
> We sincerely appreciate the time and effort you devoted to providing a constructive review. Below, we present our responses to your comments.
>
> ## **Q1. Comparison between the $D$ copy approach and ours**
>
> > The results for the delayed full information are a bit weak I suspect. If we have constant delay, we could just run $D$ copies of the non-delayed algorithm, where each copy makes a prediction every $D$ time steps. Since the non-delayed algorithm has surrogate regret $O(1)$, this would lead to a surrogate regret bound of order $D$.
>
> > I am wondering of my construction for the constant delay actually works. If so, would this be optimal?
>
> Thank you very much for your insightful comment.
> We have examined how the approach of preparing $D$ copies of the non-delayed algorithm (as investigated by Weinberger and Ordentlich, 2002 [41]; Joulani et al., 2013 [23]) would perform in our setting. In what follows, we call this approach the "$D$-copy approach" for simplicity.
>
> As you pointed out, in the full-information feedback setting, it is indeed possible to improve the regret bound to $O(D)$. In contrast, in the bandit feedback setting, we found that our approach has advantages over the $D$-copy approach combined with the inverse-weighted gradient estimator and the pseudo-inverse matrix estimator. Specifically,
>
> - In the full-information setting, the $D$-copy approach combined with the approach of Sakaue et al. (2024, [35]) achieves a regret bound of $O(D)$, which is better than our $O(D^2)$ bound.
> - In the bandit setting with the inverse-weighted gradient estimator, the $D$-copy approach applied with our algorithm for Theorem 3.5 yields a bound of $O(\sqrt{K D T})$, which is worse than our $O(\sqrt{(K + D) T})$ bound. Interestingly, such degradation in the regret bound when using $D$ copies naively is also known in the literature of multi-armed bandits: For the classical multi-armed bandit problem, using $D$ copies yields a regret bound of $O(\sqrt{ NDT})$, while the minimax regret for the problem is $\Theta(\sqrt{(N + D) T})$ (Corollary 11 in Cesa-Bianchi et al. 2016 [9]). Thus, our bound of $O(\sqrt{(K + D) T})$ can be seen as an improvement over the naive $O(\sqrt{K D T})$ bound.
> - In the bandit setting with the pseudo-inverse matrix estimator, the $D$-copy approach with our algorithm for Theorem 3.7 achieves a bound of $O(D^{1/3} T^{2/3})$, which is the same as our bound in Theorem 5.2. However, as the ODAFTRL paper points out, it is undesirable because each algorithm only leverages $\frac{T}{D+1}$ losses (see page 4 in Flaspohler et al. 2021, [17]), limiting its empirical performance.
>
> In summary, the $D$-copy approach has an advantage in terms of worst-case regret in the full-information setting. However, in the bandit setting, our approach demonstrates superiority against the $D$-copy approach combined with the algorithms for the non-delayed bandit setting, regardless of which estimator is used.
>
> We would like to emphasize that our main contribution lies in the bandit feedback setting for online structured prediction. The full-information delayed feedback setting is handled simply by combining the decoding method of Sakaue et al. (2024, [35]) with ODAFTRL and is not central to our work; it occupies only half a page in our 9-page manuscript.
>
> That said, we agree that it is beneficial to note that the improvement is achievable by the $D$-copy approach in the full-information setting. In the revised version, we will carefully discuss how the $D$-copy approach improves the bound in the full-information setting, and how our method demonstrates advantages over the $D$-copy approach in the bandit setting, in terms of both regret bounds and average-case performance. We again appreciate the reviewer's insightful comment.
>
> ## **Regarding variable delay (mentioned in Weaknesses)**
>
> > I think that a similar surrogate regret bound should be obtainable for delay that is not constant.
>
> We have provided the algorithm and the upper bounds for the variable delay setting, which is deferred to Appendix G and mentioned in Line 379 due to space constraints.
>
> Here, we briefly describe the results.
> Let $\tau_t$ denote the delay time of the feedback at time $t$ and $\tau_\ast$ the maximum delay. We obtain an upper bound of $O(\sqrt{\tau_{1:T}}+\tau_\ast)$ in the full-information setting. In the bandit setting, by employing the inverse-weighted gradient estimator and the pseudo-inverse matrix estimator, we obtain upper bounds of $O(\sqrt{KT}+\sqrt{\tau_{1:T}}+\tau_\ast)$ and $O(T^{1/6}\sqrt{\tau_{1:T}}+\omega^{1/3}T^{2/3}+\tau_\ast)$, respectively. Please see Appendix G for the detailed statements and proofs.
>
> ## **Q2. Tightness of results for the delayed bandit setting**
>
> > It is not clear if the results for the delayed bandit setting are tight. Nevertheless, both results are new and show that the randomized decoding technique from earlier work can be extended to obtain interesting results beyond the setting the technique was originally developed for.
>
> > I am wondering if the bounds for the delayed bandit setting are tight
>
> Thank you for the important comment. Proving lower bounds and discussing the tightness of the upper bounds we provided are certainly an important direction. However, in the analysis of surrogate regret bounds, particularly under the bandit feedback setting, proving lower bounds is challenging. Below, we provide information regarding the difficulty of establishing lower bounds in the bandit feedback setting.
>
> Van der Hoeven et al. (2021, [39]) proved an \$\Omega(\sqrt{T})\$ lower bound in the "graph" feedback setup, which is a variant of the bandit feedback model. However, no surrogate regret lower bound is known for the "bandit" feedback setting, as their construction does not apply to this case (although this may suggest that the lower bound in the bandit setup is possibly \$\Omega(\sqrt{T})\$). Hence, there seems to be an additional difficulty in constructing surrogate regret lower bounds in the bandit setting. Extending such constructions to the delayed feedback setting would be even more challenging and is beyond the scope of this study. We will include these discussions in the revised version.

---

### Official Review · Reviewer_vmLL · 2025-07-02

**Clarity:** 3
**Significance:** 3
**Originality:** 3
**Rating:** 4
**Confidence:** 2

**Summary:**

This paper studies the problem of online structured prediction under bandit and delayed feedback by proposing algorithms based on surrogate regrets. Leveraging properties of structured surrogate losses and randomized decoding, the paper proposes two algorithms,  one using an inverse-weighted gradient estimator achieving a regret bound of $O(KT)$,  and another based on a pseudo-inverse matrix estimator achieving regret independent of label space size K, specifically $O(T^{2/3})$. They further extend their analysis to the delayed full-information setting, proposing a delayed FTRL algorithm that attains a regret of $O(\min⁡{D^2,D^{2/3}T^{1/3}})$.

**Questions:**

1. In section 3.4, the algorithm use RUDE decoding with $T >> B^2K$ and claimed that this is only for simplicity. Could you elaborate on why this makes things easier and how this can be lifted?

2. In section 3.4, the paper defines bandit feedback as observing the scalar loss $L_t(y^t)$, but then states the algorithm only uses the indicator $\mathbb{I}[y^t=yt]$, then does the algorithm still holds for general bandit feedback?

3. As the algorithms depend on assumptions about known invertibility and boundedness of feature spaces, can you address the computational tractability of required oracles (e.g., decoding under complex structure)?

3. In practice, I am assuming $D$ can be non uniform across steps and unknown to the algorithm beforehand. But here the algorithm needs to know $D$ in advance. Can this assumption be removed?

**Ethical Concerns:**

["NO or VERY MINOR ethics concerns only"]

**Final Justification:**

This work unifies structure prediction with Fenchel-Young loss framework and the regret bounds obtained under the framework are comparable to existing ones. From my understanding, the technical analysis and the algorithm designs are new and rigorous, and deserves acceptance.

**Limitations:**

yes

**Quality:**

3

**Strengths And Weaknesses:**

One strength of the paper is that it unifies structure prediction with Fenchel-Young loss framework and the regret bounds obtained under the framework are comparable to existing ones. From my understanding, the technical analysis and the algorithm designs are new and rigorous.

My main concerns are with the computation complexity of the algorithms. Although the framework generalize upon existing works (from multi class prediction to ranking), the algorithm has some assumptions on known invertibility and decoding oracle). I am not sure of the computation feasibility and complexity of the algorithms. I also have some minor presentation questions (see the questions part for more detailed questions).

---

> ### Author Rebuttal · Authors · 2025-07-31
>
> Thank you very much for taking the time to provide helpful reviews. Below are our responses to your comments.
>
> ## **Weakness & Q3. Computational tractability**
> > My main concerns are with the computation complexity of the algorithms. Although the framework generalize upon existing works (from multi class prediction to ranking), the algorithm has some assumptions on known invertibility and decoding oracle). I am not sure of the computation feasibility and complexity of the algorithms.
>
> > 3. As the algorithms depend on assumptions about known invertibility and boundedness of feature spaces, can you address the computational tractability of required oracles (e.g., decoding under complex structure)?
>
> We thank the reviewer for clearly stating the concern regarding the computational aspect. Below, we explain the invertibility of matrices and the computational complexity separately.
>
> **Invertibility.**
> - The matrix $\boldsymbol{V}$, which is assumed to be invertible in Assumption 3.6, is indeed invertible in the multiclass, multilabel classification, and ranking cases. The specific forms of $\boldsymbol{V}$ are given in Section 2.4.
> - Regarding the matrix $\boldsymbol{Q}$, it is invertible in the multiclass and multilabel classification cases; we confirm that its minimum eigenvalue is positive in Appendix D.4. Even if the $\boldsymbol{Q}$ is not invertible (i.e., ii-a in Assumption 3.6 fails to hold), the same result can be established under the alternative condition (ii-b), which is indeed true in the case of ranking.
>
> **Complexity.**
>
> Let us clarify the complexity of decoding and the computation of the pseudo-inverse of $\boldsymbol{P}_t$.
>
> - When decoding with RDUE (Algorithm 2), the dominant cost lies in computing $\hat{\boldsymbol{y}}\_\Omega(\boldsymbol{\theta})$, which is a solution to convex optimization over $\mathrm{conv}(\mathcal{Y})$ as described in Proposition 2.3. This is thoroughly addressed in Sakaue et al. (2024, [35] Section 3.1). In short, $\hat{\boldsymbol{y}}\_\Omega(\boldsymbol{\theta})$ can be computed via a Frank–Wolfe-type algorithm (e.g., Garber and Wolf, 2021, "Frank–Wolfe with a nearest extreme point oracle"), which typically requires $\tilde{O}(d^2)$ linear optimization steps over $\mathrm{conv}(\mathcal{Y})$; let $\tau\_{\mathrm{LO}}$ denote the time for each step.
>
> - We now explain the computation of the pseudo-inverse of $\boldsymbol{P}\_t = \mathbb{E}\_{\boldsymbol{y} \sim p\_t}[\boldsymbol{y} \boldsymbol{y}^\top]$. This is written as the sum of $\mathbb{E}\_{\boldsymbol{y} \sim p}[\boldsymbol{y} \boldsymbol{y}^\top]$ and $\mathbb{E}\_{\boldsymbol{y} \sim \mu}[\boldsymbol{y} \boldsymbol{y}^\top]$, where $p$ is a distribution over $\mathcal{Y}$ induced by RDUE and $\mu$ is the uniform distribution over $\mathcal{Y}$. The latter can be calculated analytically in the multiclass and multilabel classification and ranking cases. For the former, when $\hat{\boldsymbol{y}}\_\Omega(\boldsymbol{\theta})$ is obtained via the Frank–Wolfe algorithm, $p$ is obtained from the convex combination coefficients, whose support size is at most $O(d)$ as implied by Carathéodory’s theorem (cf. Wirth et al. 2024, "The Pivoting Framework: Frank-Wolfe Algorithms with Active Set Size Control"). Therefore, we can compute $\boldsymbol{P}\_t$ in $O(d^3)$ time, and the pseudo-inversion takes the same order of complexity.
>
> To conclude, the total per-round complexity is typically $O(d^2 \tau_{\mathrm{LO}} + d^3)$, which is feasible and arguably efficient given that $\mathcal{Y}$ may consist of up to $2^d$ labels. We will elaborate further on this point in the revised manuscript.
>
> ## **Q1. Regarding $T\ge B^2K$**
> > 1. In Section 3.4, the algorithm use RUDE decoding with $T>>B^2K$ and claimed that this is only for simplicity. Could you elaborate on why this makes things easier and how this can be lifted?
>
> Since the comment concerns the assumption $T \ge B^2K$, we suppose it pertains to Section 3.3 rather than Section 3.4, and we respond accordingly.
>
> The assumption $T \ge B^2K$ is made solely to simplify the presentation. It allows us to consider only the typical scenario where $T$ is sufficiently large to ensure $q \le 1$, which leads to a simpler expression of the regret bound in the main text.
> This assumption can be entirely removed by setting $q = \min\\{B\sqrt{K/T}, 1/2\\}$ and adding an additive term of $O(K)$ to the regret upper bound in Theorem 3.5. Please let us know if this is not the kind of response the reviewer was expecting.
>
> ## **Q2. Bandit feedback vs. indicator feedback**
> > 2. In Section 3.4, the paper defines bandit feedback as observing the scalar loss $L\_t(y\_t)$, but then states the algorithm only uses the indicator $\mathbb{I}[\hat{y_t}=y_t]$, then does the algorithm still holds for general bandit feedback?
>
> Yes, in Section 3.3, we only use the information of $\mathbb{1}[\hat{\boldsymbol{y}\_t}=\boldsymbol{y}\_t]$, and thus the same result can be obtained under a more general bandit feedback setting where only the 0-1 loss value is observed (see Remark 3.4). Specifically, since $L\_t(\hat{\boldsymbol{y}\_t}) = L(\hat{\boldsymbol{y}\_t}; \boldsymbol{y}\_t)$ is non-negative and takes zero if and only if $\hat{\boldsymbol{y}\_t} = \boldsymbol{y}\_t$ as in Assumption 2.1, $\mathbb{I}[L\_t(\hat{\boldsymbol{y}\_t}) > 0]$ is identical to the 0-1 feedback. On the other hand, in Section 3.4, observing the loss $L\_t(\hat{\boldsymbol{y}\_t})$ is crucial. This is because we use the actual loss values to compute a pseudo-inverse matrix estimator, which allows us to improve the dependency on $K$.
>
> ## **Q4. Regarding non-uniform delay**
> > 4. In practice, I am assuming $D$ can be non uniform across steps and unknown to the algorithm beforehand. But here the algorithm needs to know in advance. Can this assumption be removed?
>
> Regarding the setting where delays can be non-uniform (which we refer to as variable delays), we discuss it in Appendix G, as mentioned in line 379. Due to space limitations, we could not include this discussion in the main text.
>
> Moreover, the assumption that the delay $D$ is known in advance is not necessary. Since the ODAFTRL algorithm determines the decision point using all observed feedback, the value of $D$ becomes known from the time the first feedback is observed. We acknowledge that the current manuscript may have given the impression that $D$ must be known beforehand, and we will revise this part to clarify the explanation accordingly.

---

> > ### Comment · Reviewer_vmLL · 2025-08-06
> >
> > Thank you for the detailed response and they have sufficiently addressed my concerns.
> >
> > I maintain my positive recommendation of this work.

---

### Official Review · Reviewer_CFW5 · 2025-07-07

**Clarity:** 3
**Significance:** 2
**Originality:** 3
**Rating:** 4
**Confidence:** 3

**Summary:**

This paper studies the Online Structured Prediction problem, specifically in two problem settings: (i) bandit feedback and (ii) delayed feedback. For the bandit setting, the paper presents an algorithm with a surrogate regret bound of $O(\sqrt{KT})$. For the delayed feedback setting, the regret is $O(\sqrt{(K + D)T}$.

**Questions:**

1. Could you explain the multiclass classification model in lines 168-173? Why is $L(y; e_i)= \indic[y \neq y]$? The indicator function evaluates to 0 always in the presented form

2. The paper motivates the structured learning problem through several examples, including multiclass classification, multilabel prediction, and ranking. However, the practical significance of the proposed model remains unclear. It is not evident how this approach offers tangible benefits over standard methods in real-world scenarios. For instance, in the case of multiclass prediction, given the setup described, wouldn’t the algorithm risk learning a model that simply overfits to the training data?

3. The paper mentions the bandit setting (with incomplete information) as a key extension to the past works. However, [38] and others have studied the bandit setting. Could you explain and position the contribution w.r.t the existing literature?

4. Is the key contribution the improvement on the sublinear dependence on K? If so, given that the models considered in existing works are more general, could you explain how essentially the algorithm and proof techniques achieve the \sqrt{K} improvement beyond the simplified problem setting compared to other works?

5. The regret bound in Theorem 4.3 is different from the bound mentioned in the abstract and introduction of the papers? Can you explain?

6. Could you explain Assumption 3.2? What is the meaning of L_t(\hat{y}_t)?

7. Can you provide experimental results to support the results and improvement over existing approaches?

8. The paper claims the bound I Theorem 4.3 is independent of $T$? However, there is a dependence on $T$.

9.  Can you explain the technical challenges and the novelty in the proof approach for the delayed feedback extension?

**Ethical Concerns:**

["NO or VERY MINOR ethics concerns only"]

**Final Justification:**

I am raising the score since the author rebuttal addressed all point sin the review. The surrogate vs. general loss discussion seems to be critical for this work after reviewing the discussion with Rev  4VTE.

**Limitations:**

Yes

**Paper Formatting Concerns:**

None.

**Quality:**

3

**Strengths And Weaknesses:**

Strengths:

1. The paper is well written and easy to read.
2. The paper addresses an interesting problem setting (parts of it are unclear though).

Weaknesses:

1. There are multiple errors through the paper; the paper warrants a careful revision.

2. The abstract mentions that algorithms are numerically tested; however, there are no numerical results provided in the paper.

3. The delayed feedback extension seems to be straightforward. They key contributions are not fully cleara in the current presentation.

---

> ### Author Rebuttal · Authors · 2025-07-31
>
> Thank you very much for taking the time to provide such a thoughtful and detailed review. Below, we present our responses to your comments.
>
>
> ## **Q1. Multiclass classification model**
> > 1. Could you explain the multiclass classification model in lines 168-173? Why is $L(y;e_i)=\indic[y\neq y]$? The indicator function evaluates to $0$ always in the presented form.
>
> We apologize for the confusion caused by the typographical error. The correct definition is $L(\boldsymbol{y};\mathbf{e}\_i)=\mathbb{1}[\boldsymbol{y}\neq \mathbf{e}\_i]$. We will correct this in the revised version.
>
> ## **Q2. Practical significance**
> > 2. The paper motivates the structured learning problem through several examples, including multiclass classification, multilabel prediction, and ranking. However, the practical significance of the proposed model remains unclear.
>
> In practice,  structured prediction arises across various domains, from natural language processing to bioinformatics (BakIr et al., 2007, [2]), as discussed in Section 1. In particular, our online setting with efficient linear estimators is well suited for applications such as ad click prediction (McMahan et al., 2013, “Ad Click Prediction: A View from the Trenches”) and ranking in recommendation systems (Rendle et al., 2009, “BPR: Bayesian Personalized Ranking from Implicit Feedback”), where both real-time performance and efficiency are critical.
>
> > It is not evident how this approach offers tangible benefits over standard methods in real-world scenarios. For instance, in the case of multiclass prediction, given the setup described, wouldn’t the algorithm risk learning a model that simply overfits to the training data?
>
> Thank you for the question. We are not entirely sure what is meant by "standard method" in this context, so we would appreciate clarification if possible. If the question refers to batch learning, then the problem setting itself is fundamentally different, making direct comparison difficult.
>
> Our work considers the online learning setting, in which issues such as overfitting—commonly discussed in statistical learning—do not arise. Unlike batch learning, where a predictor is trained using a given training dataset and evaluated on a test dataset, online learning assumes that data arrives sequentially and the model (in our case, matrix $\boldsymbol{W}\_t$) is updated in an online manner. Therefore, there is no distinction between training and test data in online learning, and thus the overfitting risk present in batch learning does not exist in our setting.
>
> ## **Q3. Relation to existing literature**
> > 3. The paper mentions the bandit setting (with incomplete information) as a key extension to the past works. However, [38] and others have studied the bandit setting. Could you explain and position the contribution w.r.t the existing literature?
>
> Our work addresses a more general problem class than those in prior studies. Specifically, while Van der Hoeven (2020, [38]) and Van der Hoeven et al. (2021, [39]) focused on online **multiclass classification** under bandit feedback, our work addresses online **structured prediction**, which includes multiclass classification as a special case. At the same time, our work can be seen as an extension of the existing study on online structured prediction with full-information feedback (Sakaue et al., 2024, [35]) to bandit and delayed feedback.
>
> ## **Q4. Improvement of the dependence on $K$**
> > 4. Is the key contribution the improvement on the sublinear dependence on $K$?
>
> Yes, one of our main contributions is establishing the $O(\sqrt{KT})$ regret bound, improving upon the $O(K\sqrt{T})$ bound of Van der Hoeven (2020, [38]) and Van der Hoeven et al. (2021, [39]). We would like to remark that another, more important contribution is the regret upper bound that is explicitly independent of $K$, achieved through the use of the pseudo-inverse matrix estimator.
>
> > If so, given that the models considered in existing works are more general,
>
> We would like to clarify that the statement “the models considered in existing works are more general” is not accurate. As explained in our response to the third question, we address structured prediction, which is a more general framework than multiclass classification considered in Van der Hoeven (2020, [38]) and Van der Hoeven et al. (2021, [39]).
> One caveat is that these previous studies cover a broader class of surrogate losses, including the (smooth) hinge loss. That said, our work—based on the Fenchel–Young loss framework, following Sakaue et al. (2024, [35])—subsumes the standard logistic loss as a special case. To be precise, while we focus on the logistic loss with base $2$, their framework also accommodates the base-$K$ logistic loss. We believe that we have adequately addressed this subtle difference in the main text. Indeed, Reviewer 4Bpe noted, “I particularly appreciated the authors for their clear (and honest) comparison with related work (e.g., discussion on pages 261–269).”
>
> > could you explain how essentially the algorithm and proof techniques achieve the $\sqrt{K}$ improvement beyond the simplified problem setting compared to other works?
>
> The $\sqrt{K}$ improvement leverages the following techniques:
> - Extending the randomized decoding algorithm, originally designed for full-information feedback, to a form applicable to the bandit feedback setting (RDUE, Algorithm 2).
> - Using RDUE, we show that in the bandit feedback setting, the constant $a$ in Assumption 3.2, which appears as a $1/a$ factor in the regret bound, can be made smaller compared to Van der Hoeven (2020, [38]), based on Lemma 3.1.
> - Exploiting the negative term by using the inequality $||\boldsymbol{G}\_t||\_\mathrm{F}^2 \le bS\_t(\boldsymbol{W}\_t)$ in the proof of Theorem 3.5.
> - We carefully set the mixing probability $q$ of uniform exploration (Line 276).
>
> ## **Q5. Regret bound in Theorem 4.3**
> > 5. The regret bound in Theorem 4.3 is different from the bound mentioned in the abstract and introduction of the papers? Can you explain?
>
> There might be a misunderstanding regarding the upper bound stated in the abstract. The upper bound in Theorem 4.3 pertains to the **delayed** full-information feedback setting, whereas the bound in the abstract refers to the **non-delayed** bandit feedback setting. There is no inconsistency between the main text and the abstract. In the revised version, we will modify Line 8 of the abstract to read "for non-delayed bandit feedback" instead of "for bandit feedback", to avoid confusion.
> ## **Q6. Assumption 3.2**
>
> > 6. Could you explain Assumption 3.2?
>
> This assumption ensures that a decrease in the surrogate loss leads to a proportional decrease in the target loss. It plays a crucial role in establishing an upper bound on the target loss via the surrogate loss. Note that we refer to it as an “Assumption” merely for ease of reference; it is in fact directly guaranteed by Lemma 3.1.
>
> Importantly, one must ensure that the parameter $a$ lies in the interval $(0, 1)$. As discussed in Lines 240–242, this condition is satisfied in our structured prediction setting due to Sakaue et al. (2024, [35]) with $a = 1 - \frac{4\gamma}{\lambda \nu}$. In the case of multiclass classification, Van der Hoeven et al. (2021, [39], Lemma 1) also provide a decoding method that guarantees this condition, with $a = 1/K$.
>
> > What is the meaning of $L\_t(\hat{y}\_t)$?
>
> $L\_t(\boldsymbol{y})=L(\boldsymbol{y};\boldsymbol{y}\_t)$ is a simplified notation for the loss function at time $t$. We defined it in Line 164 and in the notation table (Appendix A). In the revised version, we will add a reminder when it reappears in a distant section, in order to improve readability.
>
> ## **Q7. Experiments**
> > The abstract mentions that algorithms are numerically tested; however, there are no numerical results provided in the paper.
>
> > 7. Can you provide experimental results to support the results and improvement over existing approaches?
>
> As referenced in Lines 70–72 and Lines 84–86, we have included experiments in Appendix H. Due to space constraints, we were unable to include the experimental results in the main text. Given the extra space in the camera-ready version, we plan to include (part of) the experiments.
>
> ## **Q8. Independence of $T$ in Theorem 4.3**
>
> > 8. The paper claims the bound I Theorem 4.3 is independent of $T$? However, there is a dependence on $T$.
>
> Note that the upper bound involves $\min$. Since $\min\\{D^2, D^{2/3}T^{1/3}\\}\le D^2$ holds, the upper bound in Theorem 4.3 is indeed independent of $T$.
>
> ## **Q9. Technical novelty in delayed feedback**
> > 9. Can you explain the technical challenges and the novelty in the proof approach for the delayed feedback extension?
>
> We believe that exploring delayed feedback in online structured prediction is a novel direction in its own right, even though it is not the primary focus of our paper (this topic occupies only one page in our 9-page manuscript). To our knowledge, delayed feedback in this setting has received very limited attention in the literature, despite its practical importance (see also the discussion in Lines 44–48), which makes it challenging due to the scarcity of directly applicable tools. To address this challenge, we have uncovered the benefit of bridging two previously separate lines of research: online convex optimization with delayed feedback and online structured prediction. Specifically, by leveraging AdaGrad-type regret bounds under delay, we show that a finite regret bound can be achieved in the full-information setting, and that in the bandit feedback setting, our approach yields tighter bounds than the $D$-copy strategy (see also our rebuttal to Reviewer jZKi).

---

> > ### Author Response · Authors · 2025-08-07
> >
> > Dear Reviewer CFW5,
> >
> > Thank you again for the time and care you have devoted to evaluating our work.
> > In our rebuttal we aimed to address each of the weaknesses and questions you raised in your initial review. We would be grateful if you could let us know whether our responses satisfactorily resolve your concerns. Should any aspect remain unclear or insufficiently answered, please feel free to point it out. We will gladly provide further clarification.
> >
> > As the discussion period ends in only two days, any response before the deadline would be invaluable to us.
> >
> > Thank you again for your time and consideration.

---

### Decision · Program_Chairs · 2025-09-17

**Decision:**

Accept (poster)

**Comment:**

This paper looks at the online Structured Prediction with bandit or delayed feedback, with interesting results on the regret growth, in particular with delays.

All the reviewers are now very positive, thanks to the rebuttal that convinced them. Due to this, I went through the paper myself, and I can see why they are all on the acceptance side, on I concur with them, this is an interesting paper with promising result.

I therefore also recommend acceptance.